# Identification of genomic enhancers through spatial integration of single-cell transcriptomics and epigenomics

Carmen Bravo González-Blas[1,2] (iD), Xiao-Jiang Quan[1,2] (iD), Ramon Duran-Romaña[1] (iD), Ibrahim Ihsan Taskiran[1,2] (iD), Duygu Koldere[1,2], Kristofer Davie[1] (iD), Valerie Christiaens[1,2], Samira Makhzami[1,2], Gert Hulselmans[1,2] (iD), Maxime de Waegeneer[1,2] (iD), David Mauduit[1,2] (iD), Suresh Poovathingal[1] (iD), Sara Aibar[1,2] (iD) & Stein Aerts[1,2,*] (iD)

## Abstract

**Single-cell technologies allow measuring chromatin accessibility and gene expression in each cell, but jointly utilizing both layers to map *bona fide* gene regulatory networks and enhancers remains challenging. Here, we generate independent single-cell RNA-seq and single-cell ATAC-seq atlases of the *Drosophila* eye-antennal disc and spatially integrate the data into a virtual latent space that mimics the organization of the 2D tissue using ScoMAP (Single-Cell Omics Mapping into spatial Axes using Pseudotime ordering). To validate spatially predicted enhancers, we use a large collection of enhancer–reporter lines and identify ~ 85% of enhancers in which chromatin accessibility and enhancer activity are coupled. Next, we infer enhancer-to-gene relationships in the virtual space, finding that genes are mostly regulated by multiple, often redundant, enhancers. Exploiting cell type-specific enhancers, we deconvolute cell type-specific effects of bulk-derived chromatin accessibility QTLs. Finally, we discover that Prospero drives neuronal differentiation through the binding of a GGG motif. In summary, we provide a comprehensive spatial characterization of gene regulation in a 2D tissue.**

**Keywords** enhancer detection; eye-antennal disc; gene regulation; single-cell omics; spatial integration

**Subject Categories** Chromatin, Transcription & Genomics; Computational Biology; Development

**Mol Syst Biol. (2020) 16: e9438**

See also: **J Carnesecchi & I Lohmann** (May 2020)

## Introduction

Cellular identity is defined by Gene Regulatory Networks (GRNs), in which transcription factors bind to enhancers and promoters to regulate target gene expression, ultimately resulting in a cell type-specific transcriptome. Single-cell technologies provide new opportunities to study the mechanisms underlying cell identity. Particularly, single-cell transcriptomics allow measuring gene expression in each cell, while single-cell epigenomics, such as single-cell ATAC-seq (Assay for Transposase-Accessible Chromatin using sequencing), serves as a read-out of chromatin accessibility (Fiers *et al*, 2018). Although these technologies and computational approaches are recently evolving to include spatial information (Karaiskos *et al*, 2017; Eng *et al*, 2019; Nitzan *et al*, 2019; Rodriques *et al*, 2019; Thornton *et al*, 2019), most approaches currently target single-cell transcriptomes. It remains a challenge how to exploit single-cell epigenomic data for resolving spatiotemporal enhancer activity and GRN dynamics, both experimentally and computationally.

In addition, while ATAC-seq is a powerful tool for predicting candidate enhancers, not all accessible regions correspond to functionally active enhancers (Shlyueva *et al*, 2014). For example, accessible sites can correspond to ubiquitously accessible promoters or binding sites for insulator proteins (Xi *et al*, 2007); to repressed or inactive regions due to binding of repressive transcription factors (Gary & Levin, 1996; Li & Arnosti, 2011; Arnold *et al*, 2013; Shlyueva *et al*, 2014); or to primed regions that are accessible across a tissue, but become only specifically activated in a subset of cell types (Jacobs *et al*, 2018). Importantly, single-cell ATAC-seq has not been fully exploited to explore these aspects yet. While most scATAC-seq studies have been carried out in mammalian systems, in which enhancer testing is not trivial, Cusanovich *et al* (2018) evaluated 31 cell type-specific enhancers predicted from scATAC-seq in the *Drosophila* embryo, finding that ~ 74% showed the expected activity patterns.

1 VIB Center for Brain & Disease Research, Leuven, Belgium
2 Department of Human Genetics, KU Leuven, Leuven, Belgium
*Corresponding author. Tel: +3216330710; E-mail: stein.aerts@kuleuven.vib.be

Another current challenge in the field of single-cell regulatory genomics is how to integrate epigenomic and transcriptomic information. Although some experimental approaches have been developed for profiling both the epigenome and the transcriptome of the same cell (Cao *et al*, 2018; Chen *et al*, 2019b; Liu *et al*, 2019), currently either the quality of the measurements, or the throughput, is still significantly lower compared to each independent single-cell assay. For example, sci-CAR (Single-cell Combinatorial Indexing Chromatin Accessibility and mRNA) or SNARE-seq (Single-Nucleus Chromatin Accessibility and mRNA Expression sequencing) on human cells achieved a median of 1,000–4,000 UMIs (Unique Molecular Identifiers) and 1,500–3,000 fragments per cell, while the coverage with non-integrative methods, such as 10x, is around 20,000 UMIs and 10,000 fragments per cell for scRNA-seq and scATAC-seq, respectively (Cao *et al*, 2018; Chen *et al*, 2019b; preprint: Pervolarakis *et al*, 2019). Methods that achieve high sensitivity, such as scCAT-seq (single-cell Chromatin Accessibility and Transcriptome sequencing) (Liu *et al*, 2019), are based on microwell plates rather than droplet microfluidics, making their throughput limited.

Given the current limitations of combined omics methods, the computational integration of independent high-sensitivity assays provides a valuable alternative. For example, Seurat (Stuart *et al*, 2019) and Liger (Welch *et al*, 2019) have been used to integrate independently sequenced single-cell transcriptomes and single-cell epigenomes. Nevertheless, these methods require the "conversion" of the genomic region accessibility matrix into a gene-based matrix, and how to perform such a conversion is an unresolved issue. Some studies have used the accessibility around the Transcription Start Site (TSS) as proxy for gene expression (Bravo González-Blas *et al*, 2019); others aggregate the accessibility regions that are co-accessible (i.e., correlated) with the TSS of the gene in a certain space (Pliner *et al*, 2018). However, promoter accessibility is not always correlated with gene expression. Furthermore, enhancers can be located very far from their target genes—upstream or downstream, up to 1 Mbp in mammalian genomes, or up to 100–200 kb in *Drosophila*, often with intervening non-target genes in between—and relationships between enhancers and target genes are often not one-to-one (i.e., an enhancer can have multiple targets, and a gene can be regulated by more than one enhancer) (Shlyueva *et al*, 2014). Enhancer–promoter interactions can also be predicted using Hi-C approaches at the bulk level (Ghavi-Helm *et al*, 2019); however, these methods have limited sensitivity at single-cell resolution (Nagano *et al*, 2015).

The *Drosophila* third-instar larval eye-antennal disc provides an ideal biological system for the spatial modeling of gene regulation at single-cell resolution. The eye-antennal disc comprises complex, dynamic, and spatially restricted cell populations in two dimensions. The antennal disc consists of four concentric rings (A1, A2, A3, and arista), each with a different transcriptome and different combinations of master regulators. For example, both Hth and Cut regulate the outer antennal rings (A1 and A2), with additional expression of Dll in A2, while Dll, Ss, and Dan/Danr are key for the development of the inner rings (A3 and arista), among others (Dong *et al*, 2002; Emerald *et al*, 2003). On the other hand, a continuous cellular differentiation process from anterior to posterior occurs in the eye disc, in which progenitor cells differentiate into neuronal (i.e., photoreceptors) and non-neuronal (i.e., cone cells, bristle, and pigment cells) cell types. This differentiation wave is driven by the morphogenetic furrow (MF). Posterior to the MF, R8 photoreceptors are specified first, and then, they sequentially recruit R2/R5, R3/R4, and R7 photoreceptors and cone cells to form hexagonally packed units called ommatidia (Roignant & Treisman, 2009; Fig 1A). In summary, the heterogeneity of cell types and differentiation trajectories results in diverse—static and dynamic—GRNs, which can be modeled with a combination of experimental and computational approaches.

In this work, we first generate a scRNA-seq and a scATAC-seq atlas of the eye-antennal disc. Second, taking advantage of the fact that the disc proper is a 2D tissue, we spatially map these single-cell profiles on a latent space that mimics the eye-antennal disc, called the virtual eye-antennal disc. Next, by exploiting publicly available enhancer–reporter data (Jory *et al*, 2012), we assess the relationship between enhancer accessibility and activity. Third, we use these virtual cells, for which both epigenomic and transcriptomic data are available, to derive links between enhancers and target genes using a new regression approach. Fourth, we use a panel of 50 bulk ATAC-seq profiles across inbred lines to predict cell type-specific caQTLs (chromatin accessibility QTLs). Finally, we use our findings to characterize the role of Prospero in the accessibility of photoreceptor enhancers. In summary, we provide a comprehensive characterization of gene regulation in the eye-antennal disc, using a strategy that is applicable to other tissues and organisms. Our results can be explored as a resource on SCope (Davie *et al*, 2018) (http://scope.ae rtslab.org/#/Bravo_et_al_EyeAntennalDisc) and the UCSC Genome Browser (http://genome.ucsc.edu/s/cbravo/Bravo_et_al_EyeAnte nnalDisc), and we provide an R package, called ScoMAP (Single-Cell Omics Mapping into spatial Axes using Pseudotime ordering), to spatially integrate single-cell omics data and infer enhancer-to-gene relationships (https://github.com/aertslab/ScoMAP).

## Results

### A single-cell transcriptome atlas of the eye-antennal disc

First, we set out to identify the different cell populations in the eye-antennal disc and obtain their transcriptomes. We profiled 3,531 high-quality cells using scRNA-seq on three runs of the 10x Genomics platform, with a median of 20,761 UMIs and 3,094 expressed genes per cell, respectively (see Materials and Methods, Appendix Fig S1A and B). Analysis with Seurat (Stuart *et al*, 2019) revealed 17 clusters, most of which map to spatially located cell types (Fig 1B). Importantly, the structure in the tSNE (t-distributed Stochastic Neighbor Embedding)—and UMAP (Uniform Manifold Approximation and Projection) (Appendix Fig S1C), reveals two main branches, one corresponding to the antennal disc, in which clusters represent the antennal rings from outer to inner; and one corresponding to the eye disc, in which progenitors differentiate into ommatidial (i.e., photoreceptors and cone cells) and interommatidial cell types. We verified that cell clustering was driven by cell identity and not affected by batch effects (Appendix Fig S1D). We also found a subset of ommatidial cells with a high number of UMIs and genes expressed, which was annotated as doublets by DoubletFinder (McGinnis *et al*, 2019; Appendix Fig S1E–G). The higher proportion of doublets in this group is not unexpected, since ommatidia are tightly packed and are more difficult to dissociate.

   

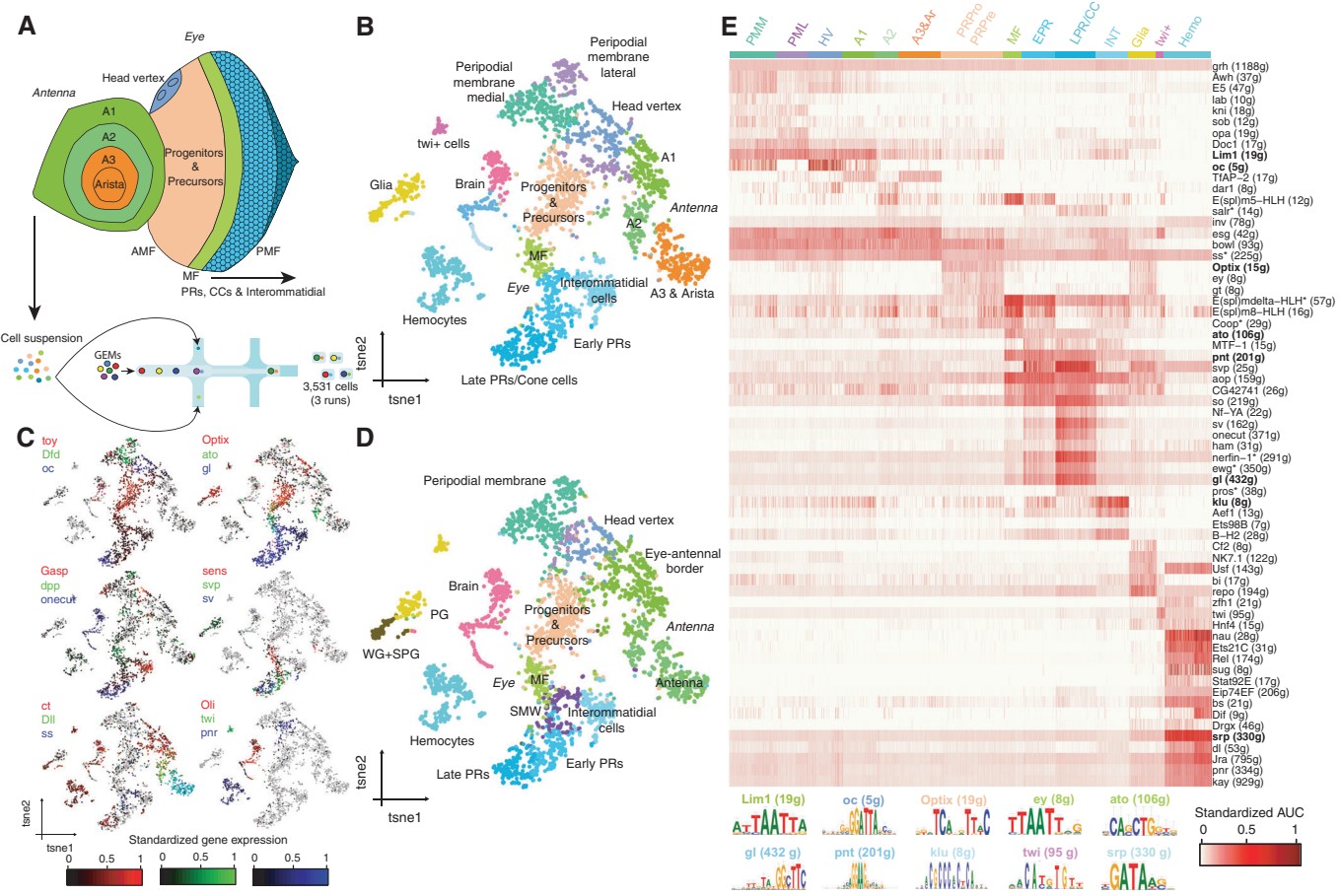

**Figure 1. scRNA-seq recapitulates cellular diversity and GRNs in the eye-antennal disc.**

A Experimental approach. scRNA-seq was performed in eye-antennal discs using 10x Genomics, resulting in a data set with 3,531 high-quality cells. Main spatial compartments in the eye-antennal disc are annotated.

B tSNE representation of the scRNA-seq data (with 3,531 cells).

C tSNE colored by the standardized gene expression of known cell type markers in the eye-antennal disc. In each plot, three marker genes are shown, using RGB encoding.

D tSNE annotated by label transfer with Seurat v3 (Stuart *et al*, 2019) using the scRNA-seq eye disc data set from Ariss *et al* (2018).

E Cell-to-regulon heatmap showing the standardized enrichment or area under the curve (AUC) from SCENIC (Aibar *et al*, 2017) for each selected regulon based on RSS in each cell. Top enriched motifs for representative regulons are shown below. Regulons marked with * are based on ChIP-seq track enrichment.

Data information: AMF: anterior to the morphogenetic furrow. Ar: arista. CC: cone cells. EPR: early photoreceptors. Hemo: hemocytes. HV: head Vertex. INT: interommatidial cells. LPR/CC: late photoreceptors and cone cells. MF: morphogenetic furrow. PG: perineurial glia. PMF: posterior to the morphogenetic furrow. PML: peripodial membrane lateral. PMM: peripodial membrane medial. PR: photoreceptors. PRPre: photoreceptor precursors. PRPro: photoreceptor progenitors. SMW: second mitotic wave. SPG: subperineurial glia. WG: wrapping glia.

To annotate these 17 cell clusters, we combined two approaches. First, we used known marker genes from literature (Fig 1C, Appendix Fig S1H). For example, we find *Dfd* expressed in the peripodial membrane clusters, with *dpp* expressed in the lateral peripodial membrane (Stultz *et al*, 2012) and *oc* as key marker of the head vertex (Blanco *et al*, 2010). In the eye disc, we find a gene expression gradient starting from *Optix* expression in progenitors and precursors, to *ato* expression in the MF, and then *gl* expression in the ommatidial and interommatidial cells. Importantly, we find *Gasp* as key marker of the interommatidial cells (Fig 1C), which plays a role in extracellular matrix integrity and assembly (Tiklová *et al*, 2013). Indeed, Gene Ontology (GO) enrichment of the genes differentially expressed in this group reveals terms related to cell–cell junction assembly and organization (*P*-val: $10^{-16}$). Meanwhile,

in the ommatidial groups we observed a gene expression gradient of markers from early photoreceptors (R8, *sens*), to intermediate (R3-4, *svp*), and late-born PRs and cone cells (R7 and cone cells, *sv*) (Mlodzik *et al*, 1990; Frankfort *et al*, 2001; Charlton-Perkins *et al*, 2011). In fact, semi-supervised analysis of these populations (see Materials and Methods) subdivides the ommatidial classes into the different photoreceptor types and cone cells (Appendix Fig S2), largely finding R8, R3/R4, and R1/R6 in the early-born PR cluster, and R7 and cone cells in the late-born photoreceptors and cone cells cluster (only 26 R2/R5 cells are detected). On the other hand, markers of the antennal rings form a gradient along the antennal cell types, from *ct* (A1 and A2), to *Dll* (A2, A3 and arista), and *ss* (A3 & arista) (Emerald *et al*, 2003). Interestingly, within A2, we find a rare subpopulation of cells expressing *ato* and *sens* (0.93%),

corresponding to the Johnston Organ Precursors (JOPs) (Nolo et al, 2000; Eberl & Boekhoff-Falk, 2007), which represent only 0.25–0.6% of the eye-antennal disc cell population (Sen et al, 2010). We also identify a population of glial cells, based on the enrichment of the transcription factor repo (Yuasa et al, 2003); a cluster of hemocytes, enriched for pnr (Minakhina et al, 2011); and a small group of cells with high expression of the transcription factor twi (1.5%), corresponding to adepithelial cells (mesodermal myoblasts), which are known to reside in most imaginal discs (Furlong et al, 2001; Beira & Paro, 2016). Accordingly, GO term enrichment using the differentially expressed genes in this group reveals terms related to mesoderm development (P-val: $10^{-4}$). Finally, we find a population of 299 cells coming from the brain expressing Oli (Oyallon et al, 2012), which represent contaminating cells from the brain due to the dissections.

To validate and further extend our cell type annotations, we used a publicly available Drop-seq data set from the eye disc containing 11,500 single-cell profiles with ~ 1× cellular coverage (Ariss et al, 2018) and a median of 517 genes detected per cell (Appendix Fig S3A and B). Using Seurat's label transferring, we mapped the cell types annotated by Ariss et al to our data set (and vice versa) and found that both annotations agreed (Fig 1D, Appendix Fig S3C–E). These labels permitted to subdivide our glial cell cluster into wrapping glia, subperineural glia, and perineural glia, and to annotate a small population of cells just posterior to the MF as the second mitotic wave (SMW), which is a round of synchronous cell division that occurs right after cells exit the MF. On the other hand, no $twi^{+}$ cells are found in the Drop-seq data set. This is likely due to the fact that these cells are located in the antennal disc, which is missing in the Drop-seq data set. Indeed, the activity of a twi enhancer (Appendix Fig S3F) is observed in the antennal disc rather than in the eye disc (Jory et al, 2012). Altogether, despite the fact that stringent cell filtering across three 10x Genomics runs resulted in only 3.5K cells (Appendix Fig S4A–D), we find that the data set forms a representative sample of all the known cell types in the tissue.

Next, we used SCENIC (Single-Cell rEgulatory Network Inference and Clustering) to identify master regulators and gene regulatory networks in the eye-antennal disc (Aibar et al, 2017), finding 175 regulons (159 motif-based regulons and 16 regulons based on ChIP-seq (Chromatin Immunoprecipitation Sequencing) tracks; see Materials and Methods). Briefly, SCENIC infers co-expression modules between transcription factors (TFs) and candidate target genes using machine learning regression techniques (e.g., random forest or gradient boosting machines), which are pruned based on the enrichment of the TF motif around the TSS of the potential target genes, resulting in regulons. While some regulons are enriched across the entire tissue [such as Grh (Jacobs et al, 2018)], many are cell type-specific (Fig 1E). For the antennal rings, we find Lim1 (A1), TfAP-2 (A1, A3 and arista), and Ss (arista and A3), in agreement with literature (Ahn et al, 2011; Emmons et al, 2007; Tsuji). In the eye disc, regulons recapitulate the GRN dynamics during the differentiation process, with Optix and Ey in the progenitor and precursor cells, Ato in the morphogenetic furrow (Appendix Fig S5A–C), So in photoreceptors, and B-H2 in interommatidial cells (Higashijima et al, 1992; Bessa et al, 2002). We further validated the Ato regulon using GeneMANIA associations (Warde-Farley et al, 2010), as well as previously published RNA-seq data from Ato gain-of-function

and loss-of-function mutants (Aerts et al, 2010; Appendix Fig S5B, D and E). Indeed, genes included in the Ato regulon are significantly upregulated in the Ato gain-of-function (NES: 2.44) and downregulated in the ato mutant (NES: −2.57), respectively, and include known Ato target genes in the MF, such as sens and sca.

In conclusion, using scRNA-seq we have identified the different populations in the eye-antennal disc and the interplay of GRNs that underlie the developmental program of this system. We provide this data as loom files that can be explored in SCope at http://scope.aertslab.org/#/Bravo_et_al_EyeAntennalDisc.

## A single-cell ATAC-seq atlas of the eye-antennal disc

Next, we performed scATAC-seq to explore the chromatin accessibility landscape of the eye-antennal disc. Using 10x chromium in two independent biological replicates, we obtained a ~ 1× coverage of the eye-antennal disc, with 15,766 scATAC profiles (Fig 2A, Appendix Fig S4E and F). We assessed the quality of the data set based on the relative enrichment of fragments around the TSS and the correlation with bulk ATAC-seq on the same tissue ($R^2$ = 0.9615; Fig 2B and C), among other quality control measures, and filtered out a total of 379 cells based on the number of fragments within bulk peaks (< 20%) and the total number of fragments (< 100 or > 10,000, Appendix Fig S6; see Materials and Methods). This resulted in a data set with 15,387 single-cell epigenomes.

Since this tissue comprises differentiating continuous rather than discrete populations, we used cisTopic (Bravo González-Blas et al, 2019), currently the best performing clustering method on dynamic populations (Chen et al, 2019a). Briefly, cisTopic exploits a topic modeling technique, Latent Dirichlet Allocation (LDA), to simultaneously classify regulatory regions into regulatory topics and cluster cells based on their regulatory topic enrichment. In other words, topics are sets of co-accessible regulatory regions across cells, where each region has a probability to belong to each topic (region-topic distribution), and the topic-cell distribution represents the overall accessibility of regions in a topic in a cell. Compared to other methods [such as SnapATAC (preprint: Fang et al, 2019) or Latent Semantic Indexing (Cusanovich et al, 2018)], cisTopic does not require defining discrete cell clusters (which is not trivial in dynamic populations) to identify cell type-specific accessible regions, because region and cell clustering are performed simultaneously. In addition, thanks to its probabilistic nature, cisTopic also works as a drop-out imputation method through the estimation of the probability of each region in each cell [by multiplying the region-topic and topic-cell probabilities (Bravo González-Blas et al, 2019)].

As recent studies have shown a higher accuracy of scATAC-seq clustering when using genomic bins compared to bulk or aggregate peaks (Chen et al, 2019a), we ran cisTopic (Bravo González-Blas et al, 2019) using three different sets of regulatory regions: (i) narrow peaks as called by MACS2 from the bulk ATAC-seq profile of the wild-type Drosophila eye-antennal disc; (ii) bulk peaks defined by extending ±250 bp from the summits called by MACS2; and (iii) cisTarget regions, defined by partitioning the entire noncoding Drosophila genome based on cross-species conservation, resulting in more than 136,000 bins with an average size of 790 bp (Herrmann et al, 2012). Accordingly, we found that the cisTopic analysis performed with cisTarget regions resulted in the highest resolution compared to using bulk peaks or summits (Appendix Fig S7).

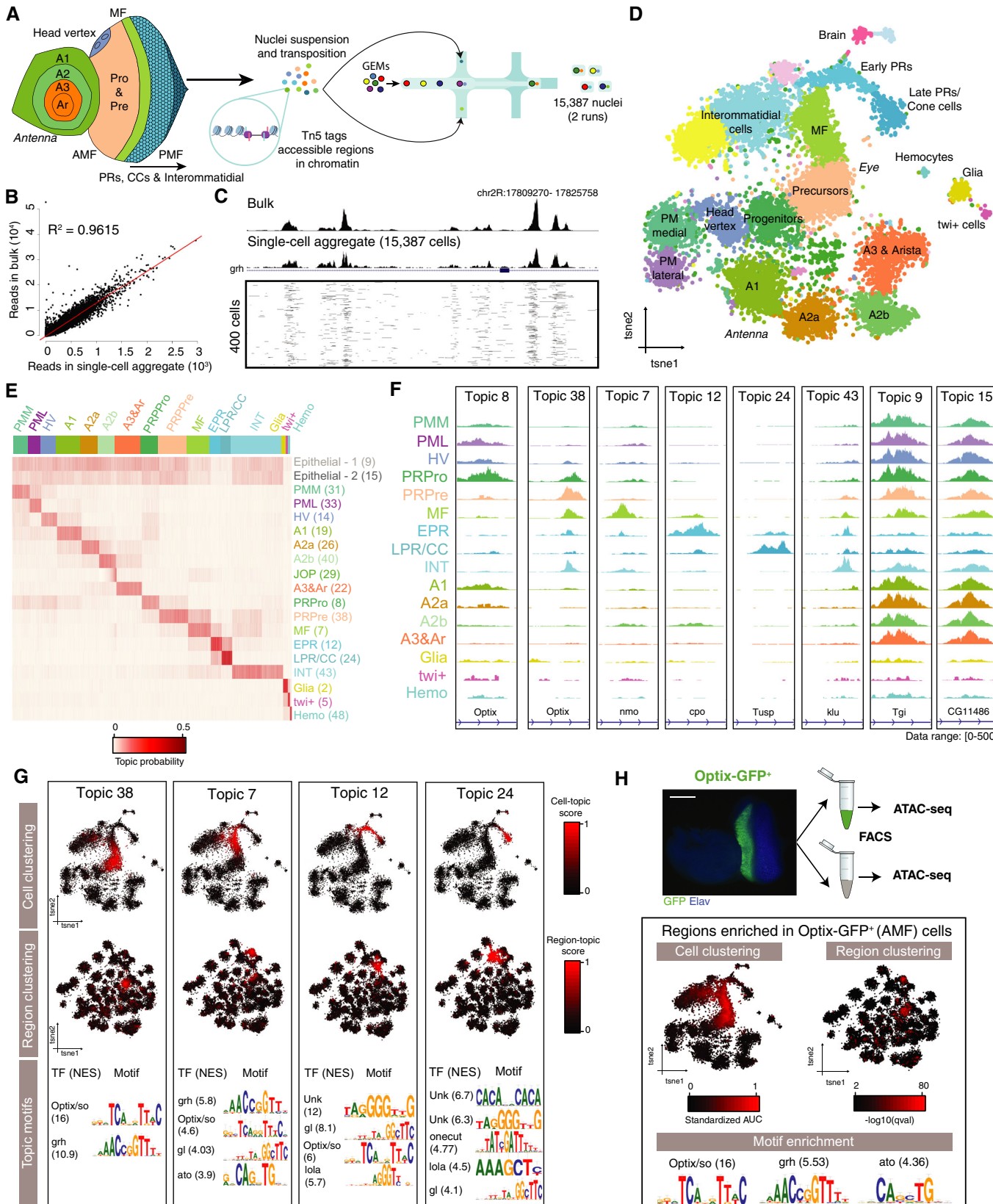

**Figure 2.**

◀

**Figure 2.  scATAC-seq recapitulates cell diversity in the eye-antennal disc.**

A   Experimental set up. 15,387 nuclei were profiled using 10x scATAC-seq.
B   Correlation between the accessibility of regions in the bulk ATAC-seq and the aggregated single-cell profiles.
C   Comparison of bulk ATAC profiles, scATAC-seq aggregate with 10x and 400 individual cells, where each row represents a cell and fragments are shown in black. The number of cells in the aggregate is indicated between brackets.
D   cisTopic cell tSNE (15,387 nuclei) colored by annotated cell type.
E   Topic-cell enrichment heatmap with selected topics.
F   Aggregate profiles per cell type in the top region of the indicated topic.
G   Topic modeling recapitulates the dynamic chromatin changes during differentiation in the eye disc. Top: cisTopic cell tSNE colored by topic enrichment. Middle: cisTopic region tSNE colored by topic enrichment. Bottom: Selected enriched motifs in each topic.
H   Bulk ATAC was performed on Optix-GFP$^+$ and Optix-GFP$^-$ FACS sorted cells (based on the activity of the Optix2/3 enhancer). cisTopic cell tSNE and region tSNE are colored based on the enrichment of regions that are differentially accessible between Optix-GFP$^+$ and Optix-GFP$^-$. Motifs enriched in the regions differentially accessible in Optix-GFP$^+$ cells are shown. Scale bar, 100 μm.

Data information: Ar: arista. EPR: early photoreceptors. Hemo: hemocytes. HV: head vertex. INT: interommatidial cells. JOP: Johnston organ precursor. LPR/CC: late photoreceptors and cone cells. MF: morphogenetic furrow. PML: peripodial membrane lateral. PMM: peripodial membrane medial. PRPre: photoreceptor precursors. PRPro: photoreceptor progenitors.

For example, small subpopulations such as brain and $twi^+$ cells could be revealed, which were otherwise mixed with ommatidial and glial cells, respectively. Hence, we used this model, with 49 topics, for further analysis.

Clustering on the topic-cell distributions (i.e., the contribution of each regulatory topic to each cell) resulted in 22 clusters, most of which map to spatially located cell types (Fig 2D). Despite the fact that cell clustering is not driven by read coverage, we find two groups that likely correspond to doublets based on read depth and percentage of reads in peaks (Appendix Fig S8A–C). Annotation of cell types from scATAC-seq is not as straightforward as for scRNA-seq, because the cluster markers now represent regulatory regions instead of genes. To address this, we exploit four different approaches: (i) motif (and ChIP-seq track) enrichment on the regulatory topics; (ii) enrichment of epigenomic signatures of FAC-sorted cell types; (iii) a novel method for deriving gene activity scores from cisTopic distributions; and (iv) label transferring from our previously annotated scRNA-seq data set.

Of the 49 predicted topics, two represent a batch effect of the run, and one represents a female sex-specific topic (Appendix Fig S8D–I), but these topics do not influence cell clustering (Appendix Fig S8J). The remaining topics represent general, cell type-specific and low contribution topics (Fig 2E and F, Appendix Figs S9 and S10A). Among the cell type-specific topics, we find a topic for each antennal ring (topics 19, 26, 40, and 22; respectively), with a subdivision of A2 into two groups (A2a and A2b, respectively). Regions in these topics, from the outer to the inner ring, are enriched for motifs (and/or ChIP-seq tracks) linked to known master regulators, such as Hth in A1, Dll in A2, and Ss in A3 and arista (Appendix Fig S9). Additionally, we identify a subpopulation of cells in A2b with accessible regions controlled by Ato, which correspond to the Johnston Organ Precursors (JOPs). Similarly, retinal developmental topics recapitulate the dynamic changes in chromatin during differentiation, with the Optix motif enriched in regions specific to the domain anterior to the MF; the Ato motif in MF-specific regions; the Glass, Sine oculis (So), and Onecut motifs in the clusters representing ommatidial cells; and the Glass, So, and Lozenge motifs in interommatidial cell types (Fig 2G). Furthermore, we find a new, highly enriched GGG motif in the genomic regions specific to ommatidial development, which can be linked to a relatively large set of candidate TFs based on motif-to-TF mappings, as will be discussed further below. We

discovered generally accessible topics as well, highly enriched for promoters (Appendix Fig S10B), some of which decrease in accessibility during ommatidial development. These epithelial topics are represented by genomic regions bound by the pioneer transcription factors Trl and Grh, based on motif and ChIP-seq enrichment (Appendix Fig S10C). Indeed, Grh has been shown to be expressed and promote chromatin opening in all epithelial cells, decreasing upon neuronal differentiation (Jacobs et al, 2018), which is also supported by our scRNA-seq data set (Appendix Fig S10D). Furthermore, we identify other cell type-specific topics for other subpopulations, such as a topic enriched for the Twist motif that identifies the $twi^+$ adepithelial cells; a topic enriched for the Serpent (Srp) motif, corresponding to hemocytes; and a topic enriched for the Repo motif, corresponding to glial cells. Finally, we distinguish two small subpopulations with topics enriched for Stripe (Sr, homologous to human neuronal activation factor EGR4), which correspond to brain cells likely attached during the dissection. Overall, despite the experimental and thresholding bias that lead to different sizes of the scRNA-seq atlas (3.5K cells) and the scATAC-seq atlas (15K cells), both data sets reveal the same populations and in similar proportions (Appendix Fig S10E).

To further validate the cell type annotations in the scATAC-seq atlas, we used our previously published ATAC-seq data from FAC-sorted cells located specifically anterior to the morphogenetic furrow, based on the activity of the Optix2/3 enhancer driving GFP (Optix-GFP$^+$; Fig 2H; Ostrin et al, 2006; Jacobs et al, 2018). We find that regions specifically accessible in these cells compared with the rest of the eye are accessible in the precursor cell cluster in the scATAC-seq data, and also show enrichment for the motif of the transcription factor Optix, in agreement with the topic specific to this population. We also re-used our previous single-cell ATAC-seq data, obtained on the Fluidigm C1, of Optix-GFP$^+$ FAC-sorted cells (Jacobs et al, 2018), and we performed an additional Fluidigm C1 run with cells FAC-sorted based on the activity of the sens-F2 enhancer (Pepple et al, 2008) (sens-GFP$^+$), which correspond to the intermediate groups in the MF and R8 photoreceptors, respectively. When mapping these cells into the topic space, we find that they cluster within the correct cell types of the 10x sc-ATAC-seq data (Appendix Fig S11A). Accordingly, we also find that the activity of the Optix 2/3 enhancer and the sens-F2 enhancer agrees with the accessibility of these regions in the matching cell types (Appendix Fig S11B and C).

                                                                        

Next, we developed a new approach for deriving a "gene activity matrix" from the topic-cell and region-topic distributions (Appendix Fig S12A). Briefly, we first multiply the region-topic and topic-cell distributions to obtain a region-cell distribution, which indicates the probability of accessibility of each region in each cell. Then, for each gene, we aggregate the probabilities of the surrounding regions (in this case, 5 kb around the TSS plus introns), resulting in a gene activity score. This new matrix, which contains scATAC-seq cells as columns and gene activities as rows, can be analyzed as a gene expression matrix. For example, we used it to score SCENIC regulons on the scATAC-seq cells to validate the master regulators found in the topics (Appendix Fig S12B). We find the Optix regulon enriched anterior to the morphogenetic furrow, the Ato regulon enriched in the MF, Onecut enriched in late ommatidial cells, and Grh enriched across all cell types except late ommatidial cells. Furthermore, we also used DoubletFinder (McGinnis *et al*, 2019), developed for scRNA-seq data, and labeled a group of cells enriched in both ommatidial and interommatidial topics as doublets (Appendix Fig S12C). In addition, we used this matrix for label transferring with our scRNA-seq data set using Seurat v3 (Stuart *et al*, 2019), finding a strong agreement between our independent RNA and ATAC-based annotations (Appendix Fig S12D and E). Importantly, we find regions enriched for a specific motif that are located in the surroundings of genes (learned from the scATAC-seq data) that are co-expressed with the corresponding transcription factor (learned from the scRNA-seq data), likely representing *bona fide* functional enhancers. For example, we find 2,769 regions enriched for the Optix/So motif, out of which 505 and 894 are in the surroundings of genes co-expressed with *Optix* and *so*, respectively. Similarly, out of the 1,859 and 1,128 regions enriched for the Atonal and the Glass motifs, 285 and 452 are close to co-expressed genes (Dataset EV1).

In summary, we provide a thorough characterization of the chromatin accessibility landscape of the eye-antennal disc, corroborated by our scRNA-seq data set. This data can also be explored at SCope (http://scope.aertslab.org/#/Bravo_et_al_EyeAntennalDisc) and UCSC (http://genome.ucsc.edu/s/cbravo/Bravo_et_al_EyeAntennalDisc).

**Spatiotemporal mapping of single-cell omics couples enhancer accessibility with functionality**

Since most of the cell types in the eye-antennal disc map to locally restricted populations, we developed a strategy to map the scRNA-seq and scATAC-seq profiles to their putative position of origin in the tissue using a template of the eye-antennal disc with 5,058 virtual cells (between the scRNA-seq and the scATAC-seq coverage, see Materials and Methods), corresponding to the 5,058 pixels in our eye-antennal disc representation (Figs 1A, 2A and 3A). The number of pixels was chosen as a middle ground between the sizes of the scRNA-seq and scATAC-seq data sets, but we also evaluated smaller and larger maps (Appendix Fig S13), and this size can be easily changed using the R package ScoMAP (Single-Cell Omics Mapping into spatial Axes using Pseudotime ordering) available at https://github.com/aertslab/ScoMAP.

To place cells in this map, we first order antennal and eye cells based on their transcriptome or epigenome into one-dimension (i.e., *pseudotime*), which correspond to the proximal-distal and anterior–posterior axes in the antenna and the eye,

respectively (Appendix Fig S14). For each cluster, we divide real and virtual cells into bins based on pseudotime and position in the corresponding axis, respectively. Finally, we map real cells onto the virtual cells in the matching bin in the virtual eye-antennal disc, with a 1-to-1 matching. When there are fewer real cells than virtual cells in the bin, real cells are sampled randomly more than once; and when more real cells are available than virtual cells, $N$ real cells are sampled, where $N$ is the number of virtual cells in that bin.

Using the mapped scRNA-seq data, we can visualize previously known gene expression patterns (Fig 3B). For example, our spatial map recapitulates expression of *hth*, *salm*, *danr*, *ct*, *Dll*, and *ss* in the antennal rings, as shown by Emerald *et al* (2003). In the eye part, patterns from anterior to posterior, with the expression of *oc* in the head vertex, *Optix* and *toy* anterior to the MF, *ato* and *dpp* in the MF, and *gl* posterior to the MF, agree with literature (Bessa *et al*, 2002).

To validate the scATAC-seq mapping, we used image data of more than 700 enhancer–reporter lines from the Janelia FlyLight project (Jory *et al*, 2012). In short, in each line a specific enhancer controls the expression of GAL4, and when crossed with a UAS-GFP reporter line, the activity of the enhancer is recapitulated by the GFP signal. These enhancer activity patterns were registered onto the virtual eye-antennal disc using a custom landmark-based pipeline (see Materials and Methods). Importantly, Janelia enhancers are wide regions (with an average size above ~ 3,000 bp) in which the functional enhancer part can often be defined by the scATAC-seq peak (Appendix Fig S15A). However, ~ 75% of the Janelia enhancers include more than one regulatory region, for whose activity correlates either with all or some of the regulatory regions within the enhancer (Appendix Fig S15B and C). Here, we defined the accessibility of a Janelia enhancer as the aggregate accessibility of all regulatory regions that it contains. By comparing the predicted accessibility pattern for each Janelia region (based on the ATAC-seq) with its activity (based on the GFP reporter), we find that accessibility and activity are correlated in 77% of the enhancers (Fig 3C); however, there are cases in which accessibility and activity are uncoupled (Fig 3D).

To assess the cell type-specificity of chromatin accessibility, we calculated an accessibility gini index for each enhancer. The gini index has been widely used in economics to assess income inequality and has been previously used in single-cell transcriptomics for the selection of cell type-specific genes (Jiang *et al*, 2016; Torre *et al*, 2018). In this context, a gini coefficient of zero means that the region is equally accessible across all cells, while a gini index of 1 means that a region is uniquely accessible in one cell. We find that specific enhancers (with a high gini score) tend to agree in their accessibility and activity, while ubiquitously accessible enhancers (with a low gini score) do not show corresponding accessibility and activity patterns (Fig 3E, Appendix Fig S16). In addition, motifs linked to transcription factors with a restricted expression, such as Glass (posterior to the MF) and Ocelliless (head vertex), are found in the specifically accessible enhancers, while motifs linked to Grainyhead, an epithelial transcription factor, are found in the generally accessible regions. Indeed, Jacobs *et al* (2018) showed that Grh is a pioneer transcription factor which directly promotes opening of all its target regions throughout the epithelial tissue of the eye-antennal disc, while their activity is restricted to certain cell

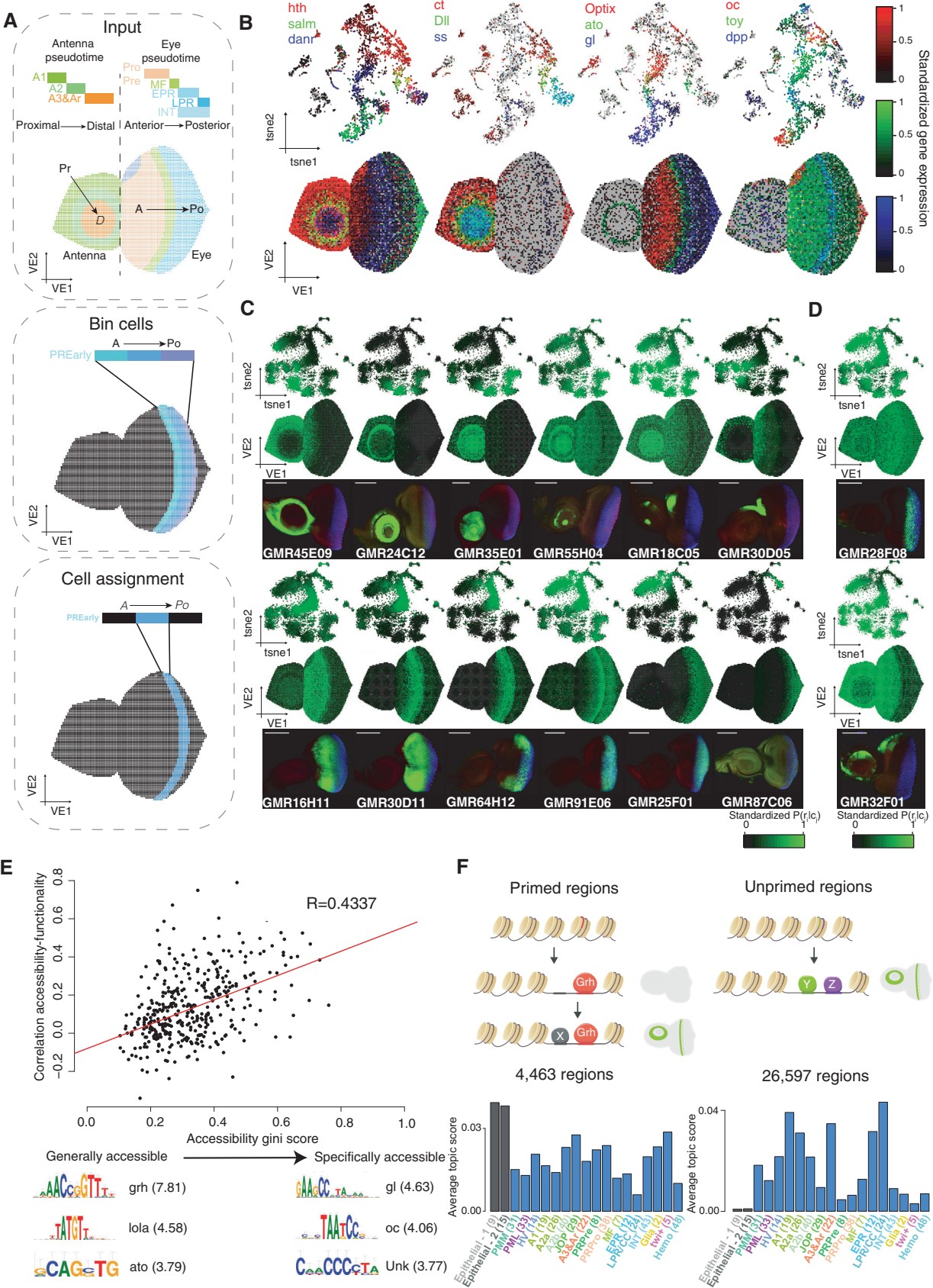

**Figure 3.**

◀

**Figure 3. Spatiotemporal mapping of single-cell RNA-seq and single-cell ATAC-seq data.**

A Computational approach for mapping single-cell RNA or single-cell ATAC-seq data into the virtual eye-antennal disc. Briefly, cells are ordered by pseudotime, corresponding to the proximal-distal axis in the antennal disc and the anterior–posterior axis in the eye disc. For each cluster, real and virtual cells are divided into the same number bins based on pseudotime and axis position, respectively. Finally, cells are mapped into the virtual cells in the matching bin.

B Gene expression correspondence between the Seurat tSNE and the virtual eye. The expression of three genes is shown per plot, using RGB encoding.

C Correspondence between region accessibility and activity for 12 Janelia-Gal4 enhancers. Top row: cisTopic cell tSNE colored by the accessibility probability of each region in each cell. Middle row: Virtual eye colored by the accessibility probability of each region in each cell. Bottom row: Confocal images showing the activity (GFP, green) of each region in eye-antennal discs. Scale bar, 100 μm.

D Discordance between region accessibility and activity for 2 Janelia-Gal4 enhancers. Top row: cisTopic cell tSNE colored by the probability of each region in each cell. Middle row: Virtual eye colored by the probability of each region in each cell. Bottom row: Confocal images showing the activity (GFP, green) of each region in eye-antennal discs. Scale bar, 100 μm.

E Relationship between the correlation between the accessibility and the activity of the regions and their distribution (as gini score). Below, representative motifs enriched in generally and specifically regions, with low (< 0.2) and high (> 0.4) gini score, respectively, are shown.

F Model describing the two classes of enhancers found. On the one hand, some enhancers (such as Grh targets) are generally accessible, but only become functional with a specific co-factor(s) binds; on the other hand, for other enhancers, accessibility is more specific and is couples with activity (based on the binding of one or more TFs). Histograms shown the average topic score for enhancers of both classes are shown.

Data information: Ar: arista. EPR: early photoreceptors. Hemo: hemocytes. HV: head vertex. INT: interommatidial cells. JOP: Johnston organ precursor. LPR/CC: late photoreceptors and cone cells. MF: morphogenetic furrow. PML: peripodial membrane lateral. PMM: peripodial membrane medial. PRPre: photoreceptor precursors. PRPro: photoreceptor progenitors.

types. Our data confirm that Grh binding results in a ubiquitous ATAC-seq signal, but not necessarily in activity. For example, among 20 Atonal target enhancers found earlier (Aerts *et al*, 2010), six are bound by Grh and are ubiquitously accessible, yet activated only in Ato-positive cells, while the other 14 enhancers are not bound by Grh and show cell type-specific accessibility and activity (Appendix Fig S17).

Thus, the scATAC-seq data corroborate a model consisting of two classes of enhancers: (i) primed enhancers, with general accessibility (e.g., by Grh binding) but specific activity based on the presence of other transcription factor/s (e.g., Ato) and (ii) unprimed regions, in which accessibility (e.g., by binding of a TF/s, as Ato) and activity are coupled (Fig 3F). Most of the enhancers of the first class belong to the general topics (with a total of 4,500 binarized regions on the representative general topics), while regions from the second class are spread across the cell type topics (with a total of 26,500 regions classified in cell type specific topics). In summary, accessibility can be used as a proxy for enhancer activity for the majority of enhancers, but there are ~ 15% of enhancers that form an exception.

## Exploiting the latent space to link enhancers to target genes

The virtual eye-antennal disc acts as a latent space in which both transcriptomic and epigenomic profiles are available in the same virtual cell. Hence, we developed a computational strategy to infer enhancer-to-gene relationships, which is also included in the ScoMAP package. Particularly, we investigated to what extent enhancers in a large space around the TSS of a gene (i.e., ±50 kb from the TSS plus introns) can predict the expression of a target gene (Fig 4A). For each gene, we calculated the following: (i) the correlation between gene expression and the accessibility probability of each candidate region across all the virtual cells and (ii) the importance of each candidate region for predicting the expression of the gene using random forest regression models, accounting for non-linear relationships. Briefly, for each gene we build a number of regression decision trees, in which the accessibility of the regions is used in the decision nodes and the gene expression is the value to predict; and the importance of each region is calculated based on

how much the variance on the gene expression is reduced based on the split (Huynh-Thu *et al*, 2010). While the random forest importance reflects the strength of the link between the region and the gene, the sign of the correlation score indicates whether the relationship is positive (> 0) or negative (< 0). Importantly, negative relationships between features tend to be noisier due to the sparse nature of the single-cell data, as not observing accessibility or gene expression may reflect drop-out events. After pruning low confidence links (see Materials and Methods), we obtained a total of 183,336 enhancer-to-gene relationships (with 95,484 and 87,229 positive and negative links, respectively).

To verify these predicted enhancer-to-gene links, we used validated associations from literature. For example, we predict *sens* expression to be exclusively regulated by one enhancer, sens-F2, in agreement with Pepple *et al* (2008) (Fig 4B, Appendix Fig S18A). In other cases, we find that gene expression is a result of combinations of enhancers. For instance, *dac* expression is mainly controlled by two redundant enhancers (3EE and 5EE), as shown by Pappu *et al* (2005). Both enhancers are accessible in the precursor cells, where *dac* is expressed (Fig 4C, Appendix Fig S18B). As a more complex example, *gl* expression is regulated by a combination of 14 enhancers, out of which three enhancers have been experimentally tested by Fritsch *et al* (2019) (Fig 4D, Appendix Fig S18C). While *gl* is expressed in all cell types posterior to the MF (both in ommatidial and interommatidial cells), some of these regions are exclusively accessible and active in interommatidial cells (i.e., subregion in enhancer 3), while others are only accessible and active in photoreceptors (i.e., enhancer 2, subregion in enhancer 3, and enhancer 4), suggesting that different enhancer combinations are involved in ommatidial cells versus interommatidial cells (Appendix Fig S19). Overall, there is a median of 22 enhancers linked to each gene (with a median of 11 positive links and 11 negative links, respectively) and only 2.4% of all genes are regulated by one enhancer (Fig 4E). These results indicate that gene expression is regulated by an intricate network of enhancer interactions. Further corroborating our predicted enhancer-to-gene associations, we find that ~ 62% of the Janelia enhancers for which accessibility and activity are coupled are positively linked (with a correlation > 0.2) to a target gene.

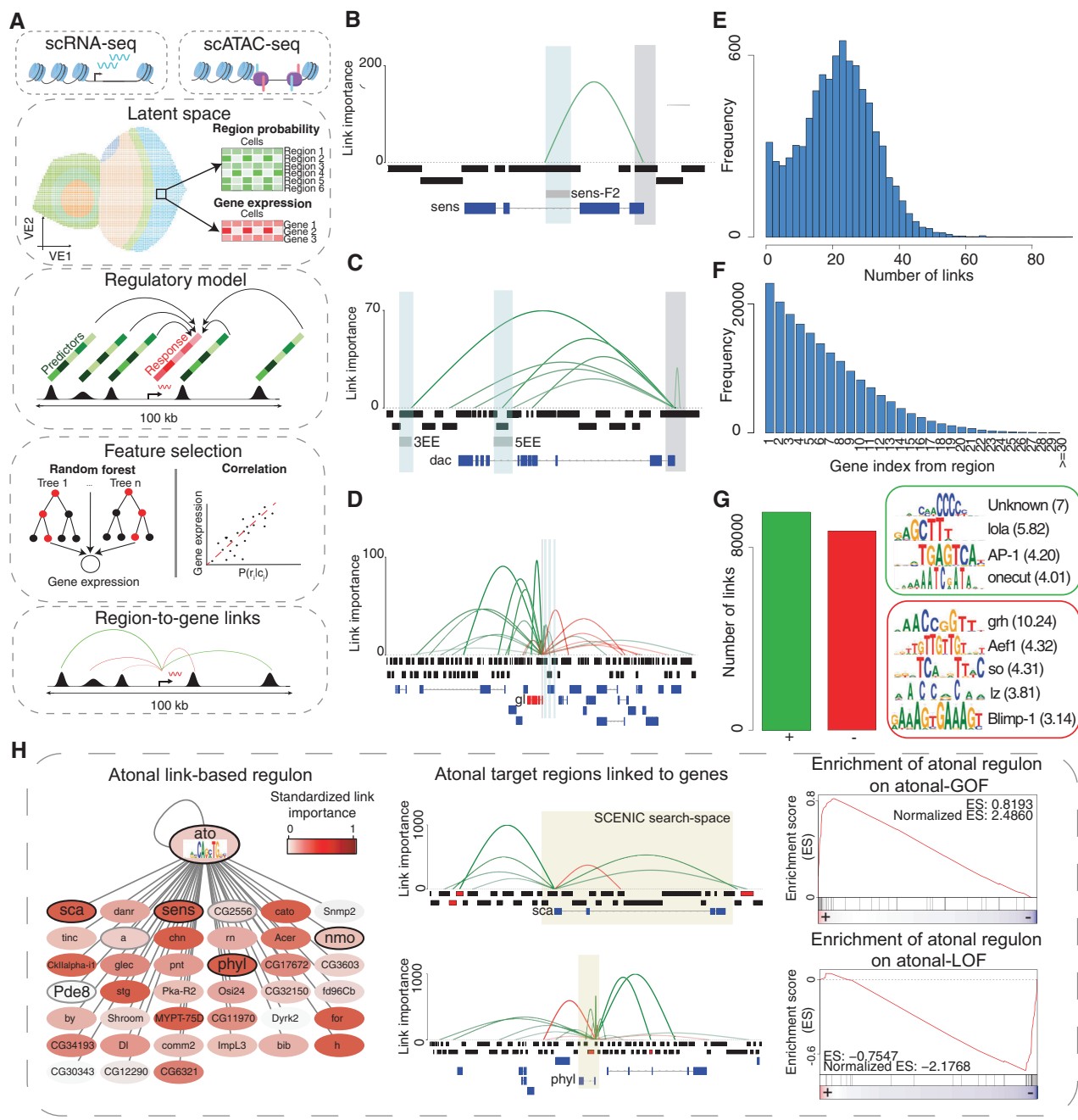

**Figure 4. Enhancer-to-target links unveil a complex multi-level regulation of gene expression.**

A   Computational approach for linking enhancer to target genes.

B   Inferred links for *senseless*. The promoter of the gene is highlighted in grey, and the validated enhancer sens-F2 is highlighted in blue.

C   Inferred links for *dacshund*. The promoter of the gene is highlighted in grey, and the validated enhancers 3EE and 5EE are highlighted in blue.

D   Inferred links for *glass*. The promoter of the gene is highlighted in grey, the validated enhancers by Fritsch *et al* (2019) are highlighted in blue, and the *glass* gene is highlighted in red.

E   Number of enhancer-to-gene links per gene.

F   Number of links with genes in the ranked position based on distance from the enhancer.

G   Number of positive and negative links, with representative enriched motifs in each category with Normalized Enrichment Score (NES).

H   Link-based regulon for Atonal built using GRNBoost co-expression modules and motif enrichment on the regions linked to each potential target gene. Left: Cytoscape view of the link-based regulons. Color scale indicates the average importance of the regions enriched in the transcription factor motif for each gene. Known targets of Atonal (Aerts *et al*, 2010) are highlighted in black and grey and with a bigger font. Middle: Examples of target genes, showing the enhancer-to-region links (top), cisTarget regions (middle), and gene annotation. cisTarget regions in which the motif for the transcription factor is enriched are shown in red. The area highlighted in yellow corresponds to the motif enrichment search space used in SCENIC (Aibar *et al*, 2017). Right: GSEA (Gene Set Enrichment Analysis) plots comparing the link-based regulons with differentially expressed genes in both gain and loss of function mutants described in Aerts *et al* (2010).

Interestingly, TF genes are regulated by slightly, but significantly, more enhancers compared with non-TF genes ($H_0$: average number of positive links for TF genes (13) ≤ Average number of positive links for non-TF genes (11); *P*-value: $2 \times 10^{-4}$; Appendix Fig S20A). This is further supported by Gene Ontology analysis with GOrilla (Genes ordered by decreasing number of links, *P*-value: $5 \times 10^{-4}$; Appendix Fig S20B and C). Indeed, it has been hypothesized that TF genes require a tighter regulation because abnormalities in their expression can cause more dramatic effects compared with defects in the expression of terminal effector genes (Barolo, 2012). In addition, we find enhancer–enhancer pairs linked to the same gene with a high correlation in accessibility (with a median of 4 enhancer–enhancer pairs with a correlation > 0.8; equivalent to 3–4 redundant enhancers), being significantly higher between enhancers linked to a TF gene compared with those linked to a non-TF gene ($H_0$: average number of enhancer–enhancer pairs with a correlation above 0.8 for TF genes (13) ≤ average number of enhancer–enhancer pairs with a correlation above 0.8 for non TF genes (7); *P*-value: 0.006; Appendix Fig S20D). In agreement, ~ 73 % of the enhancer–enhancer pairs involving Janelia enhancers also show correlation between their activity patterns (Appendix Fig S21).

The multiplicity of enhancers with the same function, known as shadow enhancers, has an evolutionary basis and has been suggested to provide robustness during development (Osterwalder *et al*, 2018). In addition, redundant enhancers can compensate when an enhancer is affected by a loss-of-function mutation or deletion (Frankel *et al*, 2010; Perry *et al*, 2010). Contrary to our expectation, shadow enhancers are not less conserved than isolated enhancers (Appendix Fig S20E) and do not show more genomic variation compared to isolated enhancers (Appendix Fig S20F).

Based on the 183K enhancer-to-gene associations, we investigated the distance between regions and their predicted genes, the genomic annotation and the motif composition of the enhancers involved in these networks. Firstly, we found that enhancers do not necessarily act on their closest gene, although the nearest gene is overall the most likely target (Fig 4F). Secondly, most regions linked to a target gene fall in non-promoter regions (75 %) (Appendix Fig S20G). Indeed, for genes that show cell type-specific expression (with adjusted *P*-value < 0.05 and average log FC > 1), the accessibility of the promoter is poorly correlated with the expression of the gene (median correlation 0.15), as promoters tend to ubiquitously accessible ($H_0$: Proportion of promoters in generally accessible topics (0.54) ≤ Proportion of promoters across all topics (0.36), *P*-value < $2.2 \times 10^{-16}$). Interestingly, enhancer accessibility can be positively (95,484, of which 13,125 are uniquely positive with a correlation > 0.1) or negatively (87,229, of which 2,927 are uniquely negative with a correlation < −0.1) correlated with target gene expression (Fig 4G). Negative correlation is suggestive of gene repression, whereby a repressor binds to the enhancer, creating an accessible region, only in the cells where the gene is not expressed.

The "activating" enhancers show enrichment for motifs of the factors Lola-T/K, AP-1 and Onecut, as well as the GGG motif that we previously found in the ommatidial enhancers. On the other hand, among the "repressive" enhancers, we identified motifs linked to So/Optix, Lz, and Blimp-1. While Blimp-1 and Lz can act as repressors in *Drosophila* (Daga *et al*, 1996; Agawa *et al*, 2007), so and Optix have been suggested to act as either activators or co-

repressors (anterior to the morphogenetic furrow) during eye development (Anderson *et al*, 2012). For instance, by looking at enhancer-to-gene links related to *hth*, a gene potentially repressed by so (Lopes & Casares, 2015), we find a repressive enhancer (chr3R:10563160-10564462) with a so binding site (based on ChIP-seq) that is also enriched in the Optix-GFP$^+$ FAC-sorted cells (Appendix Fig S22). In fact, this enhancer is specifically accessible in the cells in which both *Optix* and *so* are expressed, while *hth* is repressed in these cells. This suggests that Optix and So may cooperate to repress *hth* in the eye precursor cells via this regulatory region. In addition, within the *glass* locus we also observe a repressive enhancer with accessibility in the antenna and progenitors, losing accessibility as cells approach the morphogenetic furrow as activating *glass* enhancers become accessible (Appendix Fig S18C).

Next, we used the inferred enhancer-to-target genes in an attempt to improve the inference of a "gene activity matrix" from the scATAC-seq data (i.e., predicting gene expression from ATAC-seq peaks). Briefly, instead of aggregating the probability of all the regions around a certain space around the TSS (i.e., 5 kb upstream the TSS and introns as used above) of the gene of interest, we calculate the gene activity score by the weighted sum (weighted by importance) of the accessibility probabilities of the enhancers linked to each gene. We were able to recapitulate previously observed gene expression patterns (Fig 1C, Appendix Fig S20H), supporting the robustness of the inferred and binarized links. For instance, we found the expression gradient of *ct*, *Dll*, and *ss* from outer to inner antennal rings, and a gradient from *Optix*, to *ato*, and to *gl* driving differentiation in the eye (Bessa *et al*, 2002; Emerald *et al*, 2003).

We then exploited these enhancer-to-gene associations to create new regulons, now being able to extend the search space for motif discovery around each gene. Particularly, in comparison to the SCENIC workflow (Aibar *et al*, 2017), in which after deriving co-expression modules per TF the target genes are selected based on the enrichment of the motif/s linked to the TF in the entire sequence space around the TSS (i.e., 5 kb upstream the TSS and introns); we evaluated motif enrichment restricted to the regions that are linked to each potential target gene. Out of the 161 regulons predicted in this manner, 91 have a canonical SCENIC counterpart and have average size 2.6 times smaller than the SCENIC regulons (Appendix Fig S23A). In addition, this approach identifies new regulons that were not found with SCENIC, such as Toy and Zld, which are involved in differentiation of ommatidial cell types; Salm in ommatidial cell types; Ct in R7 and cone cells and A1; Dll in A2, A3, and arista; and Dfd in the peripodial membrane (Bessa *et al*, 2002; Emerald *et al*, 2003; Domingos *et al*, 2004; Stultz *et al*, 2012; Hamm *et al*, 2017; Appendix Fig S23B and C). We further validated the link-based regulons using differential expression rankings from Ato gain and loss-of-function mutants (versus WT), GMR-GFP$^+$ cells (versus GMR-GFP$^-$), and a loss-of-function mutant of *onecut* (versus WT). We found that the predicted genes in the Ato regulon are upregulated in the GOF mutant and downregulated in the LOF mutant; Glass predicted target genes are enriched in GMR-GFP$^+$ cells; and the predicted Onecut regulon is downregulated in the *onecut* LOF (Fig 4H; Appendix Fig S23D). In addition, we find an overlap of 24 % when comparing the predicted Glass binding sites with Glass ChIP-seq binding sites in the embryo (Davis *et al*, 2018).

In summary, we provide a new method to infer GRNs involving distal enhancers, and a resource of enhancer-to-gene relationships

that can be exploited to validate basic principles of gene regulation and infer detailed gene regulatory networks.

## Cell type-specific caQTL analysis reveals key transcription factor binding sites that impact chromatin accessibility

Having established a gene regulatory landscape at single-cell resolution, we next asked whether it can be exploited to interpret the effects of cis-regulatory variation on enhancer function. To this end, we identified chromatin accessibility quantitative trait loci (caQTLs) using a cohort of bulk eye-antennal disc ATAC-seq profiles across inbred lines (Jacobs *et al*, 2018; Fig 5A). While 21 of these samples were profiled by Jacobs *et al* (2018), we performed 29 additional bulk ATAC-seq experiments, resulting in a resource of 50 samples with highly robust ATAC-seq profiles (correlation between samples: 0.5–1; Fig 5B).

To identify caQTLs (i.e., SNPs or indels that correlate with ATAC-seq signal), we used a generalized linear model (GLM) on all the 456,893 SNPs present in the 38,179 accessible regions, finding 10,969 SNPs (2.4%) that correlate significantly with accessibility changes in the regions where they are located (adjusted *P*-val < 0.05). These ~ 10K caQTLs are found across 4,853 genomic regions (Fig 5C). Compared to the reference allele, 6,781 of these caQTLs promote chromatin closure, while the remaining 4,188 result in chromatin opening (Fig 5C). Next, we evaluated whether these caQTLs either create or break a TF motif, using a collection of more than 24,000 transcription factor motifs (Herrmann *et al*, 2012; Imrichová *et al*, 2015). Particularly, for each motif we compared the motif score between the reference enhancer sequence and the enhancer carrying the SNP, obtaining a Delta score for each caQTL and each motif. In agreement with Jacobs *et al* (2018), we found that the motif linked to Grh is significantly more associated with caQTLs than to control SNPs (adjusted *P*-val: $10^{-29}$ by Fisher's exact test) and directly explains the accessibility of 158 regions (with abs(delta) > 2, Appendix Fig S24A). However, in Jacobs *et al* we failed to detect any additional enriched motifs that are significantly affected by caQTLs.

Here, we exploited our cell type-specific topics to perform the motif enrichment analysis for each topic separately. This effectively changes the null model and aims to detect motifs that are significantly more altered in caQTLs in cell type-specific regions, compared to SNPs in cell type-specific regions. This strategy of using cell type-specific null models revealed 33 additional motifs (log(Fisher test adjusted *P*-value) > 8, in at least one topic), which explain 2,061 extra caQTLs genome wide (with abs(delta) > 2) compared with the bulk analysis.

Cell type-specific motif enrichment permits to infer in which cell types certain caQTLs are relevant. For example, caQTLs found within accessible regions anterior to the morphogenetic furrow and interommatidial cells significantly affect Optix and so binding sites (adjusted *P*-val: $10^{-2}$ by Fisher's exact test), while caQTLs in photoreceptor and cone cell regions mainly impact the GGG motif (adjusted *P*-val: $10^{-4}$ by Fisher's exact test) (Fig 5D, Appendix Fig S24B), among others, suggesting that transcription factors linked to these motifs play a role in chromatin accessibility in these specific cell types. Vice versa, when evaluating the caQTLs that affect the motifs of Grh (158), Optix/So (53), Glass (49), and the GGG motif (29), we observe that they are located in accessible regions of the matching cell type (with abs(delta) > 2; Fig 5E and F, Appendix Fig S24C).

In summary, cell type-specific signatures derived from single-cell ATAC-seq can be exploited to assess cell type-specific effects of caQTLs derived from a panel of bulk ATAC-seq profiles, providing a higher resolution and sensitivity compared with a bulk data analysis.

## A TF perturbation screen to identify GGG motif binders

While the GGG motif plays an important role in regions specifically accessible in photoreceptor neurons, the transcription factor/s that bind to it are currently unknown. In fact, this motif is enriched in regions specifically accessible in photoreceptors (Fig 2); the accessibility of regions with this motif is tightly correlated with their activity (Fig 3); these regions are related to gene activation rather than repression (Fig 4); and caQTLs affecting this motif are enriched in photoreceptor-specific enhancers (Fig 5).

To find potential transcription factors that bind to this motif, we first collected candidate TFs that are expressed in photoreceptors and that have a GGG-like motif, based on the *Drosophila* motif, or the motif of orthologous factors in other species. We also analyzed the entire modERN collection of ChIP-seq data by motif enrichment

---

**Figure 5. Cell type-specific caQTL analysis reveals key transcription factor binding sites that model the chromatin landscape during the development of the eye disc.**

A   Approach for the identification of genome-wide caQTLs using bulk ATAC-seq profiles of 50 inbred *Drosophila melanogaster* lines. Briefly, after identifying the SNPs among the lines, a generalized linear model (GLM) is used to assess whether the presence of the SNP has an effect in chromatin accessibility. Once these caQTLs are identified, we estimate the effect they have on transcription factor binding sites by comparing the motif score with the reference and alternative SNP (i.e., delta score). A positive delta score indicates that the presence of the motif is related to chromatin opening, while a negative delta score reflects that the motif cause chromatin closeness.

B   Bulk chromatin profiles of the 50 inbred lines. While 21 of these ATAC-seq experiments were performed by Jacobs *et al* (2018), we generated 29 additional profiles. Peak calling defined regions are shown in black on the top.

C   Examples of caQTLs linked to openness (left) and closeness (right) compared to the reference genome.

D   Adjusted *P*-value by Fisher's exact test comparing the proportion of caQTLs versus random SNPs affecting each motif and aggregated delta score per topic and bulk regions.

E   cisTopic cell tSNE colored by the enrichment of regions whose accessibility is affected by caQTLs that alter the highlighted binding sites.

F   Examples of caQTLs in regions that belong to different topics and affect a certain binding site. Top: Motif with delta score. Middle: Representative bulk ATAC-seq profile on lines with the reference and the alternative allele. Bottom: cisTopic cell tSNE colored by the accessibility of the region affected by the caQTL. The caQTL coordinates are, from left to right: chr3L:17392596, chr3R:14076593, chr2R:18674001 and chr2R:18674002, and chr3R:29376820.

Data information: Ar: arista. EPR: early photoreceptors. Hemo: hemocytes. HV: head vertex. INT: interommatidial cells. JOP: Johnston organ precursor. LPR/CC: late photoreceptors and cone cells. MF: morphogenetic furrow. PML: peripodial membrane lateral. PMM: peripodial membrane medial. PRPre: photoreceptor precursors. PRPro: photoreceptor progenitors.

---

and identified three TFs, namely Pros, Nerfin-1, and l(3)neo38, that have a very strong GGG motif enriched in their ChIP-seq peaks (*cisTarget* Normalized Enrichment Score (NES) of 10.40, 5.93, and 5.61, respectively). In total, we selected 14 candidate TFs, particularly Pros, Lola (isoforms L and T), Nerfin-1 (FlyORF constructs CC and HA), l(3)neo38, Sp1, Ttk (isoforms Ttk88 and Ttk69), Lz, Lov,

Psq, and Fru (alleles EY09280 and E0Y2366). Next, we overexpressed each of these 14 TFs in the posterior part of the eye disc using GMR-GAL4 as driver, and for each TF, we analyzed phenotypic changes as well as bulk ATAC-seq. As our screen is based on bulk ATAC-seq, we favored TF overexpression versus RNAi because some of these TFs are expressed in a small subset of cells, and their

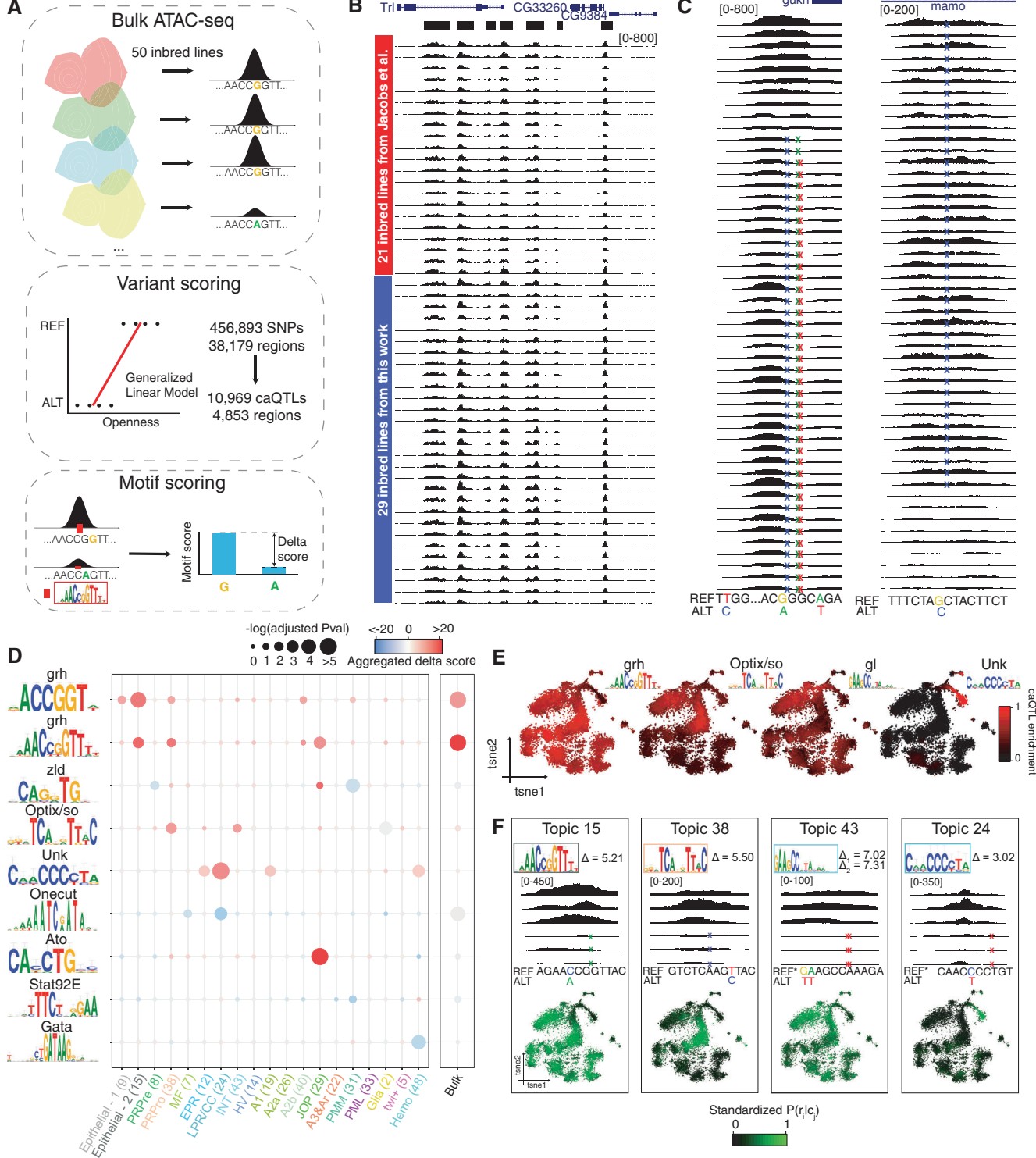

**Figure 5.**

removal would result in only very small changes in a bulk ATAC-seq signal.

We assessed whether overexpression of these TFs in the posterior part of the eye disc resulted in an adult eye phenotype, which was the case for 9 of the 14 TFs (Appendix Fig S25). Of these, overexpression of Pros and Lola-T had the most severe phenotype, resulting in lethality in the early pupa stage. Overexpression of l(3)neo38, Nerfin-1, and Sp1 caused a rough eye phenotype and provoked by defects in the development of photoreceptors (Iyer *et al*, 2016); the overexpression of Ttk69 gave rise to a small eye and loss of photoreceptors; and overexpression of Lz and Lola-L led to loss of pigment and rough eye phenotype.

To assess the changes caused in the chromatin landscape by the overexpression of these TFs and to investigate whether GGG regions are affected, we performed bulk ATAC-seq on the eye-antennal disc for each TF gain of function. We clustered all ATAC-seq data across all TFs using cisTopic (on bootstrapped data, see Materials and Methods), revealing two topics whose regions are highly enriched in the GGG motif, namely topics 4 and 18 (Fig 6A and B). Both topics represent regions that become highly accessible upon overexpression of Pros, with regions in topic 18 also weakly increasing in accessibility upon overexpression of other TFs, including Nerfin-1 and l(3)neo38 (Appendix Fig S26A and B).

Only Pros overexpression results in a strong opening of both early and late-born photoreceptor GGG enriched regions, while overexpression of other TFs has a weak effect (Fig 6C, Appendix Fig S26C and D). On the other hand, topic 4, which contains regions uniquely accessible upon Pros overexpression, is more strongly enriched in the late-born photoreceptor regions found in the scATAC-seq data compared with regions in topic 18, which contains regions that slightly increase in accessibility upon overexpression of other TFs, such as Nerfin-1 and l(3)neo38 (Appendix Fig S26E and F). These results agree with the phenotype observed in the third-instar larvae eye disc: Pros overexpression has a strong impact throughout photoreceptor development, while the effects of Nerfin-1 and l(3)neo38 are milder and largely affect the structure of early ommatidia (Appendix Fig S26G).

In the wild-type eye-antennal disc, Nerfin-1 and l(3)neo38 are expressed in early and late-born photoreceptors, while Pros expression is limited to late-born photoreceptors (Fig 6D). This suggests that Nerfin-1 and l(3)neo38 could be the early openers of the GGG enriched regions, while Pros would act in late-recruited photoreceptors. In fact, the embryonic ChIP-seq profiles of these transcription factors support their binding to the photoreceptor GGG enriched regions, especially for Pros and Nerfin-1 (Appendix Fig S26H). When comparing the GGG regions bound by these factors in the embryo, we find that 50–65% of the sites are shared by the three

transcription factors (Fig 6E). Differential motif enrichment analysis between shared and transcription factor-specific binding sites reveals that the shared sites are highly enriched for GGG motifs (adjusted $P$-value: $10^{-14}$), meaning that the three TFs can bind to regions with strong GGG motifs. On the other hand, regions specifically bound by l(3)neo38 are enriched for the canonical l(3)neo38 binding site (adjusted $P$-value: $10^{-17}$); regions uniquely bound by Pros are enriched for a GATC motif, previously reported as being associated with Prospero binding sites and linked to Lola-N (Southall *et al*, 2014) (adjusted $P$-value: $10^{-10}$); and regions uniquely bound by Nerfin-1 are enriched for the Ara/Caup/Mirr motif (adjusted $P$-value: $10^{-5}$). Indeed, both Nerfin-1 and Mirr have been reported to be involved in axon guidance (Kuzin *et al*, 2005; Karim & Moore, 2011).

In summary, given the high enrichment of the GGG motif within Pros ChIP-seq peaks in the embryo, the strong opening of GGG enriched regions upon Pros overexpression, and its expression in late photoreceptors, we propose Prospero as a key regulator of late-born photoreceptors (R7) and cone cells through the binding of the GGG motif. In addition, our data suggest that in early photoreceptors, in which Pros is not expressed, Nerfin-1 and l(3)neo38 can be weaker binders of GGG enriched regions.

## Discussion

Single-cell technologies provide unprecedented insights into the dynamics of gene regulation across all cell types within a tissue. However, these techniques require the dissociation of the tissue, resulting in the loss of spatial information. While new experimental techniques are arising to preserve spatial information while profiling single cells, these mainly target single-cell transcriptomics and methods that profile genome-wide transcription are limited in resolution (Ståhl *et al*, 2016; Burgess, 2019; Thornton *et al*, 2019). Alternatively, new computational approaches have been developed, such as novoSpaRc (Nitzan *et al*, 2019); however, *de novo* spatial relationships are only possible on one-dimensional tissues and otherwise require of a gene expression reference map (Karaiskos *et al*, 2017). In this work, we present a semi-supervised approach to map omics data into a virtual template by extracting axial information via pseudotime ordering, available as an R package called ScoMAP (https://github.com/aertslab/ScoMAP). The main limitations of this approach are that (i) it can be currently only applied to 1D or 2D tissues, (ii) it requires *a priori* information about at least one landmark between the real and the virtual cells and the direction of the axis, and (iii) it assumes symmetry around the axes, meaning that other gradients may be lost as cells are spread randomly in each

---

**Figure 6. Prospero mediates terminal photoreceptor differentiation by binding the GGG motif.**

A cisTopic topic-cell heatmap, based on a model with 21 topics. For running cisTopic, 50 single-cell profiles were bootstrapped from the 15 bulk ATAC-seq profiles of the GMR-GAL4 UAS-TF and wild-type lines included in the screen.

B Highlighted topics showing a representative topic region (top) and representative enriched motifs with their Normalized Enrichment Score (NES).

C Heatmaps showing the normalized coverage of the early photoreceptor GGG enriched regions and late GGG enriched photoreceptor regions on the selected GMR-GAL4 UAS-TF lines.

D Seurat cell tSNE colored by the expression of l(3)neo38, Nerfin-1, and Prospero.

E Venn diagram showing the overlap between the GGG enriched binding sites of Prospero, Nerfin-1, and l(3)neo38. Differentially enriched motif in each class is shown with their adjusted $P$-value.

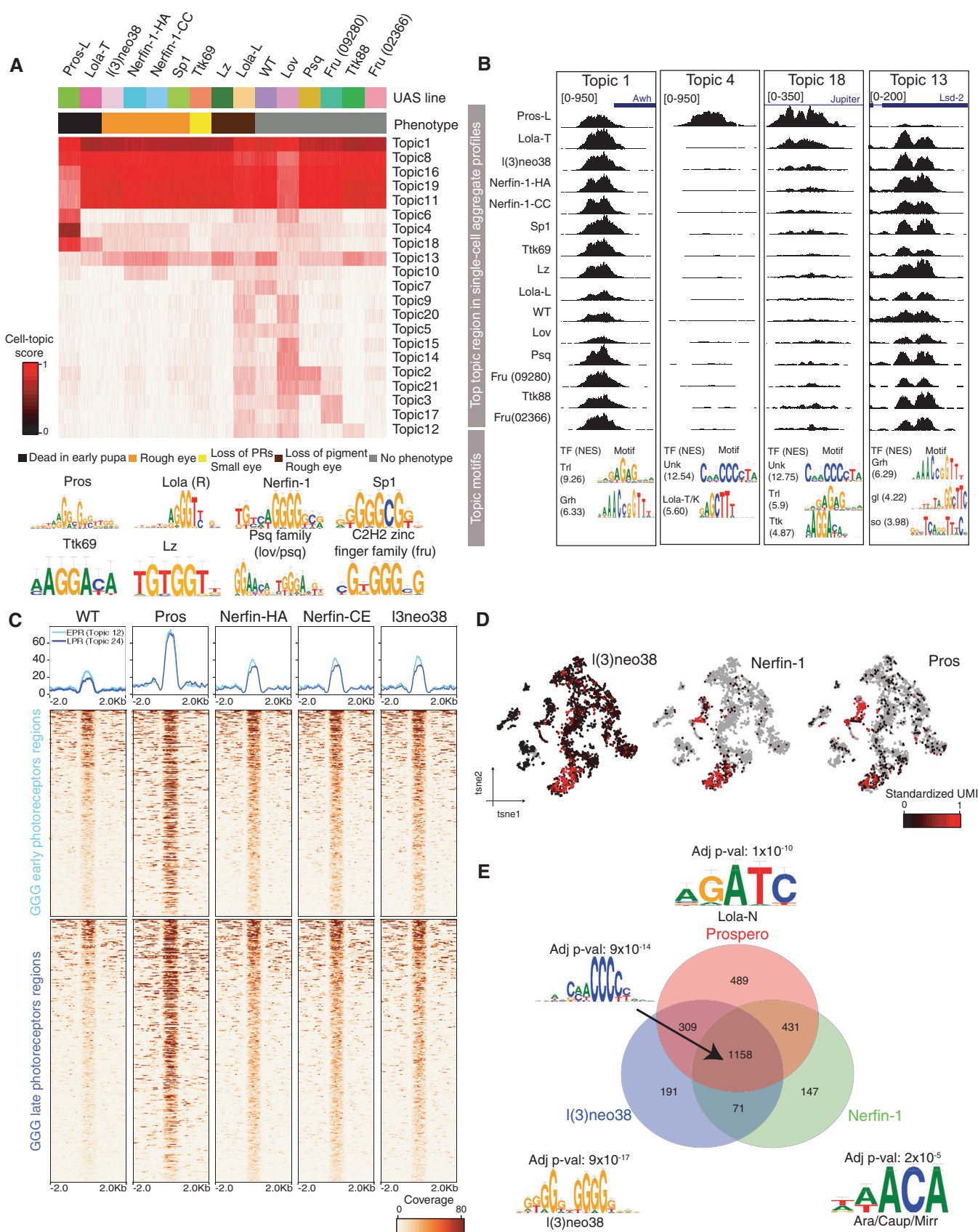

**Figure 6.**

bin. Nevertheless, the spatial gene expression atlas resulting from the mapping of scRNA-seq accurately recapitulates known gene expression patterns and allows to generate virtual gene expression profiles for any gene, at a resolution comparable with novoSpaRc (Nitzan *et al*, 2019).

Whereas spatial inference has been reported based on scRNA-seq data, in this work we generate the first spatial map of a tissue from scATAC-seq data. This accessibility atlas effectively predicts enhancer–reporter activity for more than 700 enhancers from the Janelia FlyLight Project, with ~ 85% of enhancers showing matching accessibility and activity patterns. The remaining enhancers (~ 15%) are binding sites of the epithelial pioneer transcription factor Grainyhead (Jacobs *et al*, 2018), which primes these regions in all the epithelial cells without resulting in enhancer activity. Indeed, pioneer transcription factors are able to displace nucleosomes, resulting in an ATAC-seq signal; and despite that they are necessary, their binding is not sufficient for activity (Jacobs *et al*, 2018). Thus, enhancer accessibility can be achieved either by the binding of pioneer factors or through the cooperative binding of multiple TFs. These results highlight both the power of using scATAC-seq as a proxy of enhancer activity and the need for caution when dealing with pioneer factors.

The virtual map also acts as a latent space in which scATAC-seq and scRNA-seq data are available for each virtual cell. While experimental approaches for the simultaneous profiling of epigenome and transcriptome are emerging (Cao *et al*, 2018; Chen *et al*, 2019b; Liu *et al*, 2019), these do not achieve the same throughout and sensitivity compared with the independent assays yet. Computationally, Granja *et al* (2019) have taken a similar approach, in which cells are mapped into the same latent space and for each single-cell transcriptome, the aggregate scATAC-seq profile of the closest neighbors is assigned. The resulting integrated profiles allow inferring relationships between enhancers and target genes. While Pliner *et al* (2018) have tackled this problem uniquely using scATAC-seq data, Granja *et al* (2019) used Pearson correlation between the chromatin accessibility and gene expression. In this work, we extend this approach by also using random forest models to assess non-linear relationships. Of note, these approaches are not robust to pioneer sites, whose accessibility and activity are unpaired. For example, in our approach a validated intronic enhancer of Atonal and Grainyhead in *sca* (Aerts *et al*, 2010) is missed, as the enhancer is ubiquitously accessible while only functional in the morphogenetic furrow, where the gene is expressed. Nevertheless, for the remaining 85% of the enhancers in which accessibility and activity are coupled, in this system, we have been able to reconstruct novel and validated enhancer-to-target gene links.

The predicted links between enhancers and target genes support that (i) the probability of an enhancer regulating a gene decreases exponentially with the distance and the number of non-intervening genes in between, as also reported by others (Sanyal *et al*, 2012; de Laat & Duboule, 2013; Shlyueva *et al*, 2014); and (ii) genes are regulated by several—and in some cases, redundant—enhancers, with a median of 22 enhancers linked to each gene. Indeed, Cannavò *et al* (2016) reported in the *Drosophila* embryo that ~ 64% of the mesodermal loci have redundant (or shadow) enhancers, of which ~ 60% contain more than one pair of shadow enhancers. In agreement, we find that ~ 80% of the genes are regulated by shadow enhancers (6,937 out of 8,307 genes), out of which ~ 72% are regulated by at least three shadow enhancers (4,900 out of 6,937 genes). Transcription factors are more tightly regulated, being linked with a higher number of enhancers (with an average of 13 positive links per gene) and having almost twice the number of redundant enhancers compared with non-TFs genes. As abnormalities in the expression of transcription factor genes can have more severe phenotypes compared with final effector genes, having more —and redundant—enhancers may provide evolutionary robustness. In addition, the majority of shadow enhancers are partially redundant, meaning that they can be uniquely essential on other developmental stages or tissues, or under adverse environmental conditions (Frankel *et al*, 2010; Lam *et al*, 2015; Cannavò *et al*, 2016).

Of note, almost ~ 50% of the inferred links are negatively correlated with their target genes. While polycomb-mediated repression has been shown to reduce region accessibility (Fitzgerald & Bender, 2001), other studies suggest that, although repressed enhancers are less accessible than active enhancers, they still show accessibility compared with the non-regulatory genome (Bozek *et al*, 2019). Such effect can be observed in the embryonic *eve stripe 2* enhancer, which is active (and more accessible) in the second embryonic stripe, while repressed (and less accessible) in the rest (Small *et al*, 1992). Meanwhile, in the eye-antennal disc, where it is not active nor repressed, there is no accessibility (Appendix Fig S27). Thus, accessible regions do not only correspond to primed or active enhancers, promoters, and insulators, but also to repressed enhancers.

Several works have focused on the inference of GRNs from single-cell data, mostly exploiting scRNA-seq to infer co-expression patterns between TFs and potential target genes (Chen & Mar, 2018). In an attempt to reduce the number of false-positive targets due to activating cascade effects, we introduced SCENIC (Aibar *et al*, 2017), which additionally evaluates the enrichment of binding sites for the TF around the TSS of the putative target genes. On the other hand, other studies have exploited single-cell ATAC-seq to find target enhancers with binding sites for specific TFs. For example, chromVAR (Schep *et al*, 2017) aggregates regulatory regions based on motif enrichment and then evaluates these modules on single-cell ATAC-seq data, while cisTopic (Bravo González-Blas *et al*, 2019) performs motif enrichment on sets of co-accessible enhancers inferred from scATAC-seq profiles (i.e., topics) to find common master regulators. However, none of these approaches incorporates knowledge about the TF nor target gene expression. Here, we aim for the first time to integrate all these layers—transcription factor binding sites, chromatin, and gene expression—to infer GRNs, by deriving co-expression modules between genes and transcription factors (from the scRNA-seq data) and pruning them based on the enrichment of the TF motif in the enhancers that regulate these genes (based on the enhancer-to-target gene links derived from the integration of scATAC and scRNA-seq data). Such networks *de facto* have enhancers, rather than genes, as nodes (i.e., TF-Enhancer-Gene networks).

As bulk profiles may mask true biological signal (due to the proportions of the different cell types), single-cell data have been used to deconvolute cell type-specific signals from bulk RNA-seq data (Baron *et al*, 2016), permitting to exploit large cohorts with bulk omics data, complemented with only one single-cell reference atlas. Here, we investigated the impact of genomic variation on cell type-specific enhancers. For example, we revealed the relevance of Atonal binding sites for opening Johnston's organ precursor-specific

regions and the GGG motif, previously unlinked to any transcription factor, for opening photoreceptor regions. Interestingly, Atonal has been shown to be a key transcription factor for the specification of sensory neurons (Eberl & Boekhoff-Falk, 2007) and bHLH proteins have been proposed to act as pioneer transcription factors in certain contexts (Soufi *et al*, 2015), such as the mammalian family member Ascl1 (Pataskar *et al*, 2016).

The importance of the GGG motif in neuronal enhancers was evident in most of our analyses; however, its interpretation was a challenge because the binding TFs were unknown. While yeast one-hybrid (Y1H) experiments have been previously used to reverse-engineer which transcription factors can bind a motif of interest, lowly expressed TFs may be underrepresented in the cDNA library and interactions that occur *in vivo* may be missed (such as those dependent of post-transcriptional modifications) (Southall *et al*, 2014; Fuxman Bass *et al*, 2016). Here, we have used a novel *in vivo* approach, in which we identify the changes that overexpression of potential TF candidates causes in chromatin accessibility at the bulk ATAC-seq level. Although this strategy allows to characterize the effects of TF overexpression directly on the tissue of interest, it also has limitations, such as the limited throughput of *in vivo* genetic screens (one TF per experiment, compared to dozens of TFs that can be tested by Y1H or Perturb-ATAC (Rubin *et al*, 2019) *in vitro*). This requires making a stringent selection of potential candidates

that can be further bounded by the existence of compatible tools, such as UAS-TF lines. In addition, the changes in chromatin may not be direct, but these effects can be partially ruled out using external data available, such as ChIP-seq.

We found that the neuronal precursor transcription factor Prospero acts as the strongest binder of the GGG motif, followed by Nerfin-1 and l(3)neo38. In fact, overexpression of each of them, but especially Prospero, results in the opening of GGG regions; and all three transcription factors, especially Pros and Nerfin-1, can bind to the GGG motif. Based on the expression of these transcription factors, we hypothesize that Nerfin-1—and l(3)neo38 —are the early binders of the GGG motif, while Pros can bind to these regions in the late-born photoreceptors, where it is expressed. In fact, Pros and Nerfin-1 have been reported to share direct targets during CNS differentiation (Froldi *et al*, 2015) and have been found to be key regulators during the photoreceptor and retinal differentiation in other organisms, such as zebrafish, chicken, and mammals (Dyer *et al*, 2003; Edqvist *et al*, 2006; Nelson *et al*, 2007; Forbes-Osborne *et al*, 2013).

In summary, we provide a comprehensive and user-friendly single-cell resource of the *Drosophila*'s eye-antennal disc. We envision that our computational strategies and enhancer resource will be of value not only to the *Drosophila* community, but also to the field of single-cell regulatory genomics in general.

# Materials and Methods

## Reagents and Tools table

| Reagent/Resource | Reference or source | Identifier or catalog number |
| --- | --- | --- |
| **Experimental models** | | |
| Hybrid of DGRP-551, DGRP-360, DGRP-907 and DGRP-913 | In-house | – |
| Transgenic line with P{Optix-EGFP.2-3} | Graeme Mardon | – |
| w;F2-B/CyO (ato-targeted intronic sens enhancer) | Graeme Mardon | – |
| w[1118]; P{y[+t7.7] w[+mC]=GMR45E09-GAL4}attP2 | Bloomington Drosophila Stock Center | 49564 |
| w[1118]; P{y[+t7.7] w[+mC]=GMR24C12-GAL4}attP2 | Bloomington Drosophila Stock Center | 49076 |
| w[1118]; P{y[+t7.7] w[+mC]=GMR35E01-GAL4}attP2 | Bloomington Drosophila Stock Center | 45619 |
| w[1118]; P{y[+t7.7] w[+mC]=GMR55H04-GAL4}attP2 | Bloomington Drosophila Stock Center | 39134 |
| w[1118]; P{y[+t7.7] w[+mC]=GMR18C05-GAL4}attP2 | Bloomington Drosophila Stock Center | 47330 |
| w[1118]; P{y[+t7.7] w[+mC]=GMR30D05-GAL4}attP2 | Bloomington Drosophila Stock Center | 49534 |
| w[1118]; P{y[+t7.7] w[+mC]=GMR16H11-GAL4}attP2 | Bloomington Drosophila Stock Center | 47473 |
| w[1118]; P{y[+t7.7] w[+mC]=GMR30D11-GAL4}attP2 | Bloomington Drosophila Stock Center | 48098 |
| w[1118]; P{y[+t7.7] w[+mC]=GMR91E06-GAL4}attP2/TM3, Sb[1] | Bloomington Drosophila Stock Center | 47166 |
| w[1118]; P{y[+t7.7] w[+mC]=GMR25F01-GAL4}attP2 | Bloomington Drosophila Stock Center | 49127 |
| w[1118]; P{y[+t7.7] w[+mC]=GMR87C06-GAL4}attP2 | Bloomington Drosophila Stock Center | 40482 |
| w[1118]; P{y[+t7.7] w[+mC]=GMR28F08-GAL4}attP2 | Bloomington Drosophila Stock Center | 45172 |
| w[1118]; P{y[+t7.7] w[+mC]=GMR32F01-GAL4}attP2 | Bloomington Drosophila Stock Center | 49359 |
| w[1118]; P{w[+mC]=UAS-GFP.nls}8 | Bloomington Drosophila Stock Center | 4776 |
| DGRP-379 | Bloomington Drosophila Stock Center | 25189 |
| DGRP-391 | Bloomington Drosophila Stock Center | 25191 |
| DGRP-437 | Bloomington Drosophila Stock Center | 25194 |

**Reagents and Tools table**   (continued)

| Reagent/Resource | Reference or source | Identifier or catalog number |
|---|---|---|
| DGRP-555 | Bloomington Drosophila Stock Center | 25198 |
| DGRP-712 | Bloomington Drosophila Stock Center | 25201 |
| DGRP-59 | Bloomington Drosophila Stock Center | 28129 |
| DGRP-91 | Bloomington Drosophila Stock Center | 28136 |
| DGRP-101 | Bloomington Drosophila Stock Center | 28138 |
| DGRP-109 | Bloomington Drosophila Stock Center | 28140 |
| DGRP-129 | Bloomington Drosophila Stock Center | 28141 |
| DGRP-350 | Bloomington Drosophila Stock Center | 28176 |
| DGRP-352 | Bloomington Drosophila Stock Center | 28177 |
| DGRP-374 | Bloomington Drosophila Stock Center | 28185 |
| DGRP-382 | Bloomington Drosophila Stock Center | 28189 |
| DGRP-392 | Bloomington Drosophila Stock Center | 28194 |
| DGRP-441 | Bloomington Drosophila Stock Center | 28198 |
| DGRP-584 | Bloomington Drosophila Stock Center | 28212 |
| DGRP-776 | Bloomington Drosophila Stock Center | 28229 |
| DGRP-796 | Bloomington Drosophila Stock Center | 28233 |
| DGRP-802 | Bloomington Drosophila Stock Center | 28235 |
| DGRP-808 | Bloomington Drosophila Stock Center | 28238 |
| DGRP-810 | Bloomington Drosophila Stock Center | 28239 |
| DGRP-853 | Bloomington Drosophila Stock Center | 28250 |
| DGRP-57 | Bloomington Drosophila Stock Center | 29652 |
| DGRP-439 | Bloomington Drosophila Stock Center | 29658 |
| DGRP-32 | Bloomington Drosophila Stock Center | 55015 |
| DGRP-319 | Bloomington Drosophila Stock Center | 55018 |
| DGRP-566 | Bloomington Drosophila Stock Center | 55028 |
| DGRP-627 | Bloomington Drosophila Stock Center | 55030 |
| wt; GMR-Gal4 | LNG flystock | – |
| P{w[+mC]=UAS-pros.L}L3a, w[*] | Bloomington Drosophila Stock Center | 32244 |
| y[1] w[*]; P{w[+mC]=UAS-lola.4.7}3 | Bloomington Drosophila Stock Center | 28828 |
| M{UAS-l(3)neo38.ORF.3xHA.GW}ZH-86Fb | FlyORF | F000093 |
| M{UAS-nerfin-1.ORF.3xHA.GW}ZH-86Fb | FlyORF | F000461 |
| M{UAS-nerfin-1.ORF-CC}ZH-21F | FlyORF | F004559 |
| M{UAS-Sp1.ORF.3xHA.GW}ZH-86Fb | FlyORF | F001783 |
| w[*]; P{w[+mC]=UAS-ttk.p69}2 | Bloomington Drosophila Stock Center | 7361 |
| w[*]; P{w[+mC]=UAS-lz.B}3 | Bloomington Drosophila Stock Center | 33836 |
| w[*]; Pin[1]/CyO; P{w[+mC]=UAS-lola.L}3a | Bloomington Drosophila Stock Center | 28829 |
| w[1118]; P{w[+mC]=EP}lov[EP1162]/CyO | Bloomington Drosophila Stock Center | 16994 |
| M{UAS-psq.ORF-VN}ZH-86Fb | Bloomington Drosophila Stock Center | F004846 |
| y[1] w[*]; P{w[+mC] y[+mDint2]=EPgy2}fru[EY09280]/TM3, Sb[1] | Bloomington Drosophila Stock Center | 17551 |
| P{w[+mC]=UAS-ttk.p88}1, w[*] | Bloomington Drosophila Stock Center | 7360 |
| y[1] w[67c23]; P{w[+mC] y[+mDint2]=EPgy2}fru[EY02366] | Bloomington Drosophila Stock Center | 15564 |
| **Antibodies** | | |
| Rabbit anti-GFP | Invitrogen | A11122 |
| Rat anti-elav | DSHB | 7E8A10 |
| Mouse anti-pros | DSHB | MR1A |

                                                        

**Reagents and Tools table**  (continued)

| Reagent/Resource | Reference or source | Identifier or catalog number |
| --- | --- | --- |
| Goat anti-rabbit Alexa Fluor 647 | Invitrogen | A32733 |
| Donkey anti-rat Alexa Fluor 488 | Invitrogen | A21208 |
| Donkey anti-mouse Alexa Fluor 555 | Invitrogen | A31570 |
| **Oligonucleotides and sequence-based reagents** | | |
| ATAC-seq primers | Buenrostro *et al* (2015) | Fwd:-'AATGATACGGCGACCACCGAGATCTACACTCGTCGGCAGCGTCAGATGTG'//Rev:-'CAAGCAGAAGACGGCATACGAGATXXXXXXGTCTCGTGGGCTCGGAGATGT' |
| **Chemicals, enzymes and other reagents** | | |
| Dispase | Sigma-Aldrich | D4818 |
| Collagenase | Invitrogen | 17100_017 |
| Acridine Orange/Propidium Iodide Stain | Logos Bio | F23001 |
| Chromium Single Cell 3′ Library & Gel Bead Kit v2 | 10x Genomics | PN-120237 |
| Chromium Single Cell A Chip Kit | 10x Genomics | PN-120236 |
| Chromium i7 Multiplex Kit | 10x Genomics | PN-120262 |
| Nuclei Isolation Kit: Nuclei EZ Prep | Sigma | NUC101-1KT |
| Nextera DNA Library prep kit | Illumina | FC-121-1030 |
| RNAqueous-Micro Total RNA Isolation Kit | ThermoFisher | AM1931 |
| Dynabeads mRNA Purification Kit | ThermoFisher | 61006 |
| TruSeq Stranded mRNA Library Prep | Ilumina | 20020595 |
| MEGAscript T7 Transcription Kit | ThermoFisher | AM1333 |
| Chromium Single Cell ATAC Library & Gel Bead Kit | 10x Genomics | PN-1000110 |
| Chromium Chip E Single Cell ATAC Kit | 10x Genomics | PN-1000086 |
| Chromium i7 Multiplex Kit N | 10x Genomics | PN-1000084 |
| MiniElute PCR Purification Kit | Qiagen | 28004 |
| AMPure XP beads | Beckman Coulter | A63880 |
| SPRIselect Reagent kit | Beckman Coulter | B23318 |
| Vectashield | Vector Laboratories | H-1000-10 |
| **Software** | | |
| CellRanger (v2.0.2) | 10x Genomics | |
| pySCENIC (V0.9.1) | https://github.com/aertslab/pySCENIC | |
| Cytoscape | https://cytoscape.org | |
| Seurat (v2.3.4/v3.0.1) | CRAN | |
| ScopeLoomR (v0.4.0) | https://github.com/aertslab/SCopeLoomR | |
| Ea-utils (v1.12) | https://expressionanalysis.github.io/ea-utils/ | |
| FastQC (v0.1) | https://www.bioinformatics.babraham.ac.uk/projects/fastqc/ | |
| Bowtie2 (v2.2.5) | http://bowtie-bio.sourceforge.net/bowtie2/index.shtml | |
| Samtools (v1.2) | http://www.htslib.org | |
| Picard | https://broadinstitute.github.io/picard/ | |
| cisTopic (v0.2.2) | https://github.com/aertslab/cisTopic | |
| MACS (v2.1.2.1) | https://github.com/taoliu/MACS | |
| Subread (v2.0.0) | http://subread.sourceforge.net | |
| Deseq2 (v1.18.1) | Bioconductor | |
| CellRanger ATAC (v1.0.0) | 10x Genomics | |

**Reagents and Tools table**   (continued)

| Reagent/Resource | Reference or source | Identifier or catalog number |
|---|---|---|
| RcisTarget (v1.5.0) | Bioconductor | |
| Harmony (v1.0) | https://github.com/immunogenomics/harmony | |
| AUCell (v1.5.2) | Bioconductor | |
| DoubletFinder (v2.0.1) | https://github.com/chris-mcginnis-ucsf/DoubletFinder | |
| jpeg (v0.1-8) | CRAN | |
| ScoMAP (v0.1.0) | https://github.com/aertslab/ScoMAP | |
| destiny (v3.0.0) | Bioconductor | |
| lawstat (v3.2) | CRAN | |
| GENIE3 (v1.8.0) | Bioconductor | |
| fitdistrplus (v1.0-11) | CRAN | |
| Binarize (v1.3) | CRAN | |
| Gviz (v1.22.3) | Bioconductor | |
| GRNBoost (v0.1.5) | https://github.com/aertslab/GRNBoost | |
| VCFtools (v0.1.14) | https://vcftools.github.io/index.html | |
| Bedtools (v2.28.0) | https://bedtools.readthedocs.io/en/latest/ | |
| Deeptools (v3.3.1) | https://deeptools.readthedocs.io/en/develop/ | |
| Fiji | https://imagej.net/Fiji | |
| MAST (v1.4.1) | Bioconductor | |
| **Other** | | |
| 10um pluriStrainer | Imtec Diagnostics | 435001050 |
| Dounce homogeneizer | Sigma-Aldrich | D8938 |
| LUNA-FL Dual Fluorescence Cell Counter | Logos Bio | L20001 |
| C1000 Touch Thermal Cycler | Bio-rad | 1851148 |
| NextSeq 500 | Illumina | |
| Olympus FV1200 confocal microscope | Olympus | |

## Methods and Protocols

### Fly husbandry and genotypes

A detailed description of the lines used in this work is provided in Dataset EV2. A wild-type line, hybrid of DGRP-551, DGRP-360, DGRP-907, and DGRP-913, was used on the single-cell RNA-seq and single-cell ATAC-seq experiments with 10x Genomics. For cell sorting (followed by bulk and single-cell ATAC-seq with Fluidigm C1), we used a sens-F2B-GFP transgenic line (Pepple et al, 2008). For measuring enhancer activity in a subset of lines from the Janelia FlyLight Project, we selected the stocks (with Bloomington number): 49564, 49076, 45619, 39134, 47330, 49534, 47473, 48098, 47166, 49127, 40482, 45172, and 49359 and crossed them with a UAS-eGFP line (Bloomington number: 4776). For the analysis of caQTLs, we performed bulk ATAC-seq on 29 lines from the Drosophila Genetics Reference Panel (with Bloomington number): 25189, 25191, 25194, 25198, 25201, 28129, 28136, 28138, 28140, 28141, 28176, 28177, 28185, 28189, 28194, 28198, 28212, 28229, 28233, 28235, 28238, 28239, 28250, 29652, 29658, 55015, 55018, 55028, and 55030. For the genetic screen, we used the following lines from the Bloomington Drosophila Stock Center: 32244, 28828, 7361, 33836, 28829, 16994, 17551, 7360, and 15564 and the following from FlyORF: F000093, F000461, F004559, F001783, and F004846. These lines were crossed with a GMR-GAL4 line. All flies were raised and crossed at 25°C on a yeast-based medium.

### Dissociation of eye-antennal discs into single cells

Wandering third-instar larvae were collected, and a total of ~ 30 eye-antennal discs were dissected and transferred into a tube containing 200 µl of ice-cold PBS. The sample was centrifuged at 800 $g$ for 5 min, and after removing the supernatant, 50 µl of dispase (3 mg/ml; Sigma-Aldrich_D4818-2mg) and 70 µl of collagenase (100 mg/ml; Invitrogen_17100-017) were added. The tissue was dissociated during 45–60 min at 25°C at 50 rpm, pipetting up and down every 15 min to disrupt clumps of cells. Cells were washed with 1 ml of ice-cold PBS and resuspended in 400 µl of PBS 0.04% BSA. The cells were passed through a 10-µm pluriStrainer (ImTec Diagnostics_435001050), and cell viability and concentration were assessed by the LUNA-FL Dual Fluorescence Cell Counter.

### Single-cell RNA-seq (10x Genomics)

Single-cell libraries were generated using the GemCode Single-Cell instruments and the Single Cell 3′ Library & Gel Bead Kit v2 and ChIP Kit from 10x Genomics, following the protocol provided by the manufacturer. Briefly, the eye-antennal disc cells were suspended in

PBS 0.04% BSA. About 8,700 cells were added in each reaction with a targeted cell recovery of 5,000 cells. Following the generation of nanoliter-scale Gel bead-in-EMulsions (GEMs), GEMs were reverse transcribed in a C1000 Touch Thermal Cycler (Bio-Rad) programmed at 53°C for 45 min, 85°C for 5 min, and hold at 4°C. After reverse transcription, single-cell droplets were broken and the single-strand cDNA was isolated and cleaned with Cleanup Mix containing DynaBeads (Thermo Fisher Scientific). cDNA was then amplified with a C1000 Touch Thermal Cycler programmed at 98°C for 3 min, 12 cycles of (98°C for 15 s, 67°C for 20 s, 72°C for 1 min), 72°C for 1 min, and held at 4°C twice. Subsequently, the amplified cDNA was fragmented, end-repaired, A-tailed and index adaptor ligated, with the SPRIselect Reagent Kit (Beckman Coulter) with cleanup in between steps. The post-ligation product was amplified with a C1000 Touch Thermal Cycler programmed at 98°C for 45 s, 14 cycles of (98°C for 20 s, 54°C for 30 s, 72°C for 20 s), 72°C for 1 min, and hold at 4°C. The sequencing-ready library was cleaned up with SPRIselect beads.

### Dissociation of eye-antennal discs into single nuclei

Wandering third-instar larvae were collected, and a total of ~ 30 eye-antennal discs were dissected and transferred into a tube containing 200 μl of ice-cold PBS. The sample was centrifuged at 800 *g* for 5 min, and after removing the supernatant, resuspended in 500 μl of nuclei lysis buffer (10 mM Tris–HCl (pH 7.4), 10 mM NaCl, 3 mM MgCl₂, 0.1% Tween-20, 0.1% Nonidet P-40, 0.01% Digitonin, 1% BSA, and water), and transferred to a Dounce homogenizer (Sigma-Aldrich D8938_2ml). After incubating the sample for 5 min on ice, 25 strokes were applied with the loose pestle. The sample was incubated for 10 min on ice and after applying 25 strokes with the tight pestle, transferred to a 2-ml tube. The homogenizer and the pestle were rinsed with wash buffer (10 mM Tris–HCl (pH 7.4), 10 mM NaCl, 3 mM MgCl₂, 0.1% Tween-20, 1% BSA, and water), and the solution was also transferred to the 2-ml tube. The sample was washed once with wash buffer and resuspended on 50 μl of 1× diluted nuclei buffer (10x Genomics). The nuclei were passed through a 10-μm pluriStrainer (ImTec Diagnostics_435001050), and cell viability and nuclei concentration were assessed by the LUNA-FL Dual Fluorescence Cell Counter.

### Single-cell ATAC-seq (10x Genomics)

Single-cell libraries were generated using the GemCode Single-Cell instruments and the Single Cell ATAC Library & Gel Bead Kit and ChIP Kit from 10x Genomics, following the protocol provided by the manufacturer. Briefly, the eye-antennal disc nuclei were suspended in 1× diluted nuclei buffer (10x Genomics). About 8,700 nuclei were added in each reaction with a targeted nuclei recovery of 5,000 nuclei. The samples were incubated at 37°C for 1 h with 10 μl of transposition mix (per reaction, 7 μl ATAC Buffer, and 3 μl ATAC Enzyme (10x Genomics)). Following the generation of nanoliter-scale Gel bead-in-EMulsions (GEMs), GEMs were reverse transcribed in a C1000 Touch Thermal Cycler (Bio-Rad) programmed at 72°C for 5 min, 98°C for 30 s, 12 cycles of 98°C for 10 s, 59°C for 30 s, and 72°C for 1 min, and held at 15°C. After reverse transcription, single-cell droplets were broken and the single-strand cDNA was isolated and cleaned with Cleanup Mix containing DynaBeads (Thermo Fisher Scientific). cDNA was then

amplified with a C1000 Touch Thermal Cycler programmed at 98°C for 3 min, 12 cycles of (98°C for 15 s, 67°C for 20 s, 72°C for 1 min), 72°C for 1 min, and held at 4°C twice. Subsequently, the amplified cDNA was fragmented, end-repaired, A-tailed and index adaptor ligated, with the SPRIselect Reagent Kit (Beckman Coulter) with cleanup in between steps. The post-ligation product was amplified with a C1000 Touch Thermal Cycler programmed at 98°C for 45 s, 14 cycles of (98°C for 20 s, 54°C for 30 s, 72°C for 20 s), 72°C for 1 min, and hold at 4°C. The sequencing-ready library was cleaned up with SPRIselect beads.

### Cell sorting

Wandering third-instar larvae were collected, and a total of 200 eye-antennal discs were dissected in ice-cold PBS and placed in SF900 medium. For dissociation, the tissue was placed in 400 μl of trypsin in 0.05% EDTA. The eye-antennal discs were then incubated at 37°C for 1 h with agitation, being mixed every 20 min with a pipette. After dissociation, cells were centrifuged at 800 *g* for 5 min at 4°C and washed with PBS. Finally, the cells were resuspended in 400 μl of PBS, filtered using a 40-μm cell strainer, and stained with propidium iodide (PI; final concentration 1 μg/ml) to exclude dead cells. The cells were sorted on a BD Aria I, selecting against the presence of PI and for the presence of GFP.

As many cells as possible were sorted into a microcentrifuge tube, pelleted by centrifugation at 800 *g* for 5 min at 4°C and resuspended at a concentration of 1,000 cells/μl. Single-cell ATAC-seq was performed as previously described (Buenrostro *et al*, 2015; Bravo González-Blas *et al*, 2019), using 5- to 10-μm Open App integrated fluidic circuits (IFCs) on the Fluidigm C1 and with no cell washing step. Briefly, cells were loaded (using a 40:60 ratio of RGT:cells) on a primed Open App IFC (5–10 μm, the protocol for ATAC-seq from the C1 Script Hub was used). After cell loading, the plate was visually checked under a microscope and the number of cells in each of the capture chambers was noted. Next, the sample preparation was performed on the Fluidigm C1, during which the cells underwent lysis and ATAC-seq fragments were prepared. In a 96-well plate, the harvested libraries were amplified in a 25 μl PCR. The PCR products were pooled and purified on a single MinElute PCR purification column for a final library volume of 15 μl. Quality checks were performed using the Bioanalyzer high sensitivity chips. Fragments under 150 bp were removed by bead-cleanup using AMPure XP beads (1.2× bead ratio) (Beckman Coulter).

### ATAC-seq

For the DGRP panel lines, we used the ATAC-seq protocol for eye-antennal discs as previously described (Buenrostro *et al*, 2015; Davie *et al*, 2015). Briefly, ~ 10 eye-antennal discs were dissected and lysed in 50 μl ice-cold ATAC lysis buffer (10 mM Tris–HCl, pH 7.4, 10 mM NaCl, 3 mM MgCl₂, 0.1% IGEPAL CA-630). Lysed discs were then centrifuged at 800 *g* for 10 min at 4°C, and the supernatant was discarded. The rest of the ATAC-seq protocol was performed as described previously (Buenrostro *et al*, 2015; Davie *et al*, 2015), using the following primers: Fwd:- "AATGATACGGCGACCACCGAG ATCTACACTCGTCGGCAGCGTCAGATGTG" and Rev:- "CAAGCAG AAGACGGCATACGAGATGTCTCGTGGGCTCGGAGATGT" (where X indicates barcode nucleotides). The final library was purified using a Qiagen MinElute kit (Qiagen), and Ampure XP beads (Ampure)

(1:1.2 ratio) were used to remove remaining adapters and was checked on an Agilent Bioanalyzer 2000 for assessing the average fragment size. Resulting successful libraries were sequenced with 75 bp, single-end reads on the Illumina NextSeq 500 platform. Single-end sequencing was chosen for this part of the study because we were not interested in the fragment contents (i.e., how many nucleosomes are placed between two insertion sites), rather just the profile of insertion sites, and also made the comparison with the previously existing data [i.e., the bulk ATAC-seq DGRP panel and Optix-GFP from Jacobs *et al* (2018)] easier.

For the genetic screen samples, we used the Omni-ATAC-seq protocol, as previously described (Bravo González-Blas *et al*, 2019). Briefly, ~ 10 eye-antennal discs were dissected and lysed using 50 µl of cold ATAC-Resuspension Buffer (RSB) (see Corces *et al* (2017) for composition) containing 0.1% NP-40, 0.1% Tween-20, and 0.01% digitonin, by pipetting up and down three times and incubating the cells for 3 min on ice. The lysis was washed out by adding 1 ml of cold ATAC-RSB containing 0.1% Tween-20 and inverting the tube three times. Nuclei were pelleted at 500 RCF for 10 min at 4°C, the supernatant was carefully removed, and nuclei were resuspended in 50 µl of transposition mixture (25 µl 2× TD buffer (see Corces *et al* (2017) for composition), 2.5 µl transposase (100 nM), 16.5 µl DPBS, 0.5 µl 1% digitonin, 0.5 µl 10% Tween-20, 5 µl $H_2O$) by pipetting six times up and down, followed by 30 min of incubation at 37°C at 1,000 RPM mixing rate. After MinElute clean-up and elution in 21 µl elution buffer, the transposed fragments were pre-amplified with NextEra primers by mixing 20 µl of transposed sample, 2.5 µl of both forward and reverse primers (25 µM), and 25 µl of 2× NEBNext Master Mix (program: 72°C for 5 min, 98°C for 30 s and five cycles of [98°C for 10 s, 63°C for 30 s, 72°C for 1 min] and hold at 4°C). To determine the required number of additional PCR cycles, a qPCR was performed (see Buenrostro *et al* (2013) for the determination of the number of cycles to be added). The final amplification was done with the additional number of cycles, samples were cleaned-up by MinElute, and libraries were prepped using the KAPA Library Quantification Kit as previously described (Bravo González-Blas *et al*, 2019). Samples were sequenced on an Illumina NextSeq 500 High Output chip, with 50 bp single-end reads.

### Immunohistochemistry

Imaginal eye-antennal discs from third-instar larvae were dissected and fixed in 4% formaldehyde at room temperature for 30 min. Next, they were washed in 1× PBT (PBS + 0.3% Triton X-100) during 15 min for three times and blocked in 3% BSA for 1 h at room temperature. To test enhancers, tissues were incubated with a primary antibody mixture (rabbit anti-GFP (Invitrogen) 1:1,000; rat anti-Elav (DSHB, 7E8A10) 1:50; and mouse anti-pros (DSHB) 1:200) at 4°C overnight. The samples were then washed three times with 1× PBT for 15 min at room temperature, followed by 2 h of incubation with secondary antibody mixture (Goat Anti-Rabbit—Alexa Fluor® 647; donkey anti-rat Alexa Fluor® 488; and donkey anti-mouse Alexa Fluor® 555) (Invitrogen/Life Technologies) at room temperature in the dark. The samples were washed again three times as mentioned above before mounting the eye-antennal discs on slide with VECTASHIELD (Vector Laboratories). For imaging, an Olympus FV1200 confocal microscope was used (20× dry). Fiji (Schindelin *et al*, 2012) (ImageJ v2.0.0-rc-69/1.52p) was used to merge and process the images.

### Analysis of single-cell RNA-seq data

The 10x eye-antennal disc samples were processed (alignment, barcode assignment, and UMI counting) with the Cell Ranger (version 2.0.2) count pipeline, using the *cellranger aggr* command with *–normalize = mapped,* and building the reference index upon the 3rd 2017 FlyBase release (*Drosophila. melanogaster* r6.16) (Gramates *et al*, 2017). Importantly, using CellRanger's default parameters, 483, 1,149, and 1,899 cells were selected based on their coverage (Appendix Fig S4A–C). Lowly expressed genes detected in less than 11 cells (0.3% of the cells) and with less 32 UMI counts across the data set (3 counts in 0.3% of the cells) were filtered, resulting in a data set with 8,744 genes and 3,531 cells that was analyzed using Seurat (v2.3.4). No cells were filtered based on the number of mitochondrial reads, as all cells had less than 5% (with a median of 1.69%) mitochondrial reads. Briefly, data were log-normalized with a scale factor of $10^4$ and latent variables, defined as the number of UMIs, were regressed out. For further downstream analysis, the most variable genes (1,495) were selected using *FindVariableGenes ()* with default parameters. Next, we used PCA to reduce the dimensionality of the original matrix, selecting the first 102 PCs based on a cross-validation step. These 102 PCs were used as input for the shared nearest neighbor (SNN) graph method implemented in Seurat, with a resolution of 1.2, resulting in 17 cell clusters. Differentially expressed genes for each cluster were estimated with the function *FindAllMarkers(),* using a Wilcoxon rank sum test with a logFC threshold of 0.25. tSNE and UMAP were performed with default parameters, using the first 102 PCs. In addition, DoubletFinder (McGinnis *et al*, 2019) (v2.0.1) was run using the first 102 PCs, with an estimated pK value of 0.04. Assuming a doublet formation rate of 7.5%, 246 high-confidence doublets were found and were removed for posterior analyses. For the semi-supervised clustering of photoreceptor subclasses, singlet cells in the early photoreceptors and late photoreceptors and cone cells were selected and Seurat (v2.3.4) was run as previously explained using marker genes for each photoreceptor subclass and cone cells as listed in FlyBase, comprising a total of 86 unique genes, using the first 7 PCs based on a cross-validation step. PySCENIC (Aibar *et al*, 2017) (v0.9.1) was run with default parameters, using motif and ENCODE ChIP-seq-based databases [as in i-cisTarget (Imrichová *et al*, 2015)], resulting in 175 regulons (159 motif-based regulons). Regulon Specificity Scores (RSS) were calculated as described by Suo *et al* (2018), and the Atonal regulon was used as input for GSEA analysis using as rankings the genes ordered by log fold change values calculated by GEO2R for eye-antennal disc RNA-seq profiles of a gain-of-function Atonal mutant and a loss-of-function Atonal mutant (versus WT) (Aerts *et al*, 2010). A representative gene regulatory network with regulons enriched in the morphogenetic furrow was built using Cytoscape.

Seurat (v3.0.1) was also used for transferring cluster labels between the eye disc data set from Ariss *et al* (2018) and this data (and vice versa). Brain cells from our data set were not included in the analysis, resulting in a data set with 3,232 cells and 8,744 genes, and the eye disc data set from Ariss *et al* was filtered to keep cells with more than 1,000 UMI counts and 500 genes expressed, resulting in a data set with 5,630 cells and 7,801 genes. Label transferring was performed with default parameters and PCA as dimensionality reduction method, using *vst* as selection method and 2,000 features for finding the variable features and the first 30 PCs for finding anchors and transfer the

data. Antennal cell types were not transferred between our data set and Ariss *et al* eye disc data set. Loom files with the results of these analyses were created using SCopeLoomR (Davie *et al*, 2015) (v0.4.0) and are available at http://scope.aertslab.org/#/Bravo_et_al_EyeAntennalDisc, and the processed data can be visualized at http://genome.ucsc.edu/s/cbravo/Bravo_et_al_EyeAntennalDisc.

### Analysis of FAC-sorted ATAC-seq data

ATAC-seq reads were first cleaned for adapters using fastq-m*cf.* (ea-utils v1.12) and a list of sequencing primers. Cleaned reads (FastQC v0.1) were then mapped to the 3$^{rd}$ 2017 FlyBase release (*D. melanogaster* r6.16) genome using Bowtie2 (v2.2.5) with default parameters, and sorted bam files were produced using SAMtools (v1.2). Single-cell profiles were aggregated using *samtools merge*. Normalized bigwigs were generated using the Kent software (UCSC).

The single-cell data were deduplicated using *picard MarkDuplicates*. Aggregation plots were produced using in-house scripts available at: https://github.com/aertslab/ATAC-seq-analysis, and cells were filtered manually based on the aggregation plot profiles, resulting in 74 and 72 Optix-GFP$^+$ and sens-GFP$^+$ single-cell ATAC-seq profiles (out of 96 and 384 sequenced cells, respectively). Downstream analysis was done using cisTopic (v0.2.2) (Bravo González-Blas *et al*, 2019).

On the bulk samples, peaks were called on mapped reads using MACS2 (v2.1.2.1) with the following additional options: *–nomodel –call-summits –nolambda*. Peaks in the independent samples were merged, and fragments per peak (and ctx region) in each sample were counted using *featureCounts* (Subread v2.0.0). Deseq2 (v1.18.1) was used to obtained differentially accessible peaks between positive and negative cells (with logFC > |1| and *P*-value < 0.05).

### Analysis of ChIP-seq data

ChIP-seq reads were first cleaned for adapters using fastq-m*cf.* (ea-utils v1.12) and a list of sequencing primers. Cleaned reads (FastQC v0.1) were then mapped to the 3$^{rd}$ 2017 FlyBase release (*D. melanogaster* r6.16) genome using Bowtie2 (v2.2.5) with default parameters, and sorted bam files were produced using SAMtools (v1.2). Single-cell profiles were aggregated using *samtools merge*. Normalized bigwigs were generated using the Kent software (UCSC). Peaks were called on mapped reads using MACS2 (v2.1.2.1) with the following options: *-g dm –nomodel –bdg -t Samples -c Control.*

### Analysis of single-cell ATAC-seq data

The 10x eye-antennal disc samples were processed (alignment and barcode assignment) with a customized version of the Cell Ranger ATAC (version 1.0.0) pipeline, in which the parameter PEAK_MERGE_DISTANCE was set to 50 (instead of 500) and the parameter PEAK_ODDS_RATIO was set to 4 (instead of 1/5), and the remaining parameters were used as default. Importantly, CellRanger ATAC identified 9,833 and 5,554 high-quality cells (Appendix Fig S4E and F). In addition, the reference index was built upon the 3$^{rd}$ 2017 FlyBase release (*D. melanogaster* r6.16) (Gramates *et al*, 2017). Sex was assigned to each cell based on the percentage of reads mapped to the X chromosome, as shown by Cusanovich *et al* (2018).

Downstream analysis was performed with cisTopic (Bravo González-Blas *et al*, 2019) (v0.2.2). Briefly, fragments within defined regulatory regions (such as ctx regions) were counted, resulting in a matrix with 129,553 regulatory regions and 15,387 cells, after filtering out a total of 379 cells based on the number of fragments within bulk peaks (> 20%) and the total number of fragments (between 100 and 10,000). Topic modeling was performed using 2, 10, 20, 30 to 50 (1 by 1), 60, 70, 80, 90 and 100 topics, and 500 iterations, out of which 250 were used as burn-in. Based on the highest log-likelihood, the model with 49 topics was selected. The cell-topic tSNE representation was obtained by using tSNE on the normalized topic-cell matrix (by Z-Score), without using the PCA reduction and with a perplexity of 100. Cell clustering was performed on the normalized cell-topic matrix (by Z-Score) using the shared nearest neighbor (SNN) graph method implemented in Seurat (v2.3.4), with a resolution of 1.2, resulting in 22 cell clusters. For identifying topics potentially related to batch effects (mainly experimental run and sex), we binarized the cell-topic distributions and used a proportion test comparing the proportion of cells corresponding to each experimental run or sex versus their proportion in the entire population. Two topics are significantly related to the experimental run: topic 46 with run 1 (Bonferroni-adjusted *P*-value for topic 46: $10^{-29}$) and topic 18 with run 2 (Bonferroni-adjusted *P*-value for topic 18: $10^{-217}$); and a topic was found to be related to the female sex (Bonferroni-adjusted *P*-value for topic 4: $10^{-21}$).

On the other hand, region-topic distributions were binarized with a probability threshold of 0.985. The region-topic tSNE was performed with similar parameters as before, using a perplexity of 200. The annotation of regions was done with default parameters. RcisTarget (Aibar *et al*, 2017) (v1.5.0) and i-cisTarget (Herrmann *et al*, 2012; Imrichová *et al*, 2015) were run to assess motif enrichment on the binarized topics, using a ROC threshold of 0.01, a maximum rank of 5,000, and the version 8 motif database, containing more than 20,000 motifs. The probability of each region in each cell (region-cell) was calculated using the *predictiveDistribution()* function, in which the topic-cell and the region-topic matrices are multiplied. For the enrichment of epigenomic signatures, region sets were mapped to the regions in the data set with a minimum overlap of 40% and the enrichment of the signatures in the cells was estimated using a maximum AUC rank of 12,956 (10% of the total number of regions) and cell-region rankings based on the region-cell probability matrix, while the enrichment of signatures in topics was estimated using a maximum AUC rank of 3,887 (3% of the total number of regions) and the region-topic distributions as rankings. Additionally, we projected the FAC-sorted single-cell profiles (Optix-GFP$^+$ and sens-GFP$^+$) with at least 70% of the fragments within regulatory regions into the existing topic space. Briefly, the topic-cell distributions of the new cells were estimated by multiplying the binary count matrix (cell-regions) by the region-topic distributions of the existing models. The estimated topic-cell contributions were merged with the topic-cell distributions of the original cells and normalized (by Z-Score), and batch effects were corrected with Harmony (v1.0) (Korsunsky *et al*, 2019).

Gene activity scores were estimated by aggregating the region probabilities of the regions surrounding the TSS of each gene (5 kb upstream and introns), as used for cisTarget enhancer-to-gene associations (Imrichová *et al*, 2015), and probabilities were multiplied by $10^6$ and rounded before creating the loom file. These gene activity-based matrix was used to assess the enrichment of the regulons derived from the analysis with pySCENIC (v0.9.1) in the single-cell RNA-seq data, using AUCell (Aibar *et al*, 2017)

(v1.5.2) with default parameters, with a maximum AUC rank of 439 (5% of the total number of genes). In addition, we also used DoubletFinder (v2.0.1) on this matrix, using the first 102 PCs, with an estimated pK of 0.27. Assuming a doublet formation rate of 20%, we find 13,848 high-confidence singlets. Finally, we performed label transferring between the scRNA-seq and the scATAC-seq (gene activity-based) data sets (and vice versa) with Seurat (v3.0.1). Label transferring was performed with default parameters and CCA as dimensionality reduction method, using *vst* as selection method and 3,000 features for finding the variable features and the first 20 dimensions for finding anchors and transfer the data. Loom files with the results of these analyses were created using SCopeLoomR (Davie *et al*, 2015) (v0.4.0) and are available at http://scope.aertslab.org/#/Bravo_et_al_EyeAntenna lDisc, and the processed data can be visualized at http://genome. ucsc.edu/s/cbravo/Bravo_et_al_EyeAntennalDisc.

### Projection of single-cell omics data into a virtual latent space
The eye-antennal disc representation (Figs 1A and 2A) was used to generate the virtual eye template coordinates. Importantly, for representing non-spatially restricted groups (i.e., twi$^+$ cells, hemocytes, glia, periopodial membrane groups) and clarify cell types posterior to the morphogenetic furrow (i.e., early photoreceptors, late photoreceptors and cone cells, and interommatidial cells; for both scRNA-seq and scATAC-seq) or in the antenna and anterior to the morphogenetic furrow (i.e., antennal rings A2a and A2b and precursors and progenitors, respectively, on the scATAC-seq analysis), circles were added to the representations. The template was reduced to a size of 100 × 100 pixels and was split into one image per cell type (in red color). Each image was read using the jpeg (v0.1-8) R package, and the background (in white color) was removed using k-means clustering on the RGB pixel values. Since interommatidial cells and photoreceptors are mixed posterior to the morphogenetic furrow, we intercalated photoreceptors and interommatidial cells in the early and late compartments posterior to the morphogenetic furrow. The resulting template coordinates were annotated per cell type, resulting in 5,058 cells on the eye-antennal disc representation, and a total of 5,379 and 5,526 cells for the scRNA-seq and scATAC-seq maps considering the non-spatially mapped cell types and detailed groups. Importantly, we also generated a 50 × 50 template (1,265 cells) and a 200 × 200 template (20,333 cells) finding our results were similar despite the size of the eye (Appendix Fig S13).

For mapping the scRNA-seq and the scATAC-seq data, antennal and eye disc cell types were ordered by pseudotime in each data set using the *DPT()* function from the destiny (Angerer *et al*, 2016) (v3.0.0) R package, using Seurat PCs and topic contributions of the singlet cells, respectively, as input for estimating the diffusion components. The pseudotime order represents the distal-proximal axis for the antennal cells, and the anterior–posterior axis in the eye cell types. Each cell type was divided into 10 bins based on their pseudotime order. Similarly in the virtual eye-antennal disc template, for each spatially located cell type in eye we calculated the distance to a reference vertical line located in the morphogenetic furrow (i.e., the distance is calculated on the X axis between the landmark point on the same Y coordinate); and for each spatially located group in the antenna, we calculated the distance of each virtual cell to a reference point in the center of the arista (i.e., the

length of the X-Y vector from the cell and the reference). Each cell type was then divided into 10 bins based on their distance to the reference landmark. For each cell type, we assigned a real profile from the matching bin to each virtual cell randomly (e.g., the cells in the first bin of a pseudotime ordered cell type are assigned to the virtual cells in the first bin of that cell type based on the distance to the landmark in the virtual eye). Progenitors and precursors and antennal rings A2a and A2b in the scATAC-seq mapping were assigned together to the anterior to the morphogenetic furrow and antennal ring A2 compartments based on pseudotime. For non-spatially located cell types and detailed groups, cells were sampled randomly without binning. If there are more real cells than virtual ones, random sampling is done without repetition; if there are more virtual cells than real ones, real profiles are assigned more than once. The scRNA-seq (i.e., gene expression) and scATAC-seq (i.e., region-cell probabilities) of the virtual cells are those of their matching real cell. This approach is included in the ScoMAP R package, with detailed tutorials, at https://github.com/aertslab/ScoMAP. Loom files with the results of these analyses were created using SCopeLoomR (Davie *et al*, 2015) (v0.4.0) and are available at http://scope.aertslab.org/#/Bravo_et_al_EyeAntennalDisc.

### Comparison of accessibility and activity profiles in the virtual latent space
The enhancer activity GFP signal was mapped into the virtual eye representation using a customized script which leverages MATLAB's Image Toolbox for landmark-based image registration (available at https://github.com/aertslab/Bravo_et_al_EyeAntenna lDisc/). Briefly, signals in the antenna and the eye were mapped independently, using projective and polynomial transformations, respectively, and manually selecting 6–8 landmarks per image. The GFP channel from the transformed images was read into R using the jpeg package (v0.1-8) and overlapped with the virtual eye template coordinates; if GFP signal was detected on a cell a value of 1 was given, if not, a value of 0 was assigned. After removing images with low or unclear signals, with signal out of the disc proper (e.g., remaining of the periopodial membrane or glial cells), with unsuccessful mapping, and duplicates, we obtained a matrix recapitulating the activity 390 enhancers, with each enhancer being active in a median of 106 virtual cells. Since Janelia enhancers are quite broad (i.e., 1–5 kb) and may include more than one cisTarget region, the accessibility probability of each Janelia enhancer was calculated by aggregating the region-cell probabilities of the regions falling within it. For comparing accessibility and activity in these regions, we calculated the Spearman correlation between the accessibility probabilities and the activity patterns, and the accessibility gini scores using the *gini.index()* function from the R package lawstat (v3.2). Motif enrichment was performed in the generally accessible regions (with gini index < 0.2) and specific regions (with gini index > 0.4) using i-cisTarget (Imrichová *et al*, 2015). For scoring the Atonal enhancers validated by Aerts *et al*, PWMs were scored in the enhancer sequences using Cluster-Buster (Frith *et al*, 2003) and visualized with TOUCAN (Aerts *et al*, 2005).

### Linkage of enhancers to target genes
For each gene, we identified as potential regulatory regions those included in a genomic space of ±50 kb around the TSS of the gene, including introns, resulting in a median of 54 potential

regulatory regions per gene. For genes with more than one TSS, we selected one TSS position randomly. We then combined two approaches to establish relationships between enhancer and target genes, (i) a linear strategy by calculating the Pearson correlation between the enhancer probabilities and gene expression in the virtual cells and (ii) a non-linear strategy based on random forest models, using the enhancer probabilities as predictors, the gene expression as response, and the GENIE3 R package (Aibar *et al*, 2017) (v1.8.0) to build each—gene specific—model, with 1,000 trees and default parameters. Importantly, we used all virtual cell profiles except for those representing detailed subgroups, covering 5,253 virtual cells. Correlation-based relationships were filtered to keep those below −0.181 and above 0.194, corresponding to the 1$^{st}$ and 99$^{th}$ percentiles of the normal distribution fitted to all the correlations derived with the fitdistrplus R package (v1.0-11). Random forest-derived relationships, based on the importance given to each region in each model, were filtered to keep the top relationships for each gene by binarizing the region importances per gene using BASC binarization as implemented in the Binarize R package (v1.3). We classified links as positive if they were positively correlated with their target genes ($> 0$) and negative if they were negatively correlated with their target genes ($< 0$). This resulted in a total of 183,336 enhancer-to-gene relationships, with a median of 22 links per gene. The Gviz R package (v1.22.3) was used to make figures representing links, and link tracks (with scores and sign of the relationship) are available at http://genome.ucsc.edu/s/cbravo/Bravo_et_al_EyeAntennalDisc. This approach is included in the ScoMAP R package, with detailed tutorials, at https://github.com/aertslab/ScoMAP.

For estimating GO terms related to the genes with the most and the least links, we used GOrilla (Eden *et al*, 2009) and visualized the results with REVIGO (Supek *et al*, 2011). The list of transcription factors for comparing features in TF and non-TF genes was obtained from the RcisTarget *Drosophila* database (Aibar *et al*, 2017). For estimating the number of redundant enhancers, we considered that two enhancers linked to the same gene were correlated if the Pearson correlation between their region-cell probabilities was above 0.8. For estimating the correlation in activity between enhancer–enhancer pairs, we evaluated the 63 combinations for which the activity of both enhancers was mapped into the virtual eye (with correlation $> 0.1$). For estimating region conservation, we used the phastCons27way file for dm6 available at UCSC. Gene activity scores were calculated by aggregating the region-cell probabilities of the regions linked to each gene, weighted by their signed (positive or negative effect) random forest importance. For integrating these links in the pySCENIC pipeline, we use the modules derived from GRNBoost (Moerman *et al*, 2019) (Arboreto v0.1.5) and performed the motif enrichment step (with RcisTarget (Aibar *et al*, 2017) (v1.5.0), using a ROC threshold of 0.01 and a maximum AUC rank of 5,000) on the regions linked to each gene in the module (using the region-based cisTarget databases) instead of around the TSS of the gene (using the gene-based cisTarget databases implemented in the original workflow). Genes linked to regions in which motifs linked to the transcription factor in each module were enriched (NES $> 3$) were kept as part of the regulon. The regulons were evaluated on the cells using AUCell (Aibar *et al*, 2017) (v1.5.2). For validating the link-based regulons, we used as input for GSEA the regulons and the genes ordered by decreasing

logFC for (i) Atonal gain-of-function and loss-of-function mutants (versus WT, using GEO2R), (ii) GMR$^+$ versus GMR$^-$ populations, and (iii) onecutx562 (loss-of-function mutant) versus WT, as provided by Potier *et al* (2014). Loom files with the results of these analyses were created using SCopeLoomR (Davie *et al*, 2015) (v0.4.0) and are available at http://scope.aertslab.org/#/Bravo_et_al_EyeAntennalDisc.

### caQTL analysis
#### Data preprocessing
Adapter sequences were trimmed from the raw reads using *fastq-mcf* (ea-utils v1.1.2, with default parameters and using a list containing the common Illumina adapters), and the quality of the cleaned reads was checked with FastQC (v0.1). All experiments were mapped using Bowtie2 (v2.2.5) to their personalized version on 3$^{rd}$ 2017 FlyBase release (*D. melanogaster* r6.16) genome.

Briefly, called variants in this genome assembly were retrieved from ftp://ftp.hgsc.bcm.edu/DGRP/freeze2_Feb_2013/liftover_data_for_D.mel6.0_from_William_Gilks_Oct_2015/, and for each of the 50 DGRP lines, we adapted the consensus genome (r6.16) using *seqtk mutfa* (seqtk (v1.0)), each time including their SNPs (previously called from whole genome sequencing). After the first mapping round, additional SNPs were called on the ATAC reads using SAMtools (v1.2), with the command *samtools mpileup -B –f r6.16.fasta DGRP_lineX.bam | varscan.sh mpileup2snp –output-vcf 1*. Newly called homozygous SNPs (several thousands per line) were added to the existing vcf files using VCFtools (v0.1.14). The genomes were again updated to obtain a final personalized genome for every DGRP line, strongly reducing mapping errors and increasing the sensitivity of subsequent analyses. Cleaned reads were mapped onto the final genomes using Bowtie2 (v2.2.5) again, and SAMtools (v1.2) was used for sorting and indexing.

Peaks were called on the mapped reads using MACS2 (v2.1.2.1), with the command *macs2 callpeak -g dm –nomodel–keep-dup all –call-summits*. The narrow peak files (bed format) for all the DGRP lines were merged into a single file that contained a total of 39,879 regions accessible in at least one DGRP line. After filtering out chrU, chrUextra, chrHet, and chrM regions and removing regions enriched in repeats ($> 25\%$ of the sequence) using bedtools (v2.28.0) with the command *intersectBed -v -f 0.25*, we obtained 38,179 accessible regions across this DGRP panel. For every ATAC-seq sample, we counted the number of reads falling into each accessible region using *featureCounts* (Subread v2.0.0). Normalized bigwig files were generated using the Kent software from UCSC.

#### Determination of caQTLs
Next, 209 regions with a low coverage for every DGRP line were removed (coverage of the region below 0.2 pb for every DGRP lines), ending up with 37,990 accessible regions. For each region, we extracted the normalized ATAC-seq reads for these 50 DGRP lines and linked each region to the annotated and additionally called SNPs for these lines. A total of 676,916 SNPs were assigned to their encompassing region using bedtools.2.26.0 *intersectBed* on the extended vcf file.

In this way, we obtained for each region the normalized reads for each of the 50 lines as one vector and all SNPs called inside this region as a binary matrix for the 50 lines (present = 1, absent = 0, unknown = NA). We searched for correlating region-SNP vectors

using the generalized linear model function in R. The *P*-values were adjusted using the Benjamini–Hochberg procedure in R. We identified 10,969 highly correlating SNP-region pairs referred to as caQTLs (Chromatin Accessibility Quantitative Trait Loci; adjusted *P*-value < 0.05).

### Delta motif scores

To single out motifs that correlate significantly with the open chromatin changes, a Delta motif score was calculated for every of the 24,454 unique motifs in our collection. The sequence for each of the 4,853 variable regions, that contained at least one caQTL, was extracted using *bedtools getfasta* (Bedtools v2.28.0). Next, we mutated these sequences with their encompassing caQTLs according to their effect on the open chromatin using *seqtk mutfa* (seqtk (v1.0)). For each of the 4,853 regions, we obtained two sequences, one for the accessible chromatin and one for the less accessible/closed chromatin. We scored every time both sequences with the 24,454 motifs using Cluster-Buster (Frith *et al*, 2003), with the options *–m 0 –c 0*, and retained for every motif the highest CRM score for each sequence. By subtracting the CRM score of the less accessible/closed region from the encompassing accessible region, we obtained a delta motif score for that region.

### Motif significance

For the general significance, we summed all delta scores from the 4,853 regions to obtain a cumulative delta score for each motif. We calculated a Delta motif score, following the same procedure, on 20K random SNPs that were present in an accessible region but had no effect on chromatin accessibility (GLM FDR > 0.95). We then calculated for each motif whether it was significantly more affected (|Delta score| > 3) by caQTLs compared with the non-correlating SNPs, using a Fisher's exact test.

Out of the 10K caQTLs, 6,682 caQTLs fall within a topic (60.9%), having in average 362 caQTLs per binarized topic. For the cell type-specific analysis, we summed all delta scores from all the regions containing at least one caQTL per topic to obtain a cumulative delta score for each motif. We calculated a Delta motif score, following the same procedure, on the 40K random SNPs that were present in an accessible region in a topic but had no effect on chromatin accessibility (GLM FDR > 0.95). We then calculated for each motif whether it was significantly more affected (|Delta score| > 3) by caQTLs compared with the non-correlating SNPs, using the Fishers exact test.

### Genetic screen data analysis

ATAC-seq reads were first cleaned for adapters using fastq-m*cf* (ea-utils v1.12) and a list of sequencing primers. Cleaned reads (FastQC v0.1) were then mapped to the 3$^{rd}$ 2017 FlyBase release (*D. melanogaster* r6.16) genome using Bowtie2 (v2.2.5) with default parameters, with the single-end option (to compare with the WT sample, which was single-end sequenced). Sorted bam files were produced using SAMtools (v1.2). Normalized bigwigs were generated using the Kent software (UCSC). Peaks were called on mapped reads using MACS2 (v2.1.2.1) with the following options: *-g dm –nomodel –bdg -t Sample/Control -c Sample/Control* (depending on whether we want to determine upregulated or downregulated peaks). ChIP-seq bam files were downloaded from ENCODE, and normalized bigwigs were also generated using the Kent software (UCSC). Peaks were called on mapped reads using MACS2

(v2.1.2.1) with the following options: *-g dm –nomodel –bdg -t Samples -c Control.* Normalized bigwigs are available at http://genome.ucsc.edu/s/cbravo/Bravo_et_al_EyeAntennalDisc.

For each sample, 50 single cells were simulated by bootstrapping 20,000 mapped reads (per cell) from the bulk bam files, resulting in a data set with 750 simulated single cells. Downstream analyses were performed with cisTopic (Bravo González-Blas *et al*, 2019). Briefly, we determined the number of ctx regions in which at least one read is mapped, and topic modeling was run using default parameters, with models including 2, 10 to 60 (one by one), 70, 80, 90, and 100 topics, using a total of 500 iterations, out of which 250 were used as burn-in. Based on the highest log-likelihood, we selected a model with 21 topics. Motif enrichment analysis was performed using RcisTarget and i-cisTarget, using a ROC threshold of 0.01 and maximum AUC rank of 5,000 (Imrichová *et al*, 2015; Aibar *et al*, 2017). The enrichment of epigenomic signatures in cells was performed using default parameters, using a maximum AUC rank of 12,320 (10% of the total number of ctx regions), while the enrichment of epigenomic regions within topics was done with default parameters. Coverage heatmaps were done using deepTools (v3.3.1). For identifying differentially enriched motifs between groups of regions, we first scored the ctx regions in the groups of interest with the 24,454 PWMs available in the cisTarget motif collection using Cluster-Buster (Frith *et al*, 2003), with the options– *m 0 –c.* Using this matrix, with ctx regions as columns, motifs as rows, and the value of the best cis-regulatory module (CRM) as value, we performed a likelihood ratio test between the region groups of interest, as implemented in MAST (v1.4.1). *P*-values were adjusted using the FDR method.

### Publicly available data used in this work

Eye disc Drop-seq data were obtained from GEO, with GEO accession number GSE115476, while dimensionality reduction coordinates and cell labels were retrieved from the supplementary data from Ariss *et al* (2018). Raw data from Optix-GFP$^+$ single-cell and bulk ATAC-seq, Grh ChIP-seq, and 21 bulk ATAC-seq profiles from were retrieved from GEO, with GEO accession number GSE102441. Raw Sine Oculis ChIP-seq data were retrieved from GEO, with GEO accession number GSE52943. Atonal gain-of-function and loss-of-function data were retrieved from GEO, with GEO accession number GSE16713. Differential expressed genes between GMR$^+$ FAC sorted cells and GMR$^-$ FAC sorted cells and onecutx562 versus WT were retrieved from the supplementary materials from Potier *et al* (2014). Glass, Prospero, Nerfin-1, and l(3)neo38 ChIP-seq profiles were retrieved from ENCODE, with the following experiment IDs, respectively: ENCSR472URU, ENCSR682YQM, ENCSR335NNR, and ENCSR643EOU. ATAC-seq profiles on different embryonic domains were obtained from GEO, with GEO accession number GSE118240.

### Resource description

#### SCope

Ariss—WT 11416 cells

- EAD_Ariss_WT_Seurat_SCENIC: Loom file containing dimensionality reductions (as shown by Ariss *et al*, based on analysis with Seurat, and based on pySCENIC regulons), gene expression, and regulon enrichment from pySCENIC (with regulons derived from pySCENIC in this data set) from the Drop-seq eye disc data set

from Ariss *et al* (2018). This data set contains 11,416 cells, 7,801 genes, and 140 regulons. The labels given by Ariss *et al* ("Ariss labels") and the labels transferred with Seurat from the 10x scRNA-seq data ("10x labels") are given as metadata.

- EAD_Ariss_WT_Seurat_SCENIC_regulonsfrom10x: Loom file containing dimensionality reductions (as shown by Ariss *et al*, based on analysis with Seurat, and based on pySCENIC regulons), gene expression, and regulon enrichment from pySCENIC (with motif-based regulons derived from pySCENIC in our 10x data set) from the Drop-seq eye disc data set from Ariss *et al* (2018). This data set contains 11,416 cells, 7,801 genes, and 159 regulons. The labels given by Ariss *et al* ("Ariss labels") and the labels transferred with Seurat from the 10x scRNA-seq data ("10x labels") are given as metadata.

scATAC-seq—15387 cells
- Gene
  - EAD_scATAC_AggSumPredictiveDistribution_Gene_Regulons: Loom file containing the cisTopic cell-Topic tSNE coordinates, gene activity scores based on the aggregation of region probabilities around the TSS (5 kb plus introns, multiplied by $10^6$), and regulon enrichment (on the gene activity score matrix, using the regulons derived from the analysis with pySCENIC in the 10x scRNA-seq data set). This data set contains 15,387 cells, 16,892 genes, and 175 regulons. The labels transferred from the 10x scRNA-seq data ("RNA labels") are given as metadata.
  - EAD_scATAC_AggSumPredictiveDistribution_Gene_Topics: Loom file containing the cisTopic cell-Topic tSNE coordinates, gene activity scores based on the aggregation of region probabilities around the TSS (5 kb plus introns, multiplied by $10^6$), and topic enrichment. This data set contains 15,387 cells, 16,892 genes, and 49 topics. The labels transferred from the 10x scRNA-seq data ("RNA labels") are given as metadata.
  - EAD_scATAC_AggSignedImportancePredictiveDistribution_Gene_Regulonss: Loom file containing the cisTopic cell-Topic tSNE coordinates, gene activity scores based on the aggregation of region probabilities based on the enhancer-to-gene links (multiplied by $10^8$), and regulon enrichment (on the gene activity score matrix, using the regulons derived from the analysis with pySCENIC in the 10x scRNA-seq data set). This data set contains 15,387 cells, 8,347 genes, and 175 regulons. The clusters derived from SNN clustering with Seurat on the topic-cell matrix ("Seurat_res_1.2") are given as metadata.
  - EAD_scATAC_AggSignedImportancePredictiveDistribution_Gene_Topics: Loom file containing the cisTopic cell-Topic tSNE coordinates, gene activity scores based on the aggregation of region probabilities based on the enhancer-to-gene links (multiplied by $10^8$), and topic enrichment. This data set contains 15,387 cells, 8,347 genes, and 49 topics. The clusters derived from SNN clustering with Seurat on the topic-cell matrix ("Seurat_res_1.2") are given as metadata.
- Janelia
  - EAD_scATAC_AggSumPredictiveDistribution_JaneliaRegions_Topics: Loom file containing the cisTopic cell-Topic tSNE coordinates, Janelia region probabilities based on the aggregation of the probabilities of the ctx regions that overlap with the Janelia enhancer (multiplied by $10^6$), and

topic enrichment. This data set contains 15,387 cells, 740 Janelia regions, and 49 topics. The labels transferred from the 10x scRNA-seq data ("RNA labels") and the clusters derived from SNN clustering with Seurat on the topic-cell matrix ("Seurat_res_1.2") are given as metadata.
- Ctx Regions
  - EAD_scATAC_PredictiveDistribution_CtxRegions_Topics: Loom file containing the cisTopic cell-Topic tSNE coordinates, ctx region probabilities (multiplied by $10^6$), and topic enrichment. This data set contains 15,387 cells, 129,553 ctx regions, and 49 topics. The labels transferred from the 10x scRNA-seq data ("RNA labels") and the clusters derived from SNN clustering with Seurat on the topic-cell matrix ("Seurat_res_1.2") are given as metadata.

scRNA-seq—3531 cells
- EAD_scRNAseq_LinkBasedandSeurat: Loom file containing dimensionality reductions (based on analysis with Seurat and based on pySCENIC regulons), gene expression, and link-based regulon enrichment (regulons formed by performing the motif enrichment step of the SCENIC (Aibar *et al*, 2017) workflow on the regions linked to each gene). This data set contains 3,531 cells, 8,744 genes, and 161 regulons. The labels given by cell clustering with Seurat ("Seurat_res_1.2") and the experimental run ("Experiment run") are given as metadata.
- EAD_scRNAseq_SCENICandSeurat: Loom file containing dimensionality reductions (based on analysis with Seurat and based on pySCENIC regulons), gene expression, and pySCENIC regulon enrichment (motif and ChIP-seq based). This data set contains 3,531 cells, 8,744 genes, and 175 regulons. The labels given by cell clustering with Seurat ("Seurat_res_1.2"), the experimental run ("Experiment run"), the labels transferred from Ariss *et al* ("Ariss labels"), and the labels transferred from the scATAC-seq data ("ATAC labels") are given as metadata.
- EAD_scRNAseq_SCENICandSeurat_regulonsfromAriss: Loom file containing dimensionality reductions (based on analysis with Seurat and based on pySCENIC regulons), gene expression, and pySCENIC regulon enrichment (using the regulons derived from Ariss *et al*). This data set contains 3,531 cells, 8,744 genes, and 140 regulons. The labels given by cell clustering with Seurat ("Seurat_res_1.2"), the experimental run ("Experiment run"), the labels transferred from Ariss *et al* ("Ariss labels"), and the labels transferred from the scATAC-seq data ("ATAC labels") are given as metadata.

Virtual EAD—5370 cells
- Janelia
  - Janelia_Accessibility_AggSumProb: Loom file containing the virtual eye-antennal disc coordinates and the Janelia region probabilities based on the aggregation of the probabilities of the ctx regions that overlap with the Janelia enhancer (multiplied by $10^6$). This data set contains 5,526 cells and 740 Janelia regions. The labeling of the cells in the virtual eye-antennal disc ("Zone") is given as metadata.
  - Janelia_Functionality_ImageRegistration: Loom file containing the virtual eye-antennal disc coordinates and the Janelia enhancer activity patterns mapped from the images into the virtual eye-antennal disc. This data set contains 5,058 cells and 454 Janelia mapped images (corresponding to 390 Janelia

enhancers). The labeling of the cells in the virtual eye-antennal disc ("Zone") is given as metadata.

- ATAC
  - Pseudotime-based_ATAC_VE_CtxRegions + Topics: Loom file containing the virtual eye-antennal disc coordinates, the ctx region probabilities (multiplied by $10^6$), and the topic enrichment. This data set contains 5,526 cells, 129,553 ctx regions, and 49 topics. The labeling of the cells in the virtual eye-antennal disc ("Zone") and the cell type labels based on the scATAC-seq data ("Cell type") are given as metadata.
- RNA
  - Pseudotime-based_RNA_VE: Loom file containing the virtual eye-antennal disc coordinates, gene expression, and the pySCENIC regulon enrichment (derived from the 10x scRNA-seq data). This data set contains 5,370 cells, 8,744 genes, and 175 regulons. The labeling of the cells in the virtual eye-antennal disc ("Zone"), the cell type labels based on the scRNA-seq data ("Cell type"), the labels transferred from Ariss et al ("Ariss labels"), and from the scATAC-seq data ("ATAC labels") are given as metadata.

*UCSC*

Custom tracks

- Bulk ATAC DGRP regions: Bed file containing the 38,179 regions found accessible across the 50 bulk ATAC-seq profiles from *Drosophila* inbred lines.
- Color—Standardized R2G Coor + Genie3 50 kb: BigInteract track containing links between enhancers and target genes. The track is colored by the sign of the link, which can be positive (green) or negative (red). Default threshold: 0.
- Ctx_regions: Bed file containing the 129,553 ctx regions accessible in the eye-antennal disc in the *Drosophila* genome.
- Janelia lines: Bed file containing the coordinates of the enhancers tested by the Janelia FlyLight Project (with Janelia line ID).
- Optix-GFPVSRest_logFC1: Bed file containing the regions differentially accessible in the Optix-GFP$^+$ cells compared to the Optix-GFP$^-$ cells (with *P*-value < 0.05 and logFC > 1)
- REDFly (BED): Bed file containing the coordinates of the enhancers contained in the REDfly database (Rivera et al, 2019).
- Score—Standardized R2G Coor + Genie3 50 kb: BigInteract track containing links between enhancers and target genes. The transparency of the links represents the Random Forest importance of the enhancer-to-gene link. Default threshold: 0.
- So ChIP-seq peaks: Sine oculis (so) ChIP-seq peaks determined by MACS2 peak calling after remapping the data from Jusiak et al (2014) to the 3$^{rd}$ 2017 FlyBase release (*D. melanogaster* r6.16) genome.

Eye-Antennal Disc Hub (Bravo González-Blas et al, 2019) @ aertslab.org

- 10x topics: Topic bigwig files representing the region-topic scores obtained from the analysis of the 10x scATAC-seq with cisTopic (Bravo González-Blas et al, 2019) (v0.2.2).
- Aggregate scATAC—Cell sorting: Aggregate profiles from the FAC-sorted Optix-GFP$^+$ and sens-GFP$^+$ cells as normalized bigwig files.
- ATAC DGRP—Eye disc: Bulk ATAC-seq profiles from the 50 DGRP lines used in this study as normalized bigwig files.

- ATAC Aggr EAD Clusters: Cell type-specific (based on clustering on the topic-cell matrix) aggregate profiles from the 10x scATAC-seq analysis as normalized bigwig files.
- Bulk ATAC—Cell sorting: Bulk ATAC-seq profiles from the FAC-sorted Optix-GFP$^+$ and sens-GFP$^+$ cells as normalized bigwig files.
- ENCODE ChIP-seq: Normalized bigwigs from the ChIP-seq experiments of Prospero, Nerfin-1, and l(3)neo38 (and controls) retrieved from ENCODE.
- ENCODE Normalized ChIP-seq: Control normalized bigwig files from the ChIP-seq experiments of Prospero, Nerfin-1, and l(3)neo38 retrieved from ENCODE.
- Grh ChIP-seq: Normalized bigwigs from the Grainyhead ChIP-seq experiments performed by Jacobs et al (2018) after remapping to the 3$^{rd}$ 2017 FlyBase release (*D. melanogaster* r6.16) genome.
- Predictive accessibility: Barchart track representing the region-cell probabilities (multiplied by $10^6$) per cell type for each region.
- RNA Aggr EAD Clusters: Cell type-specific (based on Seurat clustering) normalized bigwigs containing 10x scRNA-seq reads.
- scRNA Gene expression gene: Barchart track representing the normalized UMI counts per cell type for each gene.
- scRNA Gene expression transcript: Barchart track representing the normalized UMI counts (multiplied by $10^2$) per cell type for each transcript.
- so ChIP-seq: Normalized bigwigs from the Grainyhead ChIP-seq experiments performed by Jusiak et al (2014) after remapping to the 3$^{rd}$ 2017 FlyBase release (*D. melanogaster* r6.16) genome.
- TF perturbations: Bulk ATAC-seq profiles from the GMR-GAL4 UAS-TF (and control) lines included in the genetic screen.

## Data availability

The data generated for this study have been deposited in NCBI's Gene Expression Omnibus and are accessible through GEO Series accession number GSE141590 (http://www.ncbi.nlm.nih.gov/geo/query/acc.cgi?acc = GSE141590). We also provide a SCope session at http://scope.aertslab.org/#/Bravo_et_al_EyeAntennalDisc with the processed single-cell data and a UCSC hub (http://ucsctracks.aertslab.org/papers/Bravo_et_al_EyeAntennalDisc/hub.txt) and session at http://genome.ucsc.edu/s/cbravo/Bravo_et_al_EyeAntennalDisc with the processed aggregate and bulk ATAC-seq profiles, enhancer-to-gene links, and ChIP-seq tracks.

The code for spatial single-cell omics integration and inference of enhancer-to-gene links is included in the ScoMAP R package, with detailed tutorials, at https://github.com/aertslab/ScoMAP. The code to reproduce the figures in this article is available at https://github.com/aertslab/Bravo_et_al_EyeAntennalDisc/.

**Expanded View** for this article is available online.

## Acknowledgements

This work is funded by an ERC Consolidator Grant to S. Aerts (724226_cis-CONTROL), by the Special Research Fund (BOF) KU Leuven (grant PF/10/016, to S. Aerts) and F.W.O (grants G.0791.14, G.0C04.17 to S.Aerts and PhD fellowship 11F1519N to C.B.G.-B). Stocks obtained from the Bloomington Drosophila Stock Center were used in this study. Single-cell infrastructure was funded by the Hercules Foundation (grant no. AKUL/13/41). Computing was performed at the Vlaams Supercomputer Center (VSC). The VIB

BioImaging Core (Leuven platform) provided valuable insight on image processing. The authors thank to Maximilian Haeussler and Kate Rosenbloom (UCSC Genome Browser) for their help in data visualization on the UCSC Genome Browser; to the various groups that make curated position weight matrices publicly available, including T. Hughes (cis-bp), M. Bulyk (UniPROBE), A. Mathelier (JASPAR), V. Makeev (Hocomoco), and many others; to the Janelia FlyLight Project for publicly providing images and reporter lines to assess enhancer activity on imaginal discs and CNS in *Drosophila*; and to the ENCODE Consortium for publicly providing raw and processed data of a wide range of genomic assays.

## Author contributions

SAe, CBG-B, KD, and DK conceived the study; X-JQ, IIT, KD, DK, VC, SM, DM, and SP performed the experimental work; CBG-B conceptualized the computational approaches and performed the computational analyses with help of RD-R, ITT, KD, DK, GH, MW, and SAi; CBG-B made the figures and CBG-B and SAe wrote the manuscript.

## Conflict of interest

The authors declare that they have no conflict of interest.

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
