## [Review Process File · Molecular Systems Biology]

Identification of genomic enhancers through spatial integration of single-cell transcriptomics and epigenomics

Carmen Bravo González-Blas, Xiao-Jiang Quan, Ramon Duran-Romaña, Ibrahim Taskiran, Duygu Koldere, Kristofer Davie, Valerie Christiaens, Samira Makhzami, Gert Hulselmans, Maxime de Waegeneer, David Mauduit, Suresh Poovathingal, Sara Albar and Stein Aerts.

Review timeline:

Submission date:	3 rd January 2020
Editorial Decision:	7 th February 2020
Revision received:	13 th March 2020
Accepted:	18 th March 2020

Editor: Maria Polychronidou

Transaction Report:

1st Editorial Decision

7th February 2020

Thank you again for submitting your work to Molecular Systems Biology. We have now heard back from two of the three referees who agreed to evaluate your study. In the interest of time and since the evaluations of the two reviewers are rather similar we have decided to proceed with these two available reports. As you will see below, the reviewers are quite positive about the work and mostly raise relatively minor issues, which we would ask you to address in a revision.

The reviewers' recommendations are quite clear and I think that there is no need to repeat the points listed below. Please feel free to contact me in case you would like to discuss in further detail any of the issues raised. In line with the comment of reviewer #2 (who is not a computational biologist) we would ask you to make sure that the main findings and key concepts are easily accessible to the broad audience of MSB.

REFEREE REPORTS

Reviewer #2:

Manuscript MSB-20-9438

Identification of genomic enhancers through spatial integration of single-cell transcriptomics and epigenomes

In this manuscript, Gonzalez-Blas et al. generate scRNA-seq and scATAC-seq profiles using the *Drosophila* eye-antennal disc as a model, performed a spatiotemporal mapping of these data to a virtual representation of the eye disc using a computational approach, and validated predicted enhancer activities using enhancer lines from the Janelia Flylight project. They then used a computational strategy to infer enhancer to gene relationships, and used chromatin accessibility QTLs to validate the functionality of TF motifs and thus enhancer function. And finally, using this combined approach the authors identified Prospero as a neuronal differentiation factors acting through binding of a GGG motif.

Overall, this is a very important paper, as it integrates cutting-edge technologies, scRNA-seq and scATAC-seq, with novel bioinformatic tools to predict gene regulatory networks and spatiotemporal enhancer activities. Thus, this manuscript is ideally suited to be published in *Molecular Systems Biology*. However, in its current state, although in principal very well written, it is very hard to digest and mostly understandable for bioinformaticians. Thus, I recommend that the paper is re-written to make it accessible to a large target audience. For this reason, I will comment for now only on the major issues (which I can judge).

MAJOR POINTS:

1) One concern is the quality of the scRNA-seq data. The question here is: how many cells does a disc have? How does this compare to the number of sequenced cells? how does this relate to rare cell types? Related to that: how were high-quality cells defined? This needs to be more properly explained. The same is true for the scATAC-seq data. There is also a bias, around 3000 single cell transcriptomes vs 15000 single cell epigenomes. Although the coverage for ATAC-seq needs to be higher, the authors should comment on this bias.

2) The authors use quite a few none-state-of-the-art tools, however they do not explain the purpose. What is cisTopic, SCENIC etc.? In order to make it readable and understandable for a large audience, the authors should explain briefly what these tools do and why they were used and not others. The same is true for other parts: what is for example a gini score, how is this score defined in easy words? On page 7: what is a topic in biological terms? What does it represent? For example, Figure S6: what do the graphs show? A brief explanation in the text would be helpful.

3) The figures are way too dense and subpanels way too small. Like that, it is

impossible to judge the claims. For example, Fig. 2f: what does the figure show? The same is true for Fig. S3f or Fig S4b (nothing readable at all). How to judge claims from these images? This is true for all Figures and most subpanels, for example Figure 4 is unreadable. This needs to be adjusted.

4) In the same line, sometimes the authors do not refer to the appropriate subpanels but just mention a whole Figure (like Figure S4). Like that, it is very hard to follow the paper and understand which claim is supported by which data. Are all subpanels necessary? Are they described in the text?

5) One important issue is the description of new tools/strategies developed in this paper. For example, the authors developed a tool to infer enhancer-to-gene relationships, and the description is "... using forest regression models, which assess non-linear relationship." (p-14). It is hard to grasp what this means, thus it would be helpful to describe with "easy" (non-bioinformatic) words in the text what these tools do, and give a more detailed (bioinformatic) description in the M&M part.

6) In the spatio-temporal mapping of single-cell omics, the authors say that they used a template of the eye-antennal disc with 5,058 virtual cells. Does this mean the eye-antennal disc consists of 5000 cells? How does this compare to the 3000 single cell transcriptomes? Does this mean that only 3/4 of the cells of the eye-antennal disc have been sequenced?

7) The authors show that enhancers for same gene seem to have same accessibility profile, in particular for TFs as target. This is an intriguing concept that could be explained in biological terms. What might be the reasons? Is it really a question of redundancy or more a question of robustness? Could the data be used to analyze in this direction? (page 15- line 461-462)

8) Do the authors expect the 22 enhancers that regulate a gene on average to all drive similar expression patterns? How many are (expected to be) repressive? What is the difference between repressive and activating enhancers?

9) What is the average size of ATAC-seq regions identified in this study in comparison to the regions used for the *Janelia* farm enhancer reporter lines? Similar size or discrepancy?

10) For the Prospero analysis: why did the authors use over-expression instead of RNAi for functional analysis? As Pros over-expression results in pupal lethality, this would be important to show the eye phenotype (combination with Gal80 to stage-specifically induce RNAi activity). The general question is whether the Pros data/part need to be included in this manuscript, in particular as this analysis suggests but not proves that Pros is a key regulator of late-born photoreceptors. Leaving this data out and sparing it for another paper would have the advantage that the other parts could be extended and better explained.

Minor points:

- acronyms need to be explained in main text & legends

- single cell vocabulary should be more accessible to researcher outside the field

Reviewer #3:

In González-Blas et. al. the authors present a tour-de-force on the analysis and description of the eye-antennal disc in fly. The work is pioneering in the way the authors integrate the datasets they generated to not only uncover meaningful new biology, but to go beyond an atlas and dissect out core regulatory mechanisms, how those mechanisms can be altered and perturbed by genotype, and then directly test candidates to uncover a novel network of regulatory control. This paper is a great example of the power of single-cell genomics strategies used correctly as opposed to "single-cell for the sake of single-cell". My comments are all very minor, and I believe this work is well suited for publication and already seems very polished. Overall the manuscript is very well written and very detailed. The authors present the work clearly and, while it is fairly long, I believe the length is warranted as it serves as a roadmap of sorts on how these types of data can, and should, be analyzed, including multiple analysis approaches to confirm and bolster their findings. Also the introduction captures the state of the field very nicely and the discussion puts this work into context nicely. Lastly, the methods are well detailed and the data is available.

Comment throughout - the text in the figures is a bit small - it may just be due to the pdf form for the submission.

Figure 1c - the tSNE plots are quite small and a bit hard to interpret with the size

For the ATAC data, two topics were batch effect topics - did these then get removed from the matrix for further analysis? Ie in the subsequent tSNE / cluster assignment / etc... Is it possible to remove the bias captured by the topic from the dataset?

Line 301 - the finding of the "lag effect" is really interesting - could this be expanded upon by using pseudotemporal ordering approaches to capture these transitions and identify sets of loci that appear to open prior to expression and see if they are enriched for certain activating TFs (eg Ato below)? Even without additional analysis, it is a very interesting finding.

Figure 2 - some of the text is too small in this figure as well

Line 442 - are any of the *Janelia* enhancers highly negatively linked? These could be interesting.

Line 446 states an average of 13 enhancers per TF gene and 11 per other gene; above (line 440) states 22 median enhancers per gene. Is the discrepancy due to i) mean vs median or ii) the 22 are positive OR negative associations for each gene? This information is not contradictory, it could just be framed a little more clearly.

Line 470 - "not correlated", but correlation is listed only as < 0.5 , what is the actual number? 0.5 is not necessarily not correlated, just not strongly correlated.

I find the analysis of repressors particularly interesting. It would be very compelling to assess instances where genes are controlled by a mix of repressive and activating elements - eg element A - repressor and B - activator and identify cells with only A, only B, or A and B. Broadly just understanding the balance between repression and activation and whether they compete at the same gene etc...

Paragraph at 490 - is there not a risk of a circular analysis in using the enhancer-gene links that were derived from correlating ATAC and RNA data to create a new gene activity matrix that then is compared with the RNA data? Maybe it is not used in that context, but it would seem that if one has RNA data that is required to generate the gene activity scores using this manner, then there is no real need to generate gene activity scores from ATAC data when the RNA data can just be used.

The caQTL analysis is excellent and very well done. It is a great way to tease out the effects of variation in elements.

1st Revision - authors' response

13th March 2020

Manuscript MSB-20-9438 - Rebuttal

Identification of genomic enhancers through spatial integration of single-cell transcriptomics and epigenomes

Reviewer #1:

No comments were received.

Reviewer #2:

In this manuscript, Gonzalez-Blas et al. generate scRNA-seq and scATAC-seq profiles using the *Drosophila* eye-antennal disc as a model, performed a spatiotemporal mapping of these data to a virtual representation of the eye disc using a computational approach, and validated predicted enhancer activities using enhancer lines from the *Janelia Flylight* project. They then used a computational strategy to infer enhancer to gene relationships, and used chromatin accessibility QTLs to validate the functionality of TF motifs and thus enhancer function. And finally, using this combined approach the authors identified Prospero as a neuronal differentiation factors acting through binding of a GGG motif.

Overall, this is a very important paper, as it integrates cutting-edge technologies, scRNA-

seq and scATAC-seq, with novel bioinformatic tools to predict gene regulatory networks and spatiotemporal enhancer activities. Thus, this manuscript is ideally suited to be published in Molecular Systems Biology. However, in its current state, although in principal very well written, it is very hard to digest and mostly understandable for bioinformaticians. Thus, I recommend that the paper is re-written to make it accessible to a large target audience. For this reason, I will comment for now only on the major issues (which I can judge).

Thank you for the positive evaluation. We have rewritten several parts of the manuscript, further explained some of the concepts to make it accessible to a broad audience, re-adjusted the figures, and increased the figure font size for clarity.

MAJOR POINTS:

1) One concern is the quality of the scRNA-seq data. The question here is: how many cells does a disc have? How does this compare to the number of sequenced cells? how does this relate to rare cell types? Related to that: how were high-quality cells defined? This needs to be more properly explained. The same is true for the scATAC-seq data. There is also a bias, around 3000 single cell transcriptomes vs 15000 single cell epigenomes. Although the coverage for ATAC-seq needs to be higher, the authors should comment on this bias.

We agree with the reviewer that transparency about the quality of the data set is an important point that was not fully addressed in the original version. We have now included additional analyses that assess the quality of our scRNA-seq and scATAC-seq data, and provide more background and justification on the cell numbers and cell filtering thresholds.

Tissue coverage and effects of cell filtering

The eye disc comprises ~11,500 cells (Ariss et al., 2018), and we estimate that the combined eye- and antennal disc contains 15,000-20,000 cells. We have performed three 10X Genomics runs, aiming at 15,000 cells (5000 cells per lane). Compared to other tissues, such as our data on the larval and adult brain, the scRNA-seq runs on 10X Genomics for the eye-antennal disc consistently yielded more stringent CellRanger thresholds, resulting in relatively low cell counts, but with the highest quality (Fig R1). This yielded 3,531 cells combining the three runs; while for scATAC-seq we obtained much higher cell numbers using the default CellRanger threshold (15,000 cells).

Figure R1. Selection of high-quality cells using CellRanger and CellRanger ATAC. A. CellRanger plot showing the number of UMI counts per barcode in the first scRNA-seq run. In total, 483 high-quality cells were selected. **B.** CellRanger plot showing the number of UMI counts per barcode in the second scRNA-seq run. In total, 1,899 high-quality cells were selected. **C.** CellRanger plot showing the number of UMI counts per barcode in the third scRNA-seq run. In total, 1,149 high-quality cells were selected. **D.** CellRanger ATAC plot showing the number of fragments within aggregate peaks per barcode in the first scATAC-seq run. In total, 9,833 high-quality cells were selected. **E.** CellRanger ATAC plot showing the number of fragments within aggregate peaks per barcode in the second scATAC-seq run. In total, 5,554 high-quality cells were selected [Appendix Figure S4A-C, E, F].

We agree that this information was not provided in the previous version, and now we have added this information in Appendix Figure S4 [Page: 7]. From the CellRanger plots, it is clear that we could select many more cells for scRNA-seq, but we would have to compromise on the UMI and gene counts per cell. For example, if we apply the same filters as we applied in the Ariss *et al.* 2018 data set (minimally 1,000 UMIs per cell and 500 genes expressed), we would select 9,234 cells (Fig R2). However, this is not necessarily better, *provided that* the true cell types (and rare cell types) are well represented – in other words if the sampling size is high enough. Note that in the Human Cell Atlas, or Tabula Muris, none of the tissues achieves a 1X coverage, yet highly informative cell "models" can be achieved by sampling enough cells per tissue, until all cell types are well represented. The question is therefore, do the 3,531 highest-quality cells represent all cell

types? In the previous version we already extensively validated this, but we provide additional analyses below and in the revised version.

Figure R2. Seurat tSNE including 9,234 cells with at least 1,000 UMI counts and 500 genes expressed. Cells not included in our high-quality data set of 3,531 cells are shown in grey [Appendix Figure S4D].

Representation of cell types by comparison with Ariss *et al.*

When we compare our 3.5K data set with a larger data set that was previously published by Ariss *et al.* (2018), which contains 11,415 eye disc cells [Fig R3], we found that in our eye data we not only still identify rare populations (such as the Second Mitotic Wave (SMW), which comprises 2.5-5% of the eye cell population), but also obtain a similar representation of the eye cell types [Fig R3,4]. This aspect is now further discussed in the manuscript [Page: 6-7].

Figure R3. Comparison of cell types identified in the eye disc in Ariss *et al.* (2018) and this work. A. Ariss *et al.* (2018) tSNE based on the single-cell RNA-seq profiles of 11,415

cells of the eye disc, colored by the clusters defined by the authors [Appendix Figure S3A]. **B.** Bravo et al. (2020, this work) tSNE based on the single-cell RNA-seq profiles of 3,531 cells from the eye-antennal disc, colored by the clusters defined by Ariss et al. (2018) after label transferring with Seurat (Stuart et al., 2019) [Figure 1D]

Figure R4. Representation of cell types in the eye disc and data coverage in Ariss et al. (2018) and this work. **A.** Proportion of cell types in the eye disc based on the clusters defined by Ariss et al. (2018) in the original data set and in Bravo et al. (2020, this work) eye disc data after label transferring with Seurat (Stuart et al., 2019) [Appendix Figure S3E]. **B.** Normalized UMI counts per cell in Ariss et al. (2018) and Bravo et al. (2020, this work) [Appendix Figure S3A]. **C.** Normalized number of genes expressed per cell in Ariss et al. (2018) and Bravo et al. (2020, this work) [Appendix Figure S3B].

Rare cell types in the antennal disc

This is the first work including scRNA-seq data on the antennal disc. In this part of the tissue we were able to identify the cell populations from the different antennal rings, but also two rare subpopulations: (1) a subpopulation of *sens*⁺ cells in the second antennal ring (33 cells) and (2) a subpopulation expressing the transcription factor *twi* (54 cells) [Fig R5A]. The *sens*⁺ population corresponds to Johnston Organ Precursor (JOPs) cells, which are known to reside in the A2 antennal ring [Fig R5B]. Interestingly, there are only ~50-100 JOPs per antennal disc (Sen et al., 2010), meaning that this population represents only a 0.25-0.6% of the eye-antennal disc (~0.93% in our data set). The *twi*⁺ cells are ad epithelial cells (mesodermal myoblast), which are known to reside in imaginal discs (Beira and Paro, 2016). Based on the activity of a *Janelia* enhancer linked to *twi* [Fig R5C], and the lack of this cell type in the - only - eye disc data set from Ariss et al. (2018); we identified these cells as a minority population residing in the antennal disc [Page: 6]

Figure R5. Identification of rare cell populations in the antennal disc. **A.** Seurat tSNE based on 3,531 single-cell RNA-seq profiles from the eye-antennal disc colored by the expression of *senseless* (red) and *twist* (green). **B.** Expression of *senseless* in the antennal disc (marked with arrows). Image adapted from Nolo *et al.* (2010). **C.** Activity of an enhancer linked to *twist* in the antennal disc (marked with arrowheads) [Appendix Figure S3F]. Image adapted from the *Janelia Flylight Project*.

Additional quality metrics

- We find a median of 1.69% mitochondrial reads, with a maximum of 2.28%, below the recommended filtering threshold of 5%; while this information is not available in the Ariss *et al.* (2018) data set [Page: 28 (Methods)].
- To filter out potential doublets in our data set, we used DoubletFinder (McGinnis *et al.*, 2019). A total of 246 cells were excluded for posterior analyses, assuming a doublet rate of 7.5%, most of which comprise tightly packed ommatidial cells [Fig R6A,B] [Page: 5 and 28 (Methods)].
- We also assessed potential batch effects and biological biases. Particularly, mixing our three independent 10X runs, we observe that the clustering is driven by cell type, showing no apparent batch effect between the samples [Fig R6A,C] [Page: 5].

Figure R6. Doublet removal and assessment of batch effects. **A.** Seurat tSNE based on 3,531 single-cell RNA-seq profiles from the eye-antennal disc colored by cell type (Bravo et al., 2020; this work) [Figure 1B]. **B.** Seurat tSNE based on 3,531 single-cell RNA-seq profiles from the eye-antennal disc colored by DoubletFinder label [Appendix Figure S1G]. **C.** Seurat tSNE based on 3,531 single-cell RNA-seq profiles from the eye-antennal disc colored by experimental run [Appendix Figure S1D].

In conclusion, with 3,531 high-quality cells in the scRNA-seq data we obtained a representative atlas with the known cell types, including rare cell types, which match 1-1 with the cell types found by scATAC-seq. Sequencing additional samples at this stage would, given the required resources and time, in our view not balance against the benefits for the current manuscript. Adding additional cells from the existing runs is an option that we elaborated on above, but does not provide additional insight compared to using only the highest quality cells. Finally, with the raw data and codebase that we make available, the community can re-use this data set, and adjust the cell filtering thresholds.

Cell count and quality of the scATAC-seq data

For scATAC-seq we again used the default cell thresholds set by CellRanger, this time obtaining 15,387 cells from two 10X runs [Page: 7-8, 30 (Methods)], causing a strong difference in cell numbers compared with scRNA-seq, as commented by the reviewer. We agree that this can be confusing, and provide more detailed information on this difference [Page: 9, 28, 30 (Methods)]. Overall, given the sparsity of scATAC-seq, and the way that cisTopic deals with this sparsity, we find that using more cells in scATAC-seq is beneficial.

Definition of high-quality scATAC-seq cells

After the CellRanger threshold, we performed a variety of quality checks to make sure that we could use this entire data set. We filtered out only 379 cells based on the percentage of the fragments within bulk peaks ($< 20\%$; to assess noisy cells) and the total number of fragments in the cell (< 100 and $> 10,000$; to assess under-sequenced cells and potential doublets) (Fig R7A,B) [Page: 7, 30 (Methods)]. In addition, we used DoubletFinder (McGinnis et al., 2019) on the scATAC-seq gene activity matrix, identifying 13,848 high confidence singlets; and found two additional cell clusters as potential doublets based on the percentage of fragments in bulk peaks and the number of fragments (Fig R7C-F) [Page:10, 30-32 (Methods)].

Figure R7. Identification of high-quality scATAC-seq cells. **A.** Percentage of fragments in bulk peaks. Cells with less than 20% of the fragments in bulk peaks, were filtered out [Appendix Figure S6E]. **B.** Normalized (\log_{10}) number of fragments. Cells with less than 100 fragments or more than 100,000 fragments were filtered out [Appendix Figure S6F]. **C.** Annotation of singlets and doublets based on DoubletFinder (on the scATAC-seq gene accessibility matrix) [Appendix Figure S12C]. **D.** cisTopic cell tSNE (15,387 cells) colored by normalized number of fragments (\log_{10}) [Appendix Figure S8A]. **E.** cisTopic cell tSNE colored by normalized number of accessible regions (\log_{10}) [Appendix Figure S8B]. **F.** cisTopic cell tSNE colored by percentage of fragments that overlap bulk peaks [Figure S8C].

Potential batch effects and biological bias in the scATAC-seq data

As we performed two independent 10X runs, we verified how well the cells mix across these two replicates. We observed that the clustering is driven by cell type, showing no apparent batch effect between the samples [Fig R8A-B] [Page:9, 30-31]. Importantly, we find two topics that correlate with the experimental run (see *Methods*); and an additional topic enriched in female cells (identified as described by Cusanovich *et al.*, 2018) [Fig R8C-F]. Removal of these topics results in a similar cell grouping, showing that cell clustering is driven by cell type rather than by experimental run or sex (Fig R9) [Page:9, 30-31].

Figure R8. Identification of batch effect topics. **A.** cisTopic cell tSNE (15,387 nuclei) colored by annotated cell type [Figure 2D]. **B.** cisTopic cell tSNE colored by experimental run [Appendix Figure S8D]. **C.** cisTopic cell tSNE colored by topic 46 enrichment [Appendix Figure S8E]. **D.** cisTopic cell tSNE colored by topic 18 enrichment [Appendix Figure S8D]. **E.** cisTopic cell tSNE colored by assigned sex [Appendix Figure S8H]. **F.** cisTopic cell tSNE colored by topic 4 enrichment [Appendix Figure S8I].

Figure R9. Effect of the removal of batch effect topics on cell clustering. A. cisTopic cell tSNE using all topics (15,387 nuclei) colored by annotated cell type. **B.** cisTopic cell tSNE after removing the run-specific topics. **C.** cisTopic cell tSNE after removing the sex-specific topic. **D.** cisTopic cell tSNE after removing the run and the sex-specific topics [Appendix Figure S8J].

Compared to the scRNA-seq data set, we identify a similar distribution of cell types [Figure R10A]; including the rare *sens*⁺ and *twi*⁺ subpopulations in the antennal disc [Figure R10B]. To test whether 15,000 cells are needed to find these populations, we sampled 3,000, 6,000, 9,000 and 12,000 single-cell ATAC-seq and observe that at least 9,000 cells were needed to achieve the scRNA-seq resolution (Fig R11). Below this threshold, glia and *twi*⁺ cells and neuroblasts and early photoreceptors are undistinguishable, and with only 3,000 cells (as in our scRNA-seq data set), progenitors and head vertex cells are mixed. In addition, we find that even when taking the top 3,531 cells (i.e. same number of cells as in scRNA-seq) with the highest coverage, photoreceptor and brain cells are intermixed (Fig R12).

Figure R10. Cell diversity comparison between scRNA-seq and scATAC-seq. A. Proportion of cell types in the eye-antennal disc based on scRNA-seq and scATAC-seq analysis [Appendix Figure S10E]. **B.** cisTopic tSNE colors by the accessibility of the sens-F2 enhancer (chr3L:13397454-13399385, red) and a twi enhancer (chr2R:23048073-23048643).

Figure R11. Resolution on downsampled scATAC-seq data sets . A. cisTopic cell tSNE (3,000 nuclei) colored by annotated cell type **B.** cisTopic cell tSNE (6,000 nuclei) colored by annotated cell type. **C.** cisTopic cell tSNE (9,000 nuclei) colored by annotated cell type. **D.** cisTopic cell tSNE (12,000 nuclei) colored by annotated cell type.

Figure R12. cisTopic cell tSNE (3,531 nuclei with the highest coverage) colored by annotated cell type

Overall, these results show that for scRNA-seq, our sample of 3.5K cells is sufficient and integrates well with the 15K cells from scATAC-seq. Thus, having different sizes of both modalities is not necessarily a problem, which is encouraging for future studies that want to apply our method to integrated scATAC-seq and scRNA-seq data. We have included a more detailed description of these aspects in the manuscript [Page: 6-7, 9] and in the methods section [Page: 28, 30].

2) The authors use quite a few none-state-of-the-art tools, however they do not explain the purpose. What is cisTopic, SCENIC etc.? In order to make it readable and understandable for a large audience, the authors should explain briefly what these tools do and why they were used and not others. The same is true for other parts: what is for example a gini score, how is this score defined in easy words? On page 7: what is a topic in biological terms? What does it represent? For example, Figure S6: what do the graphs show? A brief explanation in the text would be helpful.

We have improved the explanation of the computational analyses for making it accessible to a broader audience, elaborating on SCENIC [Page: 7], cisTopic [Page: 8], the definition of gini score [Page:12] and other bioinformatics tools and concepts (see *Major Point 5*). SCENIC is a widely used method to infer gene regulatory networks from single-cell RNA-seq data (Fig R13) (Aibar et al., 2017). First, co-expression modules between TFs and potential target genes are derived using Random Forest or Gradient Boosting Machine models, which are machine learning algorithms used for solving regression problems -in this case-, where the TF expression is the variable to predict based on the expression target

genes. In other words, the co-expression modules for each TF will be formed by the genes that can better predict the TF expression. To reduce the number of false target genes -e.g. due to cascade activating effects-, the second step of SCENIC is to perform motif enrichment analysis around the TSS of the co-expressed genes, to only keep as targets those for which the motif of the TF is enriched. A *regulon* is the set of target genes co-expressed with a specific TF and surrounded by its binding sites. The enrichment of these regulons in the single cells can be estimated using a recovery curve approach (AUCell), in which genes in each cell are ranked based on their expression and a recovery curve is drawn based on whether genes in the regulon are present or not (i.e. one step in the Y-axis will be added in the gene in that ranking position is on the regulon). The Area Under the Curve is used as measurement of the enrichment of the regulon.

Figure R13. SCENIC workflow. **A.** In the first step, co-expression modules between transcription factors and candidate target genes are inferred with GENIE3 (Random Forest) or GRNBoost (Gradient Boosting). Each module consists of a transcription factor together with its predicted targets, purely based on co-expression. **B.** In the second step, each co-expressed module is analyzed with RcisTarget to identify enriched motifs; only modules and targets for which the motif of the TF is enriched are retained. Each TF together with its potential direct targets is a regulon. **C.** In the third step, the activity of each regulon in each cell is evaluated using AUCell, which calculates the Area Under the recovery Curve, integrating the expression ranks across all genes in a regulon. The AUCell scores are used to generate the Regulon Activity Matrix. This matrix can be binarized by setting an AUC threshold for each regulon, which will determine in which cells the regulon is “on”. **D.** The Regulon Activity Matrix can be used to cluster the cells (e.g. t-SNE) and, thereby, identify cell types and states based on the shared activity of a regulatory subnetwork. Image taken from Aibar *et al.*, 2017.

To our knowledge, SCENIC is the only scRNA-seq tool that integrates motif enrichment for the prediction of TF-target gene relationships. However, one of its caveats is that it does

not incorporate information about the accessibility of the regions surrounding the target genes; in other words, it also considers motifs that may not be within an accessible enhancer region. This aspect is tackled in Figure 4 in the manuscript, where we evaluate motif enrichment on the enhancers (inferred from the scATAC-seq data) linked to a gene, rather than in a space around the TSS. We have included these explanations in the manuscript [Page: 7, 15-16].

cisTopic (Bravo *et al.*, 2019) is one of the state-of-the-art and best performing tools for analysis of single-cell ATAC-seq data analysis (see the scATAC-seq benchmark by Chen *et al.*, 2019). cisTopic exploits a Bayesian topic modelling technique, Latent Dirichlet Allocation (LDA), to simultaneously classify regulatory regions into regulatory topics and cluster cells based on their regulatory topic enrichment [Fig R14]. In other words, topics are sets of co-accessible regulatory regions across cells, where each region belongs to a topic with a certain score (region-topic distribution), and the topic-cell enrichment represents their accessibility on each cell type. While other methods (such as LSI (Cusanovich *et al.*, 2018) or SnapATAC (Fang *et al.*, 2019)) first cluster cells and then perform differential accessibility analysis; we find that cisTopic, due to its probabilistic nature, is more suitable for this data as it comprises continuous -differentiating- rather than discrete populations. In fact, cisTopic is the best performing tool on this type of data, with the top benchmark score (1) on a scATAC-seq data set during hematopoiesis (Buenrostro *et al.*, 2018); compared to a score of 2.3 and 3 for SnapATAC and LSI, respectively. In addition, cisTopic allows to impute drop-outs in the single-cell ATAC-seq (by estimating the probability of each region in each cell), which is a key feature for the comparison of enhancer accessibility and activity and the linkage of enhancers and target genes.

Figure R14. cisTopic workflow. The input for *cisTopic* is an accessibility matrix, with regions as rows and cells as columns. Modelling with LDA is performed using a collapsed Gibbs sampler for the estimation of the region-topic and the topic-cell probability distributions. During this process, each region in each cell is iteratively assigned to a topic, based on the contribution of that topic to the cell and the contribution of that region (across

the data set) to that topic. The resulting probability distributions can be used for cell clustering (topic-cell) and region clustering (region-topic). Image taken from Bravo *et al.*, 2019.

Appendix Figure S7 presents three key aspects in the cisTopic workflow, which are (1) the definition of regulatory regions, (2) the selection of the number of topics and (3) the stability of the log-likelihood during the LDA sampling iterations. As we, and others (Chen *et al.*, 2019), have noticed that scATAC-seq clustering resolution improves when using genomic bins rather than bulk/aggregate peaks, we present a comparison of cisTopic models with different definitions of genomic regions: (1) narrow peaks as called by MACS2 from the bulk ATAC-seq profile of the wild type *Drosophila* eye-antennal disc; (2) bulk peaks defined by extending +/- 250 bp from the summits called by MACS2; and (3) cisTarget regions, defined by partitioning the entire non-coding *Drosophila* genome based on cross-species conservation, resulting in more than 136,000 bins with an average size of 790 bp (Herrmann *et al.*, 2012). Appendix Figure S7A,D,G show the selection of the number of topics, based on the highest log-likelihood, in each case; Appendix Figure S7B,E,H show the stabilization of the log-likelihood during the LDA sampling iterations (required for a valid model); and Appendix Figure S7C,F,I show the dimensionality reduction output of each analysis. As expected, we found that the cisTarget genomic bins provided a better resolution for glial, ad epithelial and brain cells. We have improved the explanations of these points on the figure legend and methods section [Page: 8].

The Gini index is a measure widely used in economics to measure the distribution of wealth (Gini, 1912), where a value of 0 indicates that wealth is equally spread (e.g. everyone has the same capital) and a value of 1 indicates maximal inequality (e.g. one person owns all the capital). This coefficient has been used previously in the single-cell field to select specifically expressed genes for downstream analyses, improving detection of rare cell types compared to methods based on variance (Li *et al.*, 2016; Torre *et al.*, 2018). For single-cell RNA-seq, a Gini index of 0 indicates an equal distribution of gene expression; while values close to 1 indicate that the gene is expressed in a small subset of cells. In this work, we use it to evaluate the specificity of chromatin accessibility of *Janelia* regions, where values close to 0 indicate that a region is equally accessible across all cells and values close to 1 indicate that a region is accessible in a specific subset of cells. By comparing this score with the correlation between the region accessibility and activity, we find that regions with a high gini score -specifically accessible- show coupled activity and accessibility; while regions with a low gini score show unpaired accessibility and activity [Figure 3E]. Additional analyses show that these generally accessible regions are binding

sites of the pioneer transcription factor Grainyhead, which promotes chromatin accessibility, but not necessarily activity [Figure 3F]. We have improved the explanation of these concepts on the manuscript [Page: 12].

3) The figures are way too dense and subpanels way too small. Like that, it is impossible to judge the claims. For example, Fig. 2f: what does the figure show? The same is true for Fig. S3f or Fig S4b (nothing readable at all). How to judge claims from these images? This is true for all Figures and most subpanels, for example Figure 4 is unreadable. This needs to be adjusted.

We apologize that our figures were too dense and difficult to read, and have now have adapted all the figures in the manuscript to ensure their readability, including:

- Figure 2 has been reorganized and panels G and H have been sent to supplement [Appendix Figure S12B,E]. Figure 2F shows the validation of the identity of the cells anterior to the morphogenetic furrow (AMF) in the scATAC-seq data. Briefly, we FAC-sorted AMF cells based on the activity of the Optix2/3 enhancer and performed bulk ATAC-seq. Regions differentially accessible in the Optix2/3-GFP⁺ sample compared to the Optix2/3-GFP⁻ sample are enriched in the cells previously labelled as AMF, overlap with the AMF topics and show enrichment for the Optix motif. We consider that this panel is important as it shows how to exploit FAC-sorted bulk ATAC-seq data (available for many cell types in many species and systems) to annotate single-cell data sets; and we have improved its explanation on the manuscript [Page: 9-10].
- In Appendix Figure S3F, we have removed the images showing the activity of enhancer linked to *twi* activity in other imaginal discs and zoomed-in in the antennal-disc. We think that this panel is important because it provides evidence that the adepithelial cells found in the data set are rare population residing in the antennal disc, together with the lack of adepithelial cells in the only-eye disc data set from Ariss *et al.* (2018).
- In Appendix Figure S5B, we have highlighted and enlarged the font of genes associated to their TF by GeneMANIA (Warde-Farley *et al.*, 2010). We think this panel is important because it helps visualizing what regulons are - a TF with their direct target genes (based on co-expression and motif enrichment); and how several regulons can coexist on a cell type.

- Figure 4 has been reorganized by showing the aggregate profiles below the links in the appendix figures [Appendix Figure S18], panel G [Appendix Figure S20H], and two subpanels on panel H to supplement [Appendix Figure S23].

4) In the same line, sometimes the authors do not refer to the appropriate subpanels but just mention a whole Figure (like Figure S4). Like that, it is very hard to follow the paper and understand which claim is supported by which data. Are all sub-panels necessary? Are they described in the text?

Thank you for noticing this. We have better referenced each of the panels in this figure [Page: 7], and improved the layout for Appendix Figure S4B.

5) One important issue is the description of new tools/strategies developed in this paper. For example, the authors developed a tool to infer enhancer-to-gene relationships, and the description is "... using forest regression models, which assess non-linear relationship." (p-14). It is hard to grasp what this means, thus it would be helpful to describe with "easy" (non-bioinformatic) words in the text what these tools do, and give a more detailed (bioinformatic) description in the M&M part.

We have improved explanations in the main text [Page:13] and the methods section [Page: 33-34] and created a new R package called ScoMAP (Single-Cell Omics Mapping into spatial Axes using Pseudotime ordering), available at: <https://github.com/aertslab/ScoMAP>. ScoMAP comprises the new strategies presented in the text namely (1) the mapping of single-cell omics data into virtual cells [Fig 3] and (2) the inference of enhancer-to-gene relationships (using Random Forest models and/or Pearson correlation) from multi-omics data, applicable to virtual and real multi-omics data sets [Fig 4]. We provide an extensive tutorial, using the eye-antennal disc as data set, with detailed step-by-step explanations (available at: <https://github.com/aertslab/ScoMAP>; under the 'Vignettes' section). In addition, we also provide the code to generate the subpanels included in the main figures at: https://github.com/aertslab/Bravo_et_al_EyeAntennalDisc.

6) In the spatio-temporal mapping of single-cell omics, the authors say that they used a template of the eye-antennal disc with 5,058 virtual cells. Does this mean the eye-antennal disc consists of 5000 cells? How does this compare to the 3000 single cell transcriptomes? Does this mean that only 3/4 of the cells of the eye-antennal disc have been sequenced?

As explained in *Major Point 1*, an eye-antennal disc comprises 15,000-20,000 cells; and while we have sampled only 3.5K cells, we are able to identify rare populations (below 0.5% of the entire population in the eye disc, such as the Johnston Organ Precursors and ad epithelial cells). In order to assess the effect of the size of the virtual template in our results, we have repeated our main analyses comparing a mini-template (with 1,265 cells), the medium-size template previously presented (with 5,088 cells) and a big-size template (with 20,333 cells) [Fig R15] [Page: 11, 32 (*Methods*); Appendix Fig S13].

Figure R15. Representation of Figure 3B using templates of different sizes. Top: Seurat tSNE (3,531 cells) colored by gene expression. Middle top: Mini virtual eye (1,265 cells) colored by gene expression. Middle bottom: Medium-size virtual eye (5,058 cells) colored by gene expression. Bottom: Big-size virtual eye (20,333 cells) colored by gene expression [Appendix Figure S13A].

We selected the initial size of the virtual eye (with 5,058 cells) to sample as many single-cell ATAC-seq profiles (~15K cells) without over-sampling single-cell RNA-seq profiles (~3.5K). Indeed, in the medium-size virtual eye, each cell is sampled a median of 1.07 times (mean: 1.6); while in the mini virtual eye, each cell is sampled a median of 0.42 (mean: 0.5; meaning that not all profiles are exploited); and in the big-size virtual eye each cell is sampled a median of 4 times (mean 6.4; meaning that profiles are re-used several time) (Fig R16). The over-sampling effect on the big eye can be observed clearly in the domain anterior to the morphogenetic furrow, where the vertical expression patterns are

artifacts of filling that bin during the cell mapping with few cells (for each bin of 211 cells, 42 cells are available) (Fig R15). Given that we now make an R package available, when additional cell numbers become available future in the for example within the Fly Cell Atlas, a user can modify the size of the template.

Figure R16. Comparison of the number of cells per cell type between the real scRNA-seq profiles and using templates of different sizes [Appendix Figure S13B].

With regards to the comparison between enhancer accessibility and activity using the virtual template, we mapped the *Janelia* enhancer activity images to the additional templates, and found similar results despite the size of the template (Fig R17). In this analysis, the main limitation is the original quality and resolution of the *Janelia* images (as downloaded from *Janelia* Flylight in png format), which is below single-cell resolution.

Figure R17. Inferring the relationship between enhancer accessibility specificity (as gini score over the region-cell probabilities) and the correlation between accessibility and activity using templates of different sizes. **A.** Relationship between the correlation between the accessibility and the activity of the regions and their distribution (as gini score) using the mini virtual eye. **B.** Relationship between the correlation between the accessibility and the activity of the regions and their distribution (as gini score) using the

medium-size virtual eye. C. Relationship between the correlation between the accessibility and the activity of the regions and their distribution (as gini score) using the big-size virtual eye [Appendix Figure S13C-E].

With regards to the enhancer-to-gene links, we have inferred the links using the mini virtual eye and the big-size virtual eye and compared to our previous links. Applying the same binarization method (BASC binarization on the RF importance and taking values on the 1st or 99th quantiles over the correlation distribution, see *Methods*), using the mini virtual eye we find 189,056 interactions (95,962 positive; 93,094 negative); using the medium-size virtual-eye we find 182,713 interactions (95,484 positive; 87,229 negative) and using the big-size virtual-eye we find 221,858 interactions (118,656 positive; 103,202 negative). Importantly, between all the analysis 118,572 inferred links appear in all analyses (65% of the links in the medium size analysis), with 146,009 links appear in both the mini-eye and the medium-size analyses (80% of the links in the medium-size analysis) and 137,861 links appear in both the medium-size and the big-size analyses (75% of the links in the medium-size analysis). In addition, the random forest importances and correlation values are highly correlated between the analyses (0.89 and 0.79 between the mini virtual eye and the medium-size virtual eye; and the 0.95 and 0.83 between the medium-size and the big-size, respectively on average), which indicates that differences in the links are rather due to the binarization step (Fig R15).

In conclusion, we find that our results are equivalent despite the size of the template, and now provide more guidance to the reader on how to determine the optimal size of the template, when using our R package.

7) The authors show that enhancers for same gene seem to have same accessibility profile, in particular for TFs as target. This is an intriguing concept that could be explained in biological terms. What might be the reasons? Is it really a question of redundancy or more a question of robustness? Could the data be used to analyze in this direction? (page 15- line 461-462)

Thank you for the suggestion, we have extended the analysis regarding shadow enhancers in the text [Page: 14; Appendix Figure S20E-F; S21]. Indeed, we find about of 3-4 shadow enhancers (based on accessibility; with a median of 4 and a mean of 8 correlated enhancer-enhancer pairs) per gene; while in TF genes we find 5-6 shadow enhancers compared to 3-4 found in non-TF genes (H_0 : average number of enhancer-enhancer pairs with a correlation above 0.8 for TF genes (13) \leq Average number of enhancer-enhancer pairs

with a correlation above 0.8 for non TF genes (7); p-value: 0.006). In addition, we find that 73% of the *Janelia* regions that overlap these correlated enhancer-enhancer pairs also show overlapping activity (Fig R18).

Figure R18. Genes are regulated by many enhancers, some of which show redundant activity (and accessibility). Enhancer-to-target links for *hth* (A) and *Appl* (B). Top: Links between enhancer and target genes, whose height represent their importance. Middle top: cisTarget regions. Middle: Genome annotation. Middle bottom: Normalized aggregate ATAC-seq profiles per cell type. Bottom: Enhancer activity (images taken from the *Janelia* Flylight Project), and virtual eye colored by the predicted enhancer accessibility (green)

and the expression of the gene (red). Redundant enhancers are highlighted in grey [Appendix Figure S21].

The multiplicity of enhancers with the same function, known as shadow enhancers, has an evolutionary basis but their role in gene regulation is not yet fully understood. One interpretation is that they provide robustness during development (Osterwalder et al., 2018). In addition, redundant enhancers can compensate when an enhancer is affected by a loss-of-function mutation or deletion (Frankel et al., 2010; Perry et al., 2010).

We agree that the reviewer's question is very interesting (whether shadow enhancers underlie redundancy or robustness), but this is a very challenging problem in the field that we feel lies outside the scope of our manuscript. If the reviewer has specific suggestions how we can address this general question further, we are happy to perform additional analysis. We have performed two analyses in this direction: (a) are shadow enhancers less conserved in evolution? (b) do shadow enhancers have less variation in the population?

(a) We compared the sequence conservation between isolated enhancers and clustered/shadow enhancers using PhastCons (27 species). We found that shadow enhancers are equally (even slightly more) conserved compared to isolated enhancers (Fig R19A, H_0 : Average conservation score on redundant enhancers (0.585) \leq Average conservation score on non-redundant enhancers (0.567) ;p-value: 1.2×10^{-13}). This may point to robustness/cooperativity, rather than redundancy, but we do not feel comfortable to make this conclusion based on this observation. We do mention this in the text.

(b) Using SNPs and caQTLs that we derive from our bulk ATAC-seq data across 50 DGRP lines, we compared the number of caQTLs between isolated versus shadow enhancers. We found that there are less caQTLs per shadow enhancer compared to isolated enhancers (Fig R19B, H_0 : Number of caQTLs on redundant enhancers (mean: 0.12) \geq Number of caQTLs on non-redundant enhancers (mean:0.20); Wilcoxon rank sum test p-value: 2.48×10^{-5}).

Figure R19. Redundant enhancers are more conserved compared to non-redundant enhancers. **A.** Mean conservation score (PhastCons) [Appendix Figure S20E]. **B.** Number of caQTLs per region [Appendix Figure S20F].

8) Do the authors expect the 22 enhancers that regulate a gene on average to all drive similar expression patterns? How many are (expected to be) repressive? What is the difference between repressive and activating enhancers?

1. Based on a set of enhancers for which we have in vivo enhancer-reporter activity data, we indeed find that many enhancers with similar accessibility also show similar activity. Overall, we find a median of 4 correlated enhancers per gene (with correlation > 0.8 , see *Major Point 7*), meaning that the remaining 18 enhancers have different patterns [Page: 13-14]. As an example of the combinatorial regulation of gene expression, we have analyzed 3 validated enhancers on the *glass* locus that have been reported to regulate gene expression in different cell types (Fritsch et al., 2019) (i.e. *glass* is expressed posterior to the morphogenetic furrow, however different enhancers regulate its expression in early photoreceptors, late photoreceptors and interommatidial cells) [Appendix Fig S19].

2. We find a median of 11 positively linked enhancers and 11 negatively linked enhancers per gene [Page: 13-14]. Importantly, due to the sparse nature of single-cell technologies, negative relationships tend to be noisier than positive

relationships (i.e the absence of expression/accessibility may be due to technical drop-outs). For comparing positively and negatively linked enhancers, we have considered regions involved in links with a correlation > 0.1 or < -0.1 (excluding regions involved in both positive and negative links), finding 13,215 and 2,927 high-confidence positive and negative relationships, respectively.

3. We have compared both types of regions through several analyses and found that the main difference is in their motif content. In negatively-linked enriched regions we identify motifs linked to *so*/*Optix*, *Lz* and *Blimp-1*; which are known repressors. On the activating enhancers we find *Lola-T/K*, *AP-1* and *Onecut* motifs; among others (Fig R20) [Page: 13-14].

Fig R20. Motif enrichment analysis in activating and repressing enhancers. Number of positive and negative links, with representative enriched motifs in each category with Normalized Enrichment Score (NES) [Figure 4G].

9) What is the average size of ATAC-seq regions identified in this study in comparison to the regions used for the *Janelia* farm enhancer reporter lines? Similar size or discrepancy?

Excellent point. The average size of *Janelia* enhancers is 3,059 bp (median: 3,256 bp); while the average size of enhancers found in this study is 858 bp (median 726 bp). In fact, the enhancers defined by scATAC-seq in this work permit to delineate which are the candidate enhancers within the relatively large *Janelia* regions [Fig R21A].

Importantly, we find that only 25% of the *Janelia* regions contain only one ATAC peak and 60% contain less than two enhancers. In some of these cases the activity pattern is the result of the activity of different enhancers contained within the same *Janelia* region [Figure R21B]; while in others one of the enhancers correlates with the activity pattern [Figure R21C]. For assessing the correlation between accessibility and activity and

calculating the accessibility gini score, we have calculated the accessibility of Janelia region by aggregating the accessibility of all the enhancers contained in the region, as it is a good representation in most of these cases, but has its limitations. We have discussed these examples in the text and hope that this is now even more transparent for the reader [Page 11-12; Appendix Figure S15].

Figure R21: Wide Janelia regions contain multiple enhancers. **A.** Example of a Janelia region containing one enhancer (highlighted) [Appendix Figure S15A]. **B.** Example of a Janelia region containing two enhancers. The activity of the Janelia region is the result of both enhancers (highlighted) [Appendix Figure S15B]. **C.** Example of a Janelia region with multiple enhancers. The activity of the Janelia image is likely the result of the highlighted regions, but not the others [Appendix Figure S15C]. Aggregate profiles are shown on top, the accessibility of the Janelia region in the virtual cells (calculated by the aggregation of the region-cell probabilities of the regions within the Janelia region) and its activity, together with the accessibility of the highlighted regions, are shown below. Janelia enhancer activity images are taken from the Janelia Flylight project.

10) For the Prospero analysis: why did the authors use over-expression instead of RNAi for functional analysis? As Pros over-expression results in pupal lethality, this would be important to show the eye phenotype (combination with Gal80 to stage-specifically induce RNAi activity). The general question is whether the Pros data/part need to be included in this manuscript, in particular as this analysis suggests but not proves that Pros is a key

regulator of late-born photoreceptors. Leaving this data out and sparing it for another paper would have the advantage that the other parts could be extended and better explained.

Thank you, this is a relevant point.

- We had chosen for over-expression because we have performed a screen with many TFs, and some of these (e.g., Pros) are only expressed a very few cells in the tissue. With RNAi, we would not expect to see strong effects on accessibility using bulk ATAC-seq on the tissue, while with over-expression we could induce strong effects in many cells of the tissue, that are observable (as we show) using bulk ATAC-seq. To perform a similar analysis using RNAi, we would have to perform *single-cell* ATAC-seq on every perturbation, which would be very interesting, but outside the scope of this work. We have mention this in the text [Page: 26].
- We agree that performing conditional (Gal80) perturbations would be interesting to test the eye phenotype, but given that we would need to do this for many (all?) of the tested TFs in our screen, and given the minor revision (1 month deadline), this was not feasible. The pupal lethality of pros over-expression is indeed interesting to follow-up, but our analysis focused on accessibility in the larval disc, which we could assess before the lethality. For the phenotype of the pros knock-down, we refer to existing literature: the phenotype on minute clones of the *pros*¹⁷ null allele and RNAi pros knock-down present lenses with slightly irregular shapes and sizes (Charlton-Perkins et al., 2011).
- The reviewer suggests to remove the over-expression screen entirely. Based on the comments of the other reviewers, and the fact that the effect of over-expression is interpretable at the level of accessibility, we would prefer to keep these data in the paper, although with reduced emphasis and without over-interpretation. We find that the question "which TF could bind to the GGG motif" is a very common question, whereby motif discovery yields a new motif without a clear prediction of which TF could bind. We believe that our small screen for candidates is informative to pinpoint or prioritize candidates, and nicely corroborates with publicly available ChIP-seq data against Prospero in the embryo. If we remove the over-expression data, the ChIP-seq analysis alone would not be convincing, given that it is performed in the embryo. The two observations together do provide a significant data point in terms of experimental validation of computational motif predictions, and as such seem valuable to keep in the manuscript.

Minor points:

- acronyms need to be explained in main text & legends
- single cell vocabulary should be more accessible to researcher outside the field

Thank you. We have included explanations on acronyms and single cell terms on their first appearance on the text.

Reviewer #3:

In González-Blas et. al. the authors present a tour-de-force on the analysis and description of the eye-antennal disc in fly. The work is pioneering in the way the authors integrate the datasets they generated to not only uncover meaningful new biology, but to go beyond an atlas and dissect out core regulatory mechanisms, how those mechanisms can be altered and perturbed by genotype, and then directly test candidates to uncover a novel network of regulatory control. This paper is a great example of the power of single-cell genomics strategies used correctly as opposed to "single-cell for the sake of single-cell". My comments are all very minor, and I believe this work is well suited for publication and already seems very polished.

Overall the manuscript is very well written and very detailed. The authors present the work clearly and, while it is fairly long, I believe the length is warranted as it serves as a roadmap of sorts on how these types of data can, and should, be analyzed, including multiple analysis approaches to confirm and bolster their findings. Also the introduction captures the state of the field very nicely and the discussion puts this work into context nicely. Lastly, the methods are well detailed and the data is available.

Comment throughout - the text in the figures is a bit small - it may just be due to the pdf form for the submission.

Thank you for the positive evaluation. We have rearranged panels and increased the fonts to improve the quality of all the figures.

Figure 1c - the tSNE plots are quite small and a bit hard to interpret with the size

The size of this panel and the fonts in the overall figure have been increased for a better resolution.

For the ATAC data, two topics were batch effect topics - did these then get removed from the matrix for further analysis? Ie in the subsequent tSNE / cluster assignment / etc... Is it possible to remove the bias captured by the topic from the dataset?

The batch effect topics were not removed for the subsequent analysis. However, cell clustering and dimensionality reduction are similar upon removal of these topics [Figure R22], showing that cell clustering is driven by cell type rather than experimental run or sex. These data are included in Appendix Fig S8J [Page: 9].

Figure R22: Effect of the removal of batch effect topics on cell clustering. **A.** cisTopic cell tSNE using all topics (15,387 nuclei) colored by annotated cell type. **B.** cisTopic cell tSNE after removing the run-specific topics. **C.** cisTopic cell tSNE after removing the sex-specific topic. **D.** cisTopic cell tSNE after removing the run and the sex-specific topics [Appendix Figure S8J].

Line 301 - the finding of the "lag effect" is really interesting - could this be expanded upon by using pseudotemporal ordering approaches to capture these transitions and identify sets of loci that appear to open prior to expression and see if they are enriched for certain

activating TFs (eg Ato below)? Even without additional analysis, it is a very interesting finding.

Thank you for noticing this. We have performed additional analyses and found that this lag effect was rather an artifact of the binarization of the predicted labels [Figure R23]. When using label transferring with Seurat v3, labels are transferred with a certain likelihood, and the label with the highest likelihood is assigned. As we see in Figure R23, in the previously thought 'lagged cells' the probabilities of the consecutive clusters is almost 50% for each, making it difficult to assign a unique label. In fact, as cells differentiate in the eye disc it is rather artificial to define binary classes, and we observe that some cells are transitioning between the clusters. Overall, we find a strong agreement between the scRNA-seq and scATAC-seq. We have removed the sentence about the potential lag from the text.

Figure R23: ScRNA-seq label transferring into the scATAC-seq . A. cisTopic cell tSNE colored by the cell type annotation [Figure 2D]. **B.** cisTopic cell tSNE colored by the scRNA-seq annotation after label transferring with Seurat v3 using cisTopic gene accessibility matrix [Appendix Figure S12E]. **C.** Seurat label transferring prediction score

for the morphogenetic furrow, early photoreceptors and late photoreceptors and cone cells labels.

Figure 2 - some of the text is too small in this figure as well

We have rearranged the panels in this figure and sent to the supplementary figures the panels G [Appendix Figure S12B] and H [Appendix Figure S12E] for a better resolution.

Line 442 - are any of the *Janelia* enhancers highly negatively linked? These could be interesting.

Agreed, this is an interesting point. In fact, we do not find highly negative correlated enhancers, and those with negative correlation correspond to generally accessible primed regions [Page:11-12, Appendix Figure S16]. For example, in the most negatively correlated region (with correlation: -0.34) we found two enhancers with broad accessibility across different cell types, and overlap with Grh ChIP-seq data [Figure R24]. This may suggest that repressors that bind (and cause accessibility), often operate in separate regions that are not active as enhancers in a classical reporter assay (thus would be negative in the *Janelia* database). On the other hand, the even-skipped stripe 2 enhancer contains binding sites for activators and repressors, so this would be a counter-example.

Figure R24: Janelia regions lowly correlated with activity correspond to primed enhancers. Example of a Janelia region with negative correlation between accessibility and activity (with correlation: -0.34). Aggregate profiles and the Grh Chip-seq profile are shown on top, while the accessibility of the Janelia region in the virtual cells (calculated by the aggregation of the region-cell probabilities of the regions within the Janelia region) and its activity, together with the accessibility of the highlighted regions, are shown below. The Janelia enhancer activity image is taken from the Janelia Flylight project. [Appendix Figure S16]

Line 446 states an average of 13 enhancers per TF gene and 11 per other gene; above (line 440) states 22 median enhancers per gene. Is the discrepancy due to i) mean vs median or ii) the 22 are positive OR negative associations for each gene? This information is not contradictory, it could just be framed a little more clearly.

Indeed, the 22 median enhancers per gene consider both positive (Correlation > 0) and negative (Correlation < 0) annotations; while the average of 13 enhancers per TF gene and 11 for the remaining only considers positive associations. Per gene, we find a median of 11 positive enhancers (mean: 11.82) and a median of 11 negative enhancers (mean: 11.04). Per TF gene, we find a median of 12 positive enhancers (mean: 12.9) and a median of 10 negative enhancers (mean: 10.85). Per non-TF gene, we find a median of 11 positive enhancers (mean 11.73) and a median of 11 negative enhancers (11.06).

We have clarified the distinction between positive and negative links in each statement in this section [Page:13-14]. For this analysis we have used positively links, as positive relationships tend to be less noisy compared to negative relationships on single-cell data. This is mostly due to the sparse nature of single-cell technologies, as not observing gene expression/region accessibility may in part be due to technical drop-outs rather

Line 470 - "not correlated", but correlation is listed only as < 0.5, what is the actual number? 0.5 is not necessarily not correlated, just not strongly correlated.

Thank you, we have clarified this point in the text [Page:14]. For the promoters of differentially expressed genes, the average correlation is 0.20 (median:0.15).

I find the analysis of repressors particularly interesting. It would be very compelling to assess instances where genes are controlled by a mix of repressive and activating elements - eg element A - repressor and B - activator and identify cells with only A, only B, or A

and B. Broadly just understanding the balance between repression and activation and whether they compete at the same gene etc...

Thank you for the suggestion. We have compared two cases which involve high-confidence repressive relationships [Page: 15]. In the first example, we find that the potential repressive region is accessible in all the cells where the gene (*glass*, *gl*) is not expressed, while activating regions and the promoter are only accessible in the cell types where the gene is expressed. Interestingly, this potential repressive region loses accessibility on the photoreceptor precursor and morphogenetic furrow stages (*glass* is expressed in early and late photoreceptors and interommatidial cells). Altogether, in this instance the linked regions are either positively or negatively correlated with the expression of the gene (Fig R25).

In the second case study, we find that activating regions are broadly accessible (including in the cells where the gene (*homeothorax*, *hth*) is not expressed), while the repressive region is only accessible in the cells where the gene is repressed. In addition, as previous work suggests that Optix and Sine oculis co-repress *hth* (Anderson et al., 2012; Lopes & Casares, 2015), we compared our data with publicly available Sine oculis ChIP-seq data and our FAC-sorted Optix-GFP⁺ ATAC-seq profile, finding that repression is likely to occur via this region (Fig R26). Altogether, regions linked to *hth* have more diverse patterns, but clearly contain a mix of positive and negative regions.

Figure R25: Repression of *glass* in the antenna and anterior to the morphogenetic furrow. Example of a potential repressor region (marked with 2, highlighted in red) negatively linked to *glass*. The promoter of *glass* is highlighted in grey, and a positively

linked region (with 1) is highlighted in green. The expression of *glass* is shown on the virtual eye in red, and the accessibility of the previously mentioned enhancers (activating, 1; repressive 2) is shown in green.

Figure R26. *Hth* is potentially repressed by an enhancer bound by *so* and also accessible in FAC-sorted *Optix-GFP*⁺ cells. **A.** Enhancer-to-target links for the *hth* gene. From top to bottom are shown: position, link importance, cisTarget regions, gene annotation, *so* ChIP-seq data (replicate 1, replicate 2 and control), *Optix-GFP*⁺ cells bulk ATAC-seq profile, *Optix-GFP*⁻ cells bulk ATAC-seq profile and scATAC-seq aggregates by cell type. The repressive enhancer is highlighted in red, the promoter of *hth* is highlighted in grey. **B.** Virtual eye-antennal disc colored by the standardized gene expression of *hth*. **C.** Virtual eye-antennal disc colored by the standardized accessibility probability of the repressive enhancer. **D.** Virtual eye-antennal disc colored by the standardized gene expression of *so* (red) and *Optix* (green).

Paragraph at 490 - is there not a risk of a circular analysis in using the enhancer-gene links that were derived from correlating ATAC and RNA data to create a new gene activity matrix that then is compared with the RNA data? Maybe it is not used in that context, but it would seem that if one has RNA data that is required to generate the gene activity scores using this manner, then there is no real need to generate gene activity scores from ATAC data when the RNA data can just be used.

Thank you for bringing this up, we agree. The enhancer-gene pairs are the key result, and we now state that showing the predicted RNA based on these links is merely used as comparison between different approaches. For example, we removed enhancers-gene pairs based on their random forest importance and correlation, and still obtain accurate gene expression predictions. This way it is rather a technical validation and it avoids circularity [Page: 15]. We moved this figure to the supplement [Appendix Figure S20H].

The caQTL analysis is excellent and very well done. It is a great way to tease out the effects of variation in elements.

Thank you.

Accepted

18th March 2020

Thank you again for sending us your revised manuscript. We are now satisfied with the modifications made and I am pleased to inform you that your paper has been accepted for publication.

YOU MUST COMPLETE ALL CELLS WITH A PINK BACKGROUND ↓
PLEASE NOTE THAT THIS CHECKLIST WILL BE PUBLISHED ALONGSIDE YOUR PAPER

Corresponding Author Name: Stein Aerts
Journal Submitted to: Molecular Systems Biology
Manuscript Number: MSB-20-9438